# Bridge Matching Sampler:
# Scalable Sampling via Generalized Fixed-Point Diffusion Matching

**Denis Blessing** [1]  **Lorenz Richter** [2,3]  **Julius Berner** [4]  **Egor Malitskiy** [1]  **Gerhard Neumann** [1]

## Abstract

Sampling from unnormalized densities using diffusion models has emerged as a powerful paradigm. However, while recent approaches that use least-squares 'matching' objectives have improved scalability, they often necessitate significant trade-offs, such as restricting prior distributions or relying on unstable optimization schemes. By generalizing these methods as special forms of fixed-point iterations rooted in Nelson's relation, we develop a new method that addresses these limitations, called Bridge Matching Sampler (BMS). Our approach enables learning a stochastic transport map between arbitrary prior and target distributions with a single, scalable, and stable objective. Furthermore, we introduce a damped variant of this iteration that incorporates a regularization term to mitigate mode collapse and further stabilize training. Empirically, we demonstrate that our method enables sampling at unprecedented scales while preserving mode diversity, achieving state-of-the-art results on complex synthetic densities and high-dimensional molecular benchmarks.

## 1. Introduction

We consider the problem of sampling from a target distribution with density

$$p_{\text{target}}(x) = \frac{\rho_{\text{target}}(x)}{\mathcal{Z}},$$

where $\rho_{\text{target}} \in C(\mathbb{R}^d, \mathbb{R}_{\geq 0})$ can be evaluated pointwise and $\mathcal{Z} = \int_{\mathbb{R}^d} \rho_{\text{target}}(x) \, \mathrm{d}x$ is the associated normalizing constant, which is generally intractable. This problem is fundamental in computational science and underlies a wide range of applications, including molecular dynamics (Noé et al., 2019; Nam et al., 2025), statistical physics (Hénin et al., 2022; Faulkner & Livingstone, 2024), and Bayesian inference (Neal, 1993; Gelman et al., 2013).

Our goal is to learn a stochastic transport map from a tractable prior distribution $p_{\text{prior}}$, e.g., a Gaussian, to the target $p_{\text{target}}$. Formally, we seek a vector field $u$ such that the stochastic differential equation (SDE)

$$\mathrm{d}X_t = \sigma(t)u(X_t, t)\mathrm{d}t + \sigma(t)\mathrm{d}B_t, \quad X_0 \sim p_{\text{prior}}, \quad (1)$$

yields a terminal marginal $\mathbb{P}^u_T = p_{\text{target}}$, where $\mathbb{P}^u$ denotes the induced path measure. Several approaches address this problem by introducing a corresponding time-reversed process

$$\mathrm{d}Y_t = -\sigma(t)v(Y_t, t)\mathrm{d}t + \sigma(t)\overleftarrow{\mathrm{d}}B_t, \quad Y_T \sim p_{\text{target}}, \quad (2)$$

with path measure $\overleftarrow{\mathbb{P}}^v$ (Richter & Berner, 2024; Vargas et al., 2024; Blessing et al., 2025b). These methods attempt to align the forward and backward path measures by minimizing a divergence $D(\mathbb{P}^u, \overleftarrow{\mathbb{P}}^v)$ over the controls $u$ and $v$. This optimization yields (generally non-unique) optimal controls $u^*$ and $v^*$ that transport the prior to the target and vice versa. However, the practical implementation of such objectives is often computationally prohibitive. The optimization typically requires the repeated simulation and storage of full trajectories of the SDE (1) to compute gradients – a bottleneck that becomes particularly challenging in high-dimensional settings.

In contrast, approaches based on bridge processes and their associated path measures admit substantially more efficient simulation procedures. Specifically, one can consider path measures of the form[1]

$$\Pi^*(X) = p_{\text{prior}}(X_0)p_{\text{target}}(X_T)\mathbb{P}_{|0,T}(X|X_0, X_T), \quad (3)$$

where $\mathbb{P}_{|0,T}$ corresponds to a tractable reference process conditioned on its endpoints, e.g., a Brownian bridge.

The measure $\Pi^*$ can be represented as the distribution of a non-Markovian stochastic process of the form

$$\mathrm{d}X_t = \sigma(t)\xi(X, t)\mathrm{d}t + \sigma(t)\mathrm{d}B_t, \quad X_0 \sim p_{\text{prior}}, \quad (4)$$

where the functional $\xi$ may depend on the entire path, and

---

[1]Karlsruhe Institute of Technology [2]Zuse Institute Berlin [3]dida [4]NVIDIA. Correspondence to: Denis Blessing <denis.blessing@kit.edu>.

*Proceedings of the 43$^{rd}$ International Conference on Machine Learning*, Seoul, South Korea. PMLR 306, 2026. Copyright 2026 by the author(s).

---

[1]This can be viewed as the *reciprocal projection* of a process with independent coupling onto the reference process $\mathbb{P}$; see Definition 2.1.

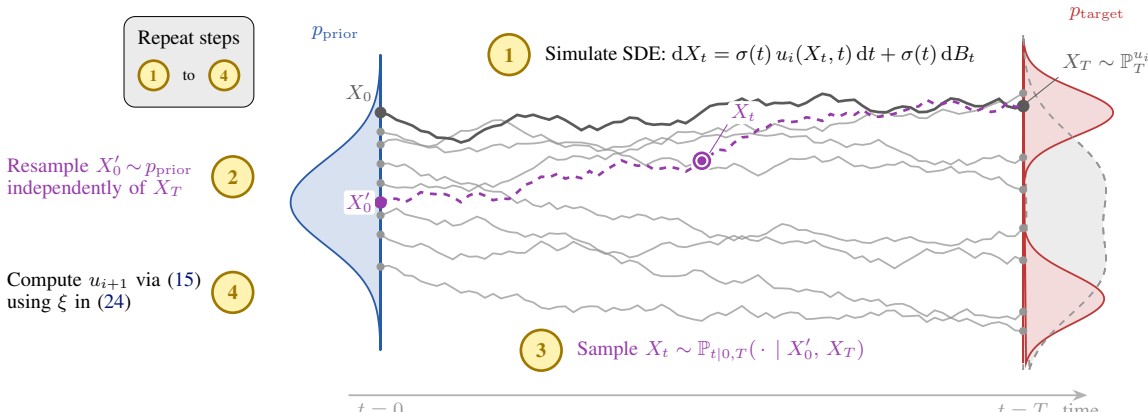

*Figure 1.* **Bridge Matching Sampler (BMS).** Illustration of one outer iteration of our fixed-point algorithm. Starting from the current drift $u_i$, we (1) *simulate* the controlled forward SDE, $dX_t = \sigma(t) u_i(X_t, t) dt + \sigma(t) dB_t$, starting from $p_{\text{prior}}$, to obtain $K$ trajectories with endpoints $(X_0, X_T)$ (grey curves); (2) *resample* the prior endpoints, replacing $X_0$ by an independent sample $X_0' \sim p_{\text{prior}}$ to form the coupling $\Pi_{0,T}^i = p_{\text{prior}} \otimes \mathbb{P}_T^{u_i}$; (3) *sample* intermediate states $X_t \sim \mathbb{P}_{t|0,T}(\cdot \mid X_0', X_T)$ from the associated Brownian bridge (dashed purple); and (4) *update* the drift to $u_{i+1}$ by regressing it on the path-dependent target $\xi(X, t)$ via the matching objective described in (15). Iterating steps 1-4 drives the time marginals of the learned process towards $p_{\text{target}}$ while maintaining a single, scalable, and stable training objective.

in particular on the terminal value $X_T$, e.g.,

$$\xi(X, t) = \sigma(t) \left( \int_t^T \sigma^2(s) \, ds \right)^{-1} \left( X_T - X_t \right) \quad (5)$$

in the case of a Brownian bridge $\mathbb{P}_{|0,T}$; see, Albergo et al. (2025); Shi et al. (2023); Liu et al. (2023). By construction, the dependence in (3) enforces the terminal marginal $X_T \sim p_{\text{target}}$; however, the process $X$ in (4) can only be simulated when samples $X_T \sim p_{\text{target}}$ are available.

The key idea pursued here is to *Markovianize* (see Definition 2.2) the non-Markovian dynamics in (4) by minimizing suitable loss functionals, yielding a Markovian control $u$ with associated path measure $\mathbb{P}^u$ as in (1). The objective is to match the time marginals, namely $\mathbb{P}_t^u = \Pi_t^*$ for all $t \in [0, T]$.

Although derived from a different perspective, several recent works employ related ideas to obtain simulation-free training objectives. Most existing approaches, however, rely on importance sampling (Akhound-Sadegh et al., 2024; OuYang et al., 2024; Woo & Ahn, 2024; Wang et al., 2025; Dern et al., 2025), which is known to deteriorate in high dimensions (Chatterjee & Diaconis, 2018; Hartmann & Richter, 2024).

In this work, we introduce a scalable algorithm for Markovianization, which we term *Bridge Matching Sampler* (BMS). Our method does not use importance sampling nor require samples from the target distribution, but only access to its unnormalized density. Moreover, our method can be viewed as a generalization of recently proposed sampling algorithms based on least-squares fixed-point iterations (often referred to as *matching* algorithms) arising from stochastic optimal control theory (Havens et al., 2025; Liu et al., 2025; Nam et al., 2025). In contrast to these approaches, however, our framework accommodates general prior distributions

*Table 1.* Overview: Our framework considers target path measures of the form $\Pi^* = \Pi_{0,T}^* \mathbb{P}_{|0,T}$ with different couplings $\Pi_{0,T}^*$. We provide expressions for $\xi$ that satisfy $u^*(x, t) = \mathbb{E}_{\Pi^*} [\xi(X, t) | X_t = x]$ and are used for the proposed fixed-point iteration scheme in Algorithm 1. For an overview of the different expressions for $\xi$ see Table 5 in Appendix E.

| Coupling | | $\xi$ | Markovian | Algorithm |
|---|---|---|---|---|
| General | $\Pi_{0,T}^*$ | Prop. 2.7 | ✗ | |
| Schrödinger (half) | $\Pi_{0,T}^* = \mathbb{P}_0 \otimes \Pi_T^*$ | Prop. 2.8 | ✓ | AS (Havens et al., 2025) |
| Schrödinger | $\Pi_{0,T}^* = \Pi_{0,T}^{SB}$ | Prop. 2.10 | ✓ | ASBS (Liu et al., 2025) |
| Independent | $\Pi_{0,T}^* = \Pi_0^* \otimes \Pi_T^*$ | Prop. 2.11 | ✗ | BMS (ours) |

and reference measures and avoids alternating optimization schemes, which are frequently unstable in practice. To further enhance stability and robustness, we introduce a damped variant of the fixed-point iteration that regularizes the update steps. As a result, BMS enables diffusion-based sampling methods to scale to higher-dimensional and more challenging problems, while outperforming existing state-of-the-art samplers in both accuracy and computational efficiency.

## 2. Generalized Fixed-Point Diffusion Matching

**Notation.** We denote by $\mathcal{U} \subset C(\mathbb{R}^d \times [0, T], \mathbb{R}^d)$ the set of admissible controls and by $\mathcal{P}$ the set of all probability measures on $C([0, T], \mathbb{R}^d)$. Moreover, we define $\mathcal{M} \subset \mathcal{P}$ to be the subset of Markov measures, i.e., the laws of all Markov processes. We define the path space measure $\mathbb{P} \in \mathcal{P}$ as the law of a $\mathbb{R}^d$-valued stochastic process $X = (X_t)_{t \in [0,T]}$ and we denote by $\mathbb{P}_s$ its marginal distribution at time $s$. Moreover, we denote by $\mathbb{P}_{|0,T}$ the path measure $\mathbb{P}$ conditioned on the random variables $X_0$ and $X_T$. We refer to Appendix A for further details on our notation and assumptions.

**Overview.** To provide a roadmap for the theoretical developments that follow, let us first present a high-level overview of our framework. We propose a fixed-point iteration that alternates between the construction of valid bridge processes and their Markovianization to update the control fields. Conceptually, each iteration $i$ consists of three steps (see also Algorithm 1 for a summary):

1. *Simulation and coupling*: We sample $X_0 \sim p_{\text{prior}}$ from the prior and simulate the SDE in (1) using the current control $u_i$ to obtain $X_T$. Together with a sample from the prior, this defines the current coupling.

2. *Reciprocal projection and target drift*: Conditioned on the coupling, we use a reference process $\mathbb{P}_{|0,T}$ to define a bridge process $\Pi^i$. Our generalized target score identity in Proposition 2.6 can now be used to derive the drift $\xi(X,t)$ for the process in (4) that corresponds to the target measure $\Pi^*$; see Table 1.

3. *Markovianization*: Finally, we obtain $u_{i+1}$ by minimizing a least-squares regression objective that fits the Markovian control $u_i$ to the target drift $\xi$ evaluated at samples drawn from the bridge process.

To present our general framework, which alternates between the construction of bridge processes – also known as *reciprocal projections* – and the computation of Markovian projections, we first introduce the relevant definitions.

**Definition 2.1** (Reciprocal class and reciprocal projection). A path measure $\Pi \in \mathcal{P}$ is in the *reciprocal class* $\mathcal{R}(\mathbb{P})$ of the reference process $\mathbb{P} \in \mathcal{M}$ if $\Pi = \Pi_{0,T}\mathbb{P}_{|0,T}$. We define the *reciprocal projection* of $\Pi$ onto $\mathcal{R}(\mathbb{P})$ as

$$\text{proj}_{\mathcal{R}(\mathbb{P})}(\Pi) := \Pi_{0,T}\mathbb{P}_{|0,T}. \qquad (6)$$

It is important to note that, for suitable reference measures $\mathbb{P}$, the conditioned process $\mathbb{P}_{|0,T}$ can be simulated efficiently. For instance, when $\mathbb{P}$ corresponds to (scaled) Brownian motion, the conditioned measure $\mathbb{P}_{|0,T}$ is the classical *Brownian bridge*, for which the non-Markovian drift $\xi$ admits the closed-form expression stated in (5). Moreover, for any $t \in [0,T]$, the marginal distribution $X_t \sim \mathbb{P}_{t|0,T}$ can be sampled directly, without integrating full trajectories. This eliminates the need to store entire paths in memory and leads to substantial computational savings, especially in high-dimensional settings.

In our general framework, we consider a (possibly non-Markovian) target path measure $\Pi^*$ subject to the boundary constraints

$$p_{\text{prior}}(x_0) = \int_{\mathbb{R}^d} \Pi^*_{0,T}(x_0, x_T)\,\mathrm{d}x_T, \quad \text{and} \qquad (7)$$

$$p_{\text{target}}(x_T) = \int_{\mathbb{R}^d} \Pi^*_{0,T}(x_0, x_T)\,\mathrm{d}x_0. \qquad (8)$$

Together with the bridge process introduced above, these

---

**Algorithm 1** Generalized Fixed-Point Diffusion Matching

**Require:** Initial $u_0$, number of iterations $I$
**Require:** Coupling $\Pi^*_{0,T}$ and corresponding $\xi$ (see Table 1)
  **for** $i \leftarrow 0, \ldots, I-1$ **do**
    *Simulation:* Simulate $X \sim \mathbb{P}^{u_i}$
    *Coupling:* Let $\Pi^i_{0,T} = \begin{cases} \mathbb{P}^{u_i}_{0,T}, & \text{ASBS (Liu et al., 2025)} \\ \mathbb{P}_0 \otimes \mathbb{P}^{u_i}_T, & \text{AS (Havens et al., 2025)} \\ p_{\text{prior}} \otimes \mathbb{P}^{u_i}_T, & \text{BMS (ours)} \end{cases}$
    *Buffer:* Store $(X_0, X_T) \sim \Pi^i_{0,T}$
    *Reciprocal projection:* Let $\Pi^i = \Pi^i_{0,T}\mathbb{P}_{|0,T}$
    *Markovianization:*
    $$u_{i+1} = \arg\min_{u \in \mathcal{U}} \mathbb{E}_{\Pi^i}\left[\int_0^T \tfrac{1}{2}\|\xi(X,t) - u(X_t,t)\|^2\mathrm{d}t\right]$$
  **end for**
  **Return** $u_I$

---

constraints define a target path measure

$$\Pi^* = \Pi^*_{0,T}\mathbb{P}_{|0,T} \in \mathcal{R}(\mathbb{P}), \qquad (9)$$

which can be represented as the law of the stochastic process

$$\mathrm{d}X_t = \sigma(t)\xi(X,t)\mathrm{d}t + \sigma(t)\mathrm{d}B_t, \quad X_0 \sim p_{\text{prior}}. \quad (10)$$

Here, the drift $\xi$ is a *path-dependent functional*, potentially depending on the entire trajectory $(X_t)_{t \in [0,T]}$. In the absence of samples from $p_{\text{target}}$, constructing a valid $\xi$ is non-trivial. We will provide a feasible representation later.

Our objective is to approximate $\Pi^*$ by a Markov diffusion process of the form

$$\mathrm{d}X_t = \sigma(t)u^*(X_t,t)\mathrm{d}t + \sigma(t)\mathrm{d}B_t, \qquad (11)$$

whose time marginals coincide with those of the non-Markovian process (10). To formalize this approximation, we next introduce the notion of *Markovian projections*; see, e.g., Gyöngy (1986); Shi et al. (2023); Liu et al. (2022); Peluchetti (2023).

**Definition 2.2** (Markovian projection). Let $\Pi^*$ be the path measure induced by the solution to the path-dependent SDE (10). The *Markovian projection* of $\Pi^*$ is a path measure $\mathbb{P}^{u^*}$ corresponding to the solution of the Markovian SDE (11) such that for every $t \in [0,T]$, the marginal distributions of both measures coincide, i.e.

$$\Pi^*_t = \mathbb{P}^{u^*}_t. \qquad (12)$$

**Lemma 2.3** (Formula for Markov control). Suppose $X$ solves the path-dependent SDE (10) where $\xi(X,t)$ is a functional of the path $X$ and $\Pi^*$ is the associated non-Markovian path measure. The drift $u^*$ of the corresponding Markovian process is given by the conditional expectation

$$u^*(x,t) = \mathbb{E}_{\Pi^*}\left[\xi(X,t)|X_t = x\right]. \qquad (13)$$

Furthermore, it can be shown that

$$u^* = \arg\min_{u \in \mathcal{U}} \; D_{\text{KL}}(\Pi^*|\mathbb{P}^u). \qquad (14)$$

We refer to Brunick & Shreve (2013) for a proof. While we do not have access to samples from $\Pi^*$, the Markovian projection formula (13) naturally suggests the fixed-point iteration

$$u_{i+1} = \Phi(u_i), \qquad (15)$$

where

$$\Phi(u_i) := \mathbb{E}_{\Pi^i}\left[\xi(X,t)|X_t\right]$$
$$= \arg\min_{u \in \mathcal{U}} \mathbb{E}_{\Pi^i}\left[\int_0^T \tfrac{1}{2}\|\xi(X,t) - u(X_t,t)\|^2 \mathrm{d}t\right]$$

and $\Pi^i := \Pi^i_{0,T}\mathbb{P}_{|0,T} \in \mathcal{R}(\mathbb{P})$. In other words, the method alternates between reciprocal projections and Markovianization, see also Algorithm 1. By construction, if $u_i = u^*$, then $u_{i+1} = u^*$, and hence $u^*$ is a fixed point of the iteration. To make the above fixed-point iteration tractable, however, we need an expression for the non-Markovian drift $\xi$.

**Remark 2.4** (Data-free version of stochastic interpolants). *We note that the Bridge Matching Sampler may be interpreted as a data-free version of stochastic interpolant methods (Albergo et al., 2025). In our setting no samples from the target measure, but only an unnormalized target density, are available, which makes training less straightforward. We refer to Appendix D.2 for details on this connection.*

### 2.1. General couplings

To derive expressions for the path-dependent drift $\xi$ in the non-Markovian SDE (10), we first recall that the optimal Markovian drifts $u^*$ and $v^*$ satisfy Nelson's identity

$$u^*(\cdot,t) + v^*(\cdot,t) = \sigma(t)\nabla \log \mathbb{P}_t^{u^*} = \sigma(t)\nabla \log \Pi_t^*, \quad (16)$$

where $\Pi_t^*$ denotes the marginal of the target path measure at time $t$. Since the time marginals are fixed and by uniqueness of the Markovian projection, the drifts $u^*$ and $v^*$ are uniquely determined for a given coupling $\Pi^*_{0,T}$. They can be characterized explicitly as follows; we refer to Appendix B.1 for the proof.

**Proposition 2.5** (Markovian projections). *Let $\Pi^* \in \mathcal{R}(\mathbb{P})$. The unique optimal drifts $u^*$ and $v^*$ corresponding to the forward and backward Markovian projections are*

$$u^*(x,t) = \mathbb{E}_{\Pi^*_{T|t}}\left[\sigma(t)\nabla_{X_t} \log \mathbb{P}_{T|t}(X_T|X_t)\Big|X_t = x\right]$$

*and*

$$v^*(x,t) = \mathbb{E}_{\Pi^*_{0|t}}\left[\sigma(t)\nabla_{X_t} \log \mathbb{P}_{t|0}(X_t|X_0)\Big|X_t = x\right].$$

Note that both $\nabla_{X_t} \log \mathbb{P}_{T|t}$ and $\nabla_{X_t} \log \mathbb{P}_{t|0}$ are tractable if the reference process is scaled Brownian motion; see Appendix E.5. Following the approach of bridge matching (Liu et al., 2022; Peluchetti, 2023), one might be tempted to set

$$\xi(X,t) = \sigma(t)\nabla_{X_t} \log \mathbb{P}_{T|t}(X_T|X_t) \qquad (17)$$

to perform the fixed-point iteration in (15). However, $\Phi(u_i)$

evaluates the expectation with respect to $\Pi^i$ rather than the true target measure $\Pi^*$. Since the Markovian projection is marginal preserving, the coupling would remain static across all iterations, resulting in $\Pi^i_{0,T} = \Pi^0_{0,T}$ for all $i$. Intuitively, this behavior is to be expected: when the expectation is taken over $\Pi^i$, the expressions in Proposition 2.5 do not incorporate any information about the prior or target boundary densities.

Instead of relying on Proposition 2.5 alone, we leverage Nelson's identity (16) and treat $v^*$ and $\nabla \log \Pi_t^*$ separately. While $v^*$ has already been characterized in Proposition 2.5, we derive an identity for $\nabla \log \Pi_t^*$ in the following proposition. We term this identity *generalized target score identity* (TSI) since it allows us to recover and generalize many of the *target score identities* in the literature (De Bortoli et al., 2024); we refer to Appendix B.1 for the proof

**Proposition 2.6** (Generalized target score identity). *Let the reference measure $\mathbb{P}$ be induced by scaled Brownian motion, $\mathrm{d}X_t = \sigma(t)\mathrm{d}B_t$, and define $\kappa(t) := \int_0^t \sigma^2(s)\,\mathrm{d}s, \gamma(t) := \frac{\kappa(t)}{\kappa(T)}$. Let $c \in C([0,T],(0,1))$ be an arbitrary function[2]. Then, for the target path measure $\Pi^* = \Pi^*_{0,T}\mathbb{P}_{|0,T}$, it holds*

$$\nabla_x \log \Pi_t^*(x) = \mathbb{E}_{\Pi^*_{0,T|t}}\left[\frac{1-c(t)}{1-\gamma(t)}\nabla_{X_0} \log \Pi^*_{0,T}(X_0, X_T)\right.$$
$$\left. + \frac{c(t)}{\gamma(t)}\nabla_{X_T} \log \Pi^*_{0,T}(X_0, X_T)\Big|X_t = x\right].$$

Combining Proposition 2.5 and Proposition 2.6, we can now get an expression for the non-Markovian drift $\xi$.

**Proposition 2.7** (Path-dependent drift of general target measure). *Under the assumptions of Proposition 2.6, the path-dependent drift $\xi$ corresponding to the non-Markovian SDE (10) that induces the target measure $\Pi^*$ is given by*

$$\sigma(t)^{-1}\xi(X,t) = \frac{1-c(t)}{1-\gamma(t)}\nabla_{X_0} \log \Pi^*_{0,T}(X_0, X_T)$$
$$+ \frac{c(t)}{\gamma(t)}\nabla_{X_T} \log \Pi^*_{0,T}(X_0, X_T) - \nabla_{X_t} \log \mathbb{P}_{t|0}(X_t|X_0).$$

While one could, in principle, use Proposition 2.7 to perform the fixed-point iteration suggested in Algorithm 1, in practice we typically do not have access to the coupling scores $\nabla \log \Pi^*_{0,T}$. However, for certain special forms of $\Pi^*_{0,T}$, these scores can be computed explicitly, leading to tractable algorithms for learning the stochastic transport map $u^*$. The remainder of this section focuses on these tractable cases.

---

[2]The function $c \in C([0,T],(0,1))$ can be seen as a control variate; see Appendix E.2 for further details.

## 2.2. Schrödinger half bridges

*Schrödinger half bridges* (SHBs) solve problems of the form

$$\min_{u \in \mathcal{U}} \mathbb{E}_{\mathbb{P}^u} \left[ \int_0^T \tfrac{1}{2} \|u(X_t, t)\|^2 \, \mathrm{d}t \right] \text{ s.t. } X_T \sim \Pi_T^* = p_{\text{target}},$$

i.e., they are characterized by a kinetic optimal drift given by a marginal constraint[3]. Using that

$$D_{\text{KL}}(\mathbb{P}^u | \mathbb{P}) = \mathbb{E}_{\mathbb{P}^u} \left[ \int_0^T \tfrac{1}{2} \|u(X_t, t)\|^2 \, \mathrm{d}t \right], \quad (18)$$

and by the chain rule of the KL divergence it is straightforward to see that the minimum is attained when the conditional measures match, $\mathbb{P}_{|T}^u = \mathbb{P}_{|T}$. Consequently, the optimal joint coupling is given by $\Pi_{0,T}^* = \Pi_T^* \mathbb{P}_{0|T}$. However, this implies that the initial marginal is $\Pi_0^* = \mathbb{E}_{\Pi_T^*}[\mathbb{P}_{0|T}]$, which generally depends on the target $\Pi_T^*$ and differs from the reference prior $\mathbb{P}_0$. To ensure that the optimal model actually starts at the prior (i.e., $\Pi_0^* = p_{\text{prior}}$), the reference must satisfy the independence condition $\mathbb{P}_{0|T} = \mathbb{P}_0$, a property commonly known as *memorylessness* (Domingo-Enrich et al., 2025). For instance, this can be enforced by using a deterministic initial condition $X_0 = x_0$, i.e. $p_{\text{prior}} = \delta_{x_0}$. Using the Brownian motion reference process and $x_0 = 0$ – a setting that is considered frequently in the literature – the generalized TSI in Proposition 2.6 then becomes

$$\nabla \log \Pi_t^*(x) = \mathbb{E}_{\Pi_{T|t}^*} \left[ \frac{1}{\gamma(t)} \nabla_{X_T} \log \Pi_T^*(X_T) | X_t = x \right].$$

Combining this with the general results for $v^*$ in Proposition 2.5 yields the following result, see Appendix B.3 for the proof.

**Proposition 2.8** (Path-dependent drift for Schrödinger half bridges). *Under the assumptions of Proposition 2.6, the path-dependent drift $\xi$ corresponding to the non-Markovian SDE (10) that induces the target measure $\Pi^* = \Pi_T^* \mathbb{P}_{|T}$ is given by*

$$\sigma(t)^{-1} \xi(X, t) = \nabla_{X_T} \log \frac{\Pi_T^*(X_T)}{\mathbb{P}_T(X_T)}. \quad (19)$$

We note that the result in Proposition 2.8, together with the associated fixed-point iteration, coincides with the *adjoint sampling* (AS) method introduced by Havens et al. (2025). While their derivation is based on a stochastic optimal control (SOC) formulation combined with the adjoint method, the same scheme arises in our setting as a special case of the general framework developed here. However, as mentioned above, the memorylessness assumptions severely restricts the choice of the reference process and the prior distribution and, in practice, these choices lead to instabilities due to large diffusion coefficients $\sigma$. In particular, for the Dirac delta prior, we need a sufficiently large coefficient to guar-

---

[3]One can also consider constraining the marginal at time $t = 0$.

antee initial exploration of the sampler. For a Gaussian prior, one requires sufficiently large diffusion coefficients such that the distribution of the time-reversed process approximates the prior at finite time $T$. In the following, we present a method that allows for the use of arbitrary prior distributions and reference processes, thereby generalizing the previously introduced Schrödinger half-bridges.

## 2.3. Schrödinger bridges

Similar to half-bridges, *Schrödinger bridges* (SBs) seek a kinetically optimal drift, however, containing both boundary constraints, i.e.,

$$\min_{u \in \mathcal{U}} \mathbb{E} \left[ \int_0^T \tfrac{1}{2} \|u(X_t, t)\|^2 \, \mathrm{d}t \right] \text{ s.t. } \begin{cases} X_0 \sim \Pi_0^* = p_{\text{prior}}, \\ X_T \sim \Pi_T^* = p_{\text{target}}. \end{cases}$$

The solution to the SB problem can be characterized using the Schrödinger system stated in the following proposition; see, e.g., Léonard (2013); Chen et al. (2016), and Appendix B.2.

**Proposition 2.9** (Schrödinger system). *Let $\varphi, \widehat{\varphi} \in C(\mathbb{R}^d \times [0, T]; \mathbb{R})$ satisfy the Schrödinger system, i.e.,*

$$\varphi_t(x) = \int \mathbb{P}_{T|t}(x_T | x) \varphi_T(x_T) \mathrm{d}x_T, \quad \varphi_0 \widehat{\varphi}_0 = \Pi_0^*, \quad (20)$$

$$\widehat{\varphi}_t(x) = \int \mathbb{P}_{t|0}(x | x_0) \widehat{\varphi}_0(x_0) \mathrm{d}x_0, \quad \varphi_T \widehat{\varphi}_T = \Pi_T^*, \quad (21)$$

*then $u^* = \sigma \nabla \log \varphi$ and $v^* = \sigma \nabla \log \widehat{\varphi}$. Moreover, the Schrödinger coupling satisfies*

$$\Pi_{0,T}^* = \Pi_{0,T}^{\text{SB}} := \mathbb{P}_{T|0} \widehat{\varphi}_0 \varphi_T. \quad (22)$$

Combining the generalized TSI in Proposition 2.6 and (22), it is straightforward to define a TSI for SBs and derive the path-dependent drift, see Appendix B.2 in the appendix for further details, where we also state a more general version in Proposition B.5.

**Proposition 2.10** (Path-dependent drift for Schrödinger bridges). *Under the assumptions of Proposition 2.6, the path-dependent drift $\xi$ corresponding to the non-Markovian SDE (10) that induces the target measure $\Pi^* = \Pi_{0,T}^{\text{SB}} \mathbb{P}_{|0,T}$ is given by*

$$\sigma(t)^{-1} \xi(X, t) = \nabla_{X_T} \log \frac{\Pi_T^*(X_T)}{\widehat{\varphi}_T(X_T)}. \quad (23)$$

The result in Proposition 2.10, together with the associated fixed-point iteration, recovers the *Adjoint Schrödinger Bridge Sampler* (ASBS) introduced by Liu et al. (2025), which was originally derived from a stochastic optimal control perspective. While Schrödinger bridges (SBs) allow for arbitrary prior distributions $p_{\text{prior}}$, their associated fixed-point iteration requires access to the terminal potential $\widehat{\varphi}_T$, which is typically unavailable. To address this, Liu et al. (2025) proposed a variant based on *iterative proportional fitting* (IPF) (Kullback, 1968), which alternates between

updating the drift $u$ and the terminal potential $\widehat{\varphi}_T$. Because these updates are interdependent, the resulting algorithm can be unstable. In the following, we propose a method that enables the use of arbitrary prior and target distributions without requiring such alternating optimization schemes.

## 2.4. Independent couplings

Recalling Proposition 2.7, which states a path-dependent drift for general target measures, let us now consider the special case when the coupling is assumed to be independent, i.e.

$$\Pi^*_{0,T} = \Pi^*_0 \otimes \Pi^*_T = p_{\text{prior}} \otimes p_{\text{target}}.$$

In this case, the coupling scores $\nabla \log \Pi^*_{0,T}$ become tractable, which directly yields the following proposition, see also Appendix B.4.

**Proposition 2.11** (Target score and drift for independent couplings). *Under the assumptions of Proposition 2.6, and additionally assuming that the coupling $\Pi^*_{0,T}$ factorizes as $\Pi^*_{0,T} = \Pi^*_0 \otimes \Pi^*_T$, the following identity holds:*

$$\nabla_x \log \Pi^*_t(x) = \mathbb{E}_{\Pi^*_{0,T|t}} \left[ \frac{1 - c(t)}{1 - \gamma(t)} \nabla_{X_0} \log \Pi^*_0(X_0) \right.$$
$$\left. + \frac{c(t)}{\gamma(t)} \nabla_{X_T} \log \Pi^*_T(X_T) \Big| X_t = x \right].$$

*Consequently, the associated drift is given by*

$$\sigma(t)^{-1} \xi(X, t) = \frac{1 - c(t)}{1 - \gamma(t)} \nabla_{X_0} \log \Pi^*_0(X_0)$$
$$+ \frac{c(t)}{\gamma(t)} \nabla_{X_T} \log \Pi^*_T(X_T) - \nabla_{X_t} \log \mathbb{P}_{t|0}(X_t|X_0).$$
$$(24)$$

With Proposition 2.11, we obtain a tractable path-dependent drift $\xi$ corresponding to the non-Markovian target measure $\Pi^*$, which can be directly employed in Algorithm 1. This allows us to pick arbitrary prior distributions and reference processes, including arbitrary diffusion coefficients $\sigma$, while still retaining a single, scalable matching objective. Together with the damping outlined in the next section, this defines our proposed *Bridge Matching Sampler* (BMS). A detailed description of the algorithm is provided in Algorithm 2 in Appendix F, which additionally highlights how the integration of replay buffers enables data reuse, thereby significantly increasing the method's sample efficiency.

**Remark 2.12** (Numerical (in-)sensitivity of the score function). *We note that our algorithm, as well as AS and ASBS, incorporates the target measure only through its derivative rather than through the density itself; see, e.g., (24). However, it is well known that the score function need not encode the same structural information as the density and may therefore be (almost) "blind" to certain characteristics of the target measure; cf. Wenliang & Kanagawa (2020); Zhang et al. (2022); Shi et al. (2024); Grenioux et al. (2026). In particular, when distinct target modes are separated by large regions of negligible probability mass, the score function yields only vanishingly small learning signals in these transition regions. Consequently, the resulting optimization problem may become numerically insensitive to important global features of the target distribution. An interesting direction for future research would therefore be to mitigate this issue by incorporating more direct information about the target density into the loss function.*

## 2.5. Damped fixed-point diffusion matching

To update the control gradually and thereby improve the robustness of the fixed-point iteration, we extend (15) by introducing the *damped fixed-point iteration*

$$u_{i+1} = \alpha \, \Phi(u_i) + (1 - \alpha) u_i, \qquad (25)$$

where $\alpha \in (0, 1]$ is the step size. This damped scheme admits a variational characterization, as formalized in the following proposition, see Appendix B.5 for the proof.

**Proposition 2.13** (Variational characterization of damped fixed-point iteration). *Consider the fixed-point operator $\Phi$ defined in (15). Then, the damped fixed-point iteration (25) satisfies that*

$$u_{i+1} = \arg\min_{u \in \mathcal{U}} \left\{ \mathbb{E}_{\Pi^i} \left[ \int_0^T \tfrac{1}{2} \|\xi(X, t) - u(X_t, t)\|^2 \mathrm{d}t \right] \right.$$
$$\left. + \eta \, \mathbb{E}_{\Pi^i} \left[ \int_0^T \tfrac{1}{2} \|u_i(X_t, t) - u(X_t, t)\|^2 \mathrm{d}t \right] \right\},$$

*with $\Pi^i = \Pi^i_{0,T} \mathbb{P}_{|0,T}$ and damping parameter $\eta = \frac{1 - \alpha}{\alpha}$.*

This variational perspective shows that the damped iteration regularizes the current drift to remain close to the previous iterate $u_i$ in the $L^2$ sense. In Appendix E.3, we show that the damped update reduces the variance of the fixed-point iteration. Similar regularization techniques are common in proximal point methods (Parikh et al., 2014; Hertrich & Gruhlke, 2024) and trust region methods (Conn et al., 2000; Schulman et al., 2015), which have recently gained traction in the diffusion model literature (Guo et al., 2025a; Blessing et al., 2025a). For further details see Appendix D.1.

In practice, damping is integrated into Algorithm 1 by modifying the Markovianization step and storing the parameters of the previous iteration to evaluate $u_i$. As shown in our experiments (Section 4), this modification substantially improves the performance of both baseline methods and our proposed approach, particularly in high-dimensional settings. We refer to Algorithm 2 for a detailed description of the algorithm, including the damped fixed-point iteration.

## 3. Related work

**Diffusion-based sampling.** Early work on diffusion-based sampling methods that use regression-based matching losses was proposed in Akhound-Sadegh et al. (2024); OuYang

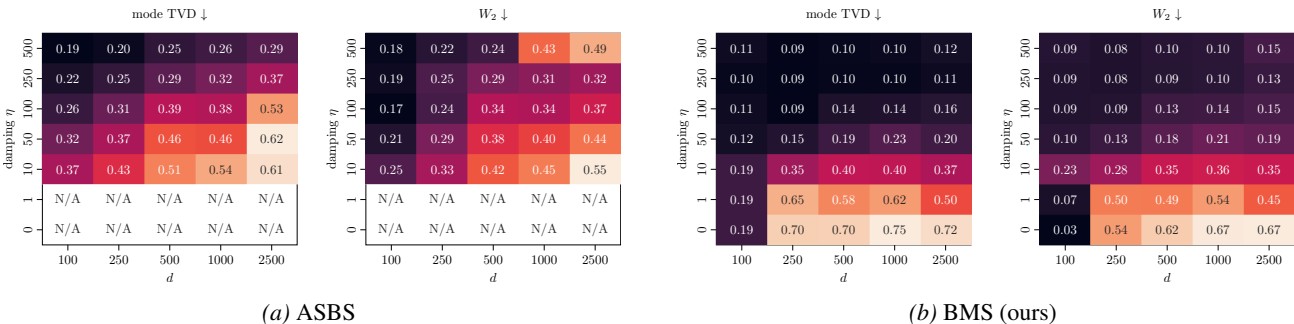

*(a)* ASBS  *(b)* BMS (ours)

*Figure 2.* Comparison of *mode TVD* and *Wasserstein-2 ($W_2$)* values on Gaussian mixture models across varying dimensions $d$ and damping values $\eta$. Figure 2a shows that ASBS exhibits instability at low damping values (N/A) and performance degradation as dimensionality increases. Figure 2b demonstrates that our method maintains stable training and consistently avoids mode collapse up to $d = 2500$, significantly scaling beyond the $d = 50$ range common in the recent literature. All values are averaged over three random seeds.

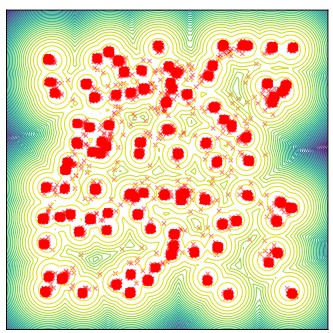

*Figure 3.* Samples obtained with BMS, projected on the first two dimensions of a GMM with $d = 100$ and 100 modes.

*Table 2.* Comparison on $n$-body particle systems across varying dimensions $d$. We report the Wasserstein-2 distance between model and long run MCMC samples ($W_2$) and energy values ($E(\cdot)\ W_2$). Results are averaged over three runs, with the best performing method highlighted in bold. Baseline results are quoted from Liu et al. (2025); Guo et al. (2025a).

| Method | DW-4 ($d = 8$) | | LJ-13 ($d = 39$) | | LJ-55 ($d = 165$) | |
|---|---|---|---|---|---|---|
| | $W_2 \downarrow$ | $E(\cdot)\ W_2 \downarrow$ | $W_2 \downarrow$ | $E(\cdot)\ W_2 \downarrow$ | $W_2 \downarrow$ | $E(\cdot)\ W_2 \downarrow$ |
| PDDS (Phillips et al., 2024) | $0.92_{\pm 0.08}$ | $0.58_{\pm 0.25}$ | $4.66_{\pm 0.87}$ | $56.01_{\pm 10.80}$ | - | - |
| SCLD (Chen et al., 2025) | $1.30_{\pm 0.64}$ | $0.40_{\pm 0.19}$ | $2.93_{\pm 0.19}$ | $27.98_{\pm 1.26}$ | - | - |
| PIS (Zhang & Chen, 2022) | $0.68_{\pm 0.28}$ | $0.65_{\pm 0.25}$ | $1.93_{\pm 0.07}$ | $18.02_{\pm 1.12}$ | $4.79_{\pm 0.45}$ | $228.70_{\pm 131.27}$ |
| DDS (Vargas et al., 2023a) | $0.92_{\pm 0.11}$ | $0.90_{\pm 0.37}$ | $1.99_{\pm 0.13}$ | $24.61_{\pm 8.99}$ | $4.60_{\pm 0.09}$ | $173.09_{\pm 18.01}$ |
| iDEM (Akhound-Sadegh et al., 2024) | $0.70_{\pm 0.06}$ | $0.55_{\pm 0.14}$ | $1.61_{\pm 0.01}$ | $30.78_{\pm 24.46}$ | $4.69_{\pm 1.52}$ | $93.53_{\pm 16.31}$ |
| AS (Havens et al., 2025) | $0.62_{\pm 0.06}$ | $0.55_{\pm 0.12}$ | $1.67_{\pm 0.01}$ | $2.40_{\pm 1.25}$ | $4.50_{\pm 0.05}$ | $58.04_{\pm 20.98}$ |
| ASBS (Liu et al., 2025) | $0.43_{\pm 0.05}$ | $\mathbf{0.20}_{\pm 0.11}$ | $1.59_{\pm 0.03}$ | $1.99_{\pm 1.01}$ | $4.00_{\pm 0.03}$ | $28.10_{\pm 8.15}$ |
| PDNS (Guo et al., 2025a) | $0.51_{\pm 0.04}$ | $0.21_{\pm 0.03}$ | $1.57_{\pm 0.01}$ | $1.01_{\pm 0.18}$ | $3.95_{\pm 0.01}$ | $21.97_{\pm 3.14}$ |
| BMS (ours) | $\mathbf{0.38}_{\pm 0.01}$ | $0.21_{\pm 0.12}$ | $\mathbf{1.54}_{\pm 0.00}$ | $\mathbf{0.53}_{\pm 0.04}$ | $\mathbf{3.80}_{\pm 0.01}$ | $\mathbf{2.36}_{\pm 0.45}$ |

et al. (2024); Woo & Ahn (2024), which used an importance-sampling estimate of the optimal drift function for variance-exploding SDEs (Song et al., 2020). Recently, a similar method for training stochastic interpolants was proposed in (Li et al., 2026) and enhanced with control variates to reduce the importance weight variance. Other methods rely on importance-weighted variants of losses that are typically used for supervised learning tasks, such as score matching (Wang et al., 2025), flow matching (Dern et al., 2025), or stochastic optimal control (Domingo-Enrich et al., 2024). To reduce importance-weight variance, prior work has proposed regularizing updates via proximal point schemes (Guo et al., 2025a), trust regions (Blessing et al., 2025a), or by defining updates along an ODE in the space of drifts (Potaptchik et al., 2025). However, all of these methods require some form of importance sampling, which can suffer from high variance (Chatterjee & Diaconis, 2018) or are challenging to compute (Vargas et al., 2024). Fixed-point methods that are based on adjoint matching (Domingo-Enrich et al., 2025) or target score matching (De Bortoli et al., 2024) do not rely on importance weights but are either restricted to certain prior distributions (Havens et al., 2025) or require multi-stage optimizations (Liu et al., 2025; He et al., 2024; Choi et al., 2025). For further discussion of

diffusion-based sampling methods, we refer to Appendix C and for a table that compactly characterizes these methods to Table 4.

**Diffusion bridges.** To enable sampling between arbitrary boundary distributions, the problem can be framed as learning a Schrödinger Bridge (SB). Shi et al. (2023) introduced *Diffusion Schrödinger Bridge Matching* (DSBM) for the data-driven setting, utilizing Iterative Markovian Fitting that alternates between projecting onto Markovian and reciprocal classes. Recently, Liu et al. (2025) extended this to energy-based sampling with the *Adjoint Schrödinger Bridge Sampler* (ASBS), which generalizes AS by relaxing the memoryless condition, allowing for arbitrary priors. However, ASBS employs an alternating optimization scheme between the control and a corrector potential, which can be empirically unstable due to the interdependence of the objectives. While our method retains the flexibility of ASBS, it only relies on a single, scalable objective. To address mode collapse in complex landscapes, Nam et al. (2025) proposed *Well-Tempered ASBS*, which augments diffusion samplers with metadynamics-inspired bias potentials along collective variables. While our work focuses on the stability and scalability of the core sampling algorithm, our method is compatible with such enhanced sampling techniques.

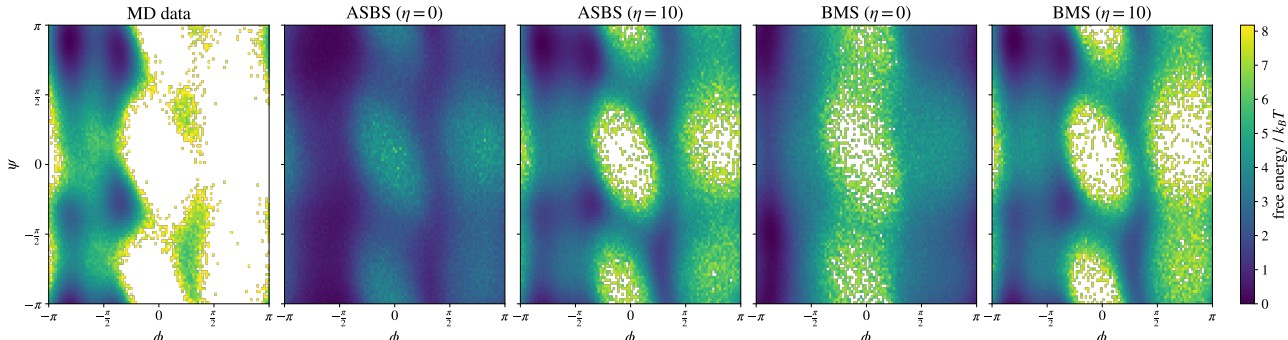

*Figure 4.* Ramachandran plots with $10^6$ samples for Alanine dipeptide for Molecular Dynamics (MD) data, ASBS (Liu et al., 2025), and our method, BMS, for damping values $\eta \in \{0, 10\}$. ASBS and BMS are trained on all-atom coordinates solely using energy evaluations.

## 4. Numerical Experiments

We evaluate the performance of our proposed BMS by comparing it to key instances of our framework: the *Adjoint sampler* (AS) (Havens et al., 2025) and the *Adjoint Schrödinger bridge sampler* (ASBS) (Liu et al., 2025). Our evaluation spans three distinct testbeds: high-dimensional synthetic Gaussian mixture models, $n$-body identical particle systems, and two molecular peptide systems. Detailed descriptions of the experimental setup, benchmark problems, and evaluation metrics are provided in Appendix F, with supplementary results available in Appendix G. The code is publicly available[4].

### 4.1. Gaussian mixture models

In this experiment, we consider Gaussian mixture targets consisting of 20 components with unit variance and means sampled uniformly from $[-20, 20]^d$. We assess the models based on their ability to capture all modes and the accuracy of their density fitting. Specifically, we utilize *mode TVD* – the total variation distance between empirical and ground-truth mixture weights – to quantify mode collapse. Additionally, we use the Wasserstein ($W_2$) distance to compare the model's empirical density with ground truth samples from the target. As shown in Figure 2, our analysis focuses on a comparison with ASBS, as the standard adjoint sampler did not produce stable or competitive results in this setting. While many contemporary studies focus on dimensions ranging from $d = 2$ to $50$, we push the complexity to $d = 2500$. We observe that while ASBS is prone to instability at low damping values $\eta$, our method remains consistently stable, although it may exhibit some mode collapse. Notably, as the dimensionality increases, ASBS suffers from a clear degradation in performance, whereas our method maintains consistent and robust results across the entire range of tested dimensions. As sampling multimodal densities in such high dimensions was previously unattainable, we conjecture that damping will be a key ingredient in the future of sampling

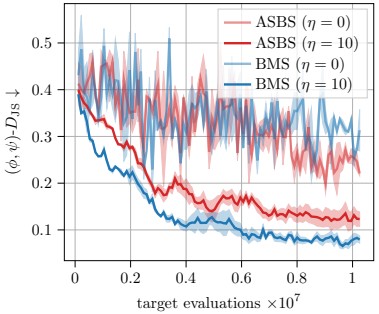

*Figure 5.* Jensen-Shannon divergence for the joint distribution of backbone dihedral angles $(\phi, \psi)$-$D_{\mathrm{JS}}$ for Alanine Dipeptide over the course of training for ASBS (Liu et al., 2025) and our proposed method, BMS, for damping values $\eta \in \{0, 10\}$.

methods. We provide further results in Appendix G, comparing the performance across different numbers of modes; see Figure 3 for a qualitative result.

### 4.2. $n$-body identical particle systems

We extend our evaluation to $n$-body systems of increasing complexity, spanning from DW-4 ($d = 8$) to the high-dimensional LJ-55 ($d = 165$). The results in Table 2 demonstrate that our approach consistently outperforms established baselines, including AS and ASBS, across both geometric Wasserstein-2 ($W_2$) and energy value distributions ($E(\cdot)\ W_2$). Critically, the most substantial benefits of our method emerge in the high-dimensional $d = 165$ regime, where it maintains superior accuracy in energy modeling while other methods begin to degrade. Furthermore, our approach exhibits significantly reduced variance across random seeds, underscoring its stability compared to existing approaches.

### 4.3. Molecular benchmarks: Peptide systems

We evaluate our method on two increasingly complex molecular benchmarks: Alanine Dipeptide (ALA2; $d = 66$) and Alanine Tetrapeptide (ALA4; $d = 126$). Notably, we train our model directly on Cartesian coordinates, a departure from existing literature which typically relies on prior do-

---

[4]https://github.com/DenisBless/
BridgeMatchingSampler

*Table 3.* Comparison of performance metrics on Alanine Dipeptide (ALA2) and Tetrapeptide (ALA4). We report the Jensen-Shannon divergence of the joint backbone dihedral angle distribution $(\phi, \psi)$ ($D_{\mathrm{JS}}$), the root-mean-squared error between the corresponding free-energy surfaces ($\mathrm{RMSE}_{\Delta F}$), and the Wasserstein-2 distance between the first two TIC components (TICA-$W_2$). Runs marked with † diverged during training; for these, results from the final checkpoint are shown. All metrics are averaged over three random seeds. The best-performing method is highlighted in bold.

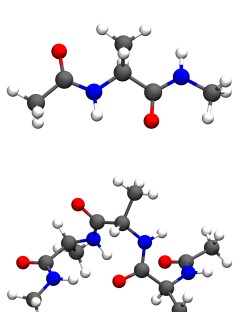

| System | Method | $\eta$ | $(\phi, \psi)$-$D_{\mathrm{JS}} \downarrow$ | $\mathrm{RMSE}_{\Delta F} \downarrow$ | TICA-$W_2 \downarrow$ |
|---|---|---|---|---|---|
| ALA2 $(d = 66)$ | ASBS (Liu et al., 2025) | 0 | $0.232_{\pm 0.018}$ | $0.204_{\pm 0.003}$ | $1.263_{\pm 0.140}$ |
| | ASBS-damped (ours) | 10 | $0.111_{\pm 0.017}$ | $0.128_{\pm 0.008}$ | $1.526_{\pm 0.150}$ |
| | BMS (ours) | 0 | $0.311_{\pm 0.050}$ | $0.742_{\pm 0.514}$ | $2.707_{\pm 0.268}$ |
| | BMS-damped (ours) | 10 | $\mathbf{0.068}_{\pm \mathbf{0.009}}$ | $\mathbf{0.079}_{\pm \mathbf{0.000}}$ | $\mathbf{0.900}_{\pm \mathbf{0.066}}$ |
| ALA4 $(d = 126)$ | ASBS† (Liu et al., 2025) | 0 | $0.489_{\pm 0.016}$ | $1.368_{\pm 0.836}$ | $474.8_{\pm 41.91}$ |
| | ASBS-damped (ours) | 50 | $0.230_{\pm 0.003}$ | $0.457_{\pm 0.073}$ | $\mathbf{1.606}_{\pm \mathbf{0.028}}$ |
| | BMS† (ours) | 0 | $0.502_{\pm 0.027}$ | $0.811_{\pm 0.158}$ | $280.3_{\pm 1.526}$ |
| | BMS-damped (ours) | 50 | $\mathbf{0.228}_{\pm \mathbf{0.011}}$ | $\mathbf{0.396}_{\pm \mathbf{0.012}}$ | $1.697_{\pm 0.023}$ |

main knowledge – such as internal coordinates and collective variables (Midgley et al., 2022; Liu et al., 2025) – to enhance exploration (Nam et al., 2025), or pretraining on molecular dynamics (MD) data with high temperatures (Akhound-Sadegh et al., 2025a). To assess the quality of the generated conformations, we analyze Ramachandran plots, which represent the joint distribution of the backbone dihedral angles $\phi$ and $\psi$. These angles characterize the local conformation of the molecular backbone and are essential for identifying stable states. We compute the Jensen-Shannon divergence of pairs of backbone dihedral angles between MD data and samples generated by the model ($D_{\mathrm{JS}}$), averaged over all residues. Similarly, we compute the root-mean-squared error between the corresponding free-energy surfaces ($\mathrm{RMSE}_{\Delta F}$). Additionally, we compute the Wasserstein-2 metric between the first two TIC components (TICA-$W_2$) to ensure the capture of global conformational dependencies. Qualitative results are presented in Figures 4 and 10 for ALA2 and Figure 11 for ALA4. Quantitative results are shown Table 3 and Figure 5.

While our method achieves state-of-the-art performance across both systems, the results underscore the critical role of the proposed damping scheme. Specifically, omitting the damping component leads to poor performance and significant training instability in both ASBS and BMS, as evidenced in Figure 5. Note that we were not able to obtain results for ALA4 without damping due to numerical instabilities. A notable characteristic across both systems is their mode-covering behavior. Unlike traditional sampling techniques that often suffer from mode-collapse – typically driven by the mode-seeking nature of the reverse KL divergence – our approach successfully maintains probability mass across all regions. We conjecture that this mass-covering property is a direct result of our fixed-point iteration; as shown in (14), the fixed point corresponds to the Markovian projection, which is inherently linked to a forward KL objective.

## 5. Conclusion

In this work, we formulate diffusion-based sampling from unnormalized target densities as a generalized fixed-point iteration grounded in Nelson's relation and a generalized target score identity (see Proposition 2.6). This perspective provides a unified theoretical framework encompassing several recently proposed matching-based algorithms, including Adjoint Sampling (Havens et al., 2025) and Adjoint Schrödinger Bridge Samplers (Liu et al., 2025). Building on this foundation, we introduce a new scalable and stable algorithm, termed the *Bridge Matching Sampler* (BMS), which addresses key limitations of existing approaches – most notably alternating objectives and restrictive choices of reference processes or priors. To further improve robustness and avoid mode collapse, we propose a damped fixed-point iteration. Empirically, BMS achieves state-of-the-art performance on challenging synthetic targets and high-dimensional molecular benchmarks, and scales successfully to multimodal distributions in high dimensions.

**Future work.** Several promising directions remain for future work. First, the performance of our sampler could be further improved by incorporating an adaptive schedule for the damping parameter $\eta$, in the spirit of trust region methods (Blessing et al., 2025a), or by integrating importance weighting strategies (see Appendix E.1 for a preliminary discussion). From a theoretical perspective, establishing convergence guarantees for the proposed fixed-point iteration, as well as conditions ensuring existence and uniqueness of its solution, remains an open problem. Finally, while our focus has been on the principled development of scalable and stable diffusion-based samplers, it would be natural to extend this framework to related settings, such as the estimation of normalizing constants (Máté et al., 2025; Guo et al., 2025b; He et al., 2025) and sampling over discrete state spaces (Holderrieth et al., 2025; Sanokowski et al., 2025a; Zhu et al., 2025; Kholkin et al., 2025; So et al., 2026; Guo et al., 2026).

## Impact Statement

This paper presents work whose goal is to advance the field of Machine Learning. There are many potential societal consequences of our work, none which we feel must be specifically highlighted here.

## Acknowledgements

The authors thank Jaemoo Choi, Christopher von Klitzing, and Henrik Schopmans for the many useful discussions. D.B. acknowledges support by funding from a Google PhD fellowship in Machine Learning and ML Foundations, the pilot program Core Informatics of the Helmholtz Association (HGF) and the state of Baden-Württemberg through bwHPC, as well as the HoreKa supercomputer funded by the Ministry of Science, Research and the Arts Baden-Württemberg and by the German Federal Ministry of Education and Research. The research of L.R. was partially funded by Deutsche Forschungsgemeinschaft (DFG) through the grant CRC 1114 "Scaling Cascades in Complex Systems" (project A05, project number 235221301).

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

# Appendix

## A. Assumptions and auxiliary results

### A.1. Additional notation

For vectors $v_1, v_2 \in \mathbb{R}^d$, we denote by $\|v\|$ the Euclidean norm and by $v_1 \cdot v_2$ the Euclidean inner product. For a real-valued matrix $A$, we denote by $\mathrm{Tr}(A)$ and $A^\top$ its trace and transpose.

For a sufficiently smooth function $f \colon \mathbb{R}^d \times [0, T] \to \mathbb{R}$, we denote by $\nabla f = \nabla_x f$ its gradient w.r.t. the spatial variables $x$ and by $\partial_t f$ and $\partial_{x_i} f$ its partial derivatives w.r.t. the time coordinate $t$ and the spatial coordinate $x_i$, respectively.

We denote by $\mathcal{N}(\mu, \Sigma)$ a multivariate normal distribution with mean $\mu \in \mathbb{R}^d$ and covariance matrix $\Sigma \in \mathbb{R}^{d \times d}$. For random variables $X_1, X_2$, we denote by $\mathbb{E}[X_1]$ and $\mathrm{Var}[X_1]$ the expectation and variance of $X_1$ and by $\mathbb{E}[X_1|X_2]$ the conditional expectation of $X_1$ given $X_2$.

### A.2. Technical assumptions

Throughout our work, we make the same assumptions as in Nüsken & Richter (2021); Domingo-Enrich et al. (2024); Shi et al. (2023), which are needed for all the objects considered to be well-defined and for the propositions to hold. Namely, we assume that:

(i) The set $\mathcal{U}$ of *admissible controls* is given by
$$\mathcal{U} = \{u \in C^1(\mathbb{R}^d \times [0, T]; \mathbb{R}^d) \,|\, \exists C > 0, \, \forall (x, s) \in \mathbb{R}^d \times [0, T], \, u(x, s) \le C(1 + \|x\|)\}. \tag{26}$$

(ii) The diffusion coefficient $\sigma(t)$ is a deterministic, continuously differentiable, and scalar-valued function of time, implying isotropic noise. To ensure the process is non-degenerate, we assume $\sigma(t) > 0$ for all $t \in [0, T]$.

(iii) For the target path measure $\Pi^* = \Pi^*_{0,T} \mathbb{P}_{|0,T} \in \mathcal{R}(\mathbb{P})$, we assume the following:

    (a) For all $x_0 \in \mathbb{R}^d$, we have $\Pi^*_{T|0} \ll \mathbb{P}_{T|0}$.

    (b) The Radon-Nikodym derivative $r_T(x_T|x_0) = \frac{\mathrm{d}\Pi^*_{T|0}}{\mathrm{d}\mathbb{P}_{T|0}}(x_T)$ is bounded for all $x_T \in \mathbb{R}^d$.

    (c) The mapping $(t, x) \mapsto \mathbb{E}_{\Pi^*}[\xi(X, t)|X_t = x]$ is locally Lipschitz and there exists $C > 0$ and $\psi \in C([0, T], \mathbb{R}_+)$ such that for any $t \in [0, T]$ and $x \in \mathbb{R}^d$, we have
$$\|\mathbb{E}_{\Pi^*_{T|t}}[\nabla \log \mathbb{P}_{T|t}(X_T|X_t)|X_t = x]\| \le C\psi(t)(1 + \|x\|). \tag{27}$$

(iv) The target boundary coupling $\Pi^*_{0,T}(x_0, x_T)$ is such that for any $t \in (0, T)$, the product $\Pi^*_{0,T}(x_0, x_T)\mathbb{P}_{t|0,T}(x|x_0, x_T)$ and its first-order spatial derivatives vanish at the limit as $\|x\| \to \infty$ or $\|x_T\| \to \infty$.

### A.3. Useful identities

Throughout this section, we consider the following stochastic processes:

(i) Reference process $(X \sim \mathbb{P})$: We fix it to a scaled Brownian motion, namely $\mathrm{d}X_t = \sigma(t)\mathrm{d}B_t$ with $X_0 \sim p_{\mathrm{prior}}$. We define the cumulative variance $\kappa(t) = \int_0^t \sigma^2(s)\mathrm{d}s$ and the relative variance $\gamma(t) = \kappa(t)/\kappa(T)$.

(ii) Controlled process $(X \sim \mathbb{P}^u)$: The Markovian diffusion defined by $\mathrm{d}X_t = \sigma(t)u(X_t, t)\mathrm{d}t + \sigma(t)\mathrm{d}B_t$.

(iii) Path-dependent process $(X \sim \Pi^*$ or $X \sim \Pi^i)$: The non-Markovian process defined by $\mathrm{d}X_t = \sigma(t)\xi(X, t)\mathrm{d}t + \sigma(t)\mathrm{d}B_t$.

**Theorem A.1** (Girsanov's theorem for path measures). *Let $u, v \in \mathcal{U}$. Then the Radon-Nikodym derivative between $\mathbb{P}^u$ and $\mathbb{P}^v$, evaluated along $X$, is given by*
$$\log \frac{\mathrm{d}\mathbb{P}^u}{\mathrm{d}\mathbb{P}^v}(X) = \int_0^T \sigma^{-1}(u - v)(X_t, t) \cdot \mathrm{d}X_t - \frac{1}{2}\int_0^T \left(\|u\|^2 - \|v\|^2\right)(X_t, t)\mathrm{d}t. \tag{28}$$

*Proof.* See, e.g., Lemma A.1 in Nüsken & Richter (2021) or Appendix E in Vargas et al. (2024). $\qquad\square$

**Proposition A.2** (Nelson's relation). *Let $\Pi^* \in \mathcal{P}$ be the target path measure with marginal densities $\Pi_t^*$. Let $u^*$ and $v^*$ be the unique optimal forward and backward controls such that the process $X$ driven by $\mathrm{d}X_t = \sigma(t)u^*(X_t, t)\mathrm{d}t + \sigma(t)\mathrm{d}B_t$ and the time-reversed process $Y$ driven by $\mathrm{d}Y_t = -\sigma(t)v^*(Y_t, t)\mathrm{d}t + \sigma(t)\mathrm{d}\bar{B}_t$ satisfy $Law(X) = Law(Y) = \Pi^*$. Then, the controls satisfy the following duality relation:*

$$u^*(x, t) + v^*(x, t) = \sigma(t)\nabla_x \log \Pi_t^*(x). \tag{29}$$

*Proof.* See e.g. Vargas et al. (2024) Proposition 2.1. □

**Lemma A.3** (Markovian projection for path measures (Brunick & Shreve, 2013)). Let $\Pi^* \in \mathcal{P}$ be a (possibly non-Markovian) path measure induced by the path-dependent SDE in (10) with drift functional $\xi(X, t)$. Then, there exists a unique Markov measure $\mathbb{P}^{u^*} \in \mathcal{M}$ induced by the Markovian SDE

$$\mathrm{d}X_t = \sigma(t)u^*(X_t, t)\mathrm{d}t + \sigma(t)\mathrm{d}B_t \tag{30}$$

with $u^*(x, t) = \mathbb{E}_{\Pi^*}[\xi(X, t)|X_t = x]$. The measure $\mathbb{P}^{u^*}$ is the Markovian projection of $\Pi^*$ in the sense that their time-marginals coincide for all $t \in [0, T]$, i.e. $\mathbb{P}_t^{u^*} = \Pi_t^*$.

*Proof.* See, e.g., Corollary 3.7 in Brunick & Shreve (2013). □

**Lemma A.4** (Fokker-Planck equation). Let $X = (X_t)_{t \in [0,T]}$ be a Markov process satisfying the SDE

$$\mathrm{d}X_t = \sigma(t)u(X_t, t)\mathrm{d}t + \sigma(t)\mathrm{d}B_t, \quad X_0 \sim \Pi_0^*, \tag{31}$$

where $u \in \mathcal{U}$ is an admissible control and $\sigma(t) \in C^1([0, T], \mathbb{R}_{>0})$ is a deterministic diffusion coefficient. Then, under Assumption (iv) the evolution of the marginal density $\Pi_t^*$ is governed by the Fokker-Planck equation

$$\partial_t \Pi_t^*(x) = -\nabla \cdot (\Pi_t^*(x)\sigma(t)u(x, t)) + \frac{1}{2}\sigma(t)^2\Delta\Pi_t^*(x). \tag{32}$$

*Proof.* See, e.g., Section 2.1 in Berner et al. (2024).

**Lemma A.5** (Orthogonal projection in $L^2$). Let $\xi$ be a path-dependent random variable such that $\mathbb{E}_{\Pi^i}[\xi^2] < \infty$, and let $\Phi(X_t) := \mathbb{E}_{\Pi^i}[\xi|X_t]$ be its Markovian projection. For any measurable Markovian function $u(X_t, t)$, the following decomposition holds:

$$\mathbb{E}_{\Pi^i}[\|\xi(X, t) - u(X_t, t)\|^2] = \mathbb{E}_{\Pi^i}[\|\xi(X, t) - \Phi(X_t)\|^2] + \mathbb{E}_{\Pi^i}[\|\Phi(X_t) - u(X_t, t)\|^2]. \tag{33}$$

*Proof.* We begin by adding and subtracting the projection $\Phi(X_t)$ inside the norm, i.e.

$$\mathbb{E}_{\Pi^i}[\|\xi(X, t) - u(X_t, t)\|^2] = \mathbb{E}_{\Pi^i}[\|(\xi(X, t) - \Phi(X_t)) + (\Phi(X_t) - u(X_t, t))\|^2]. \tag{34}$$

Expanding the square, we obtain

$$\begin{aligned} \mathbb{E}_{\Pi^i}[\|\xi(X, t) - u(X_t, t)\|^2] =& \mathbb{E}_{\Pi^i}[\|\xi(X, t) - \Phi(X_t)\|^2] + \mathbb{E}_{\Pi^i}[\|\Phi(X_t) - u(X_t, t)\|^2] \\ &+ 2\mathbb{E}_{\Pi^i}[(\xi(X, t) - \Phi(X_t)) \cdot (\Phi(X_t) - u(X_t, t))]. \end{aligned} \tag{35}$$

We now show that the cross-term vanishes using the tower property. Let $g(X_t, t) := \Phi(X_t) - u(X_t, t)$, which is a function solely of the state at time $t$:

$$\begin{aligned} \mathbb{E}_{\Pi^i}[(\xi(X, t) - \Phi(X_t), g(X_t, t))] &= \mathbb{E}_{\Pi^i}\big[\mathbb{E}_{\Pi^i}[(\xi(X, t) - \Phi(X_t)) \cdot g(X_t, t)|X_t]\big] \\ &= \mathbb{E}_{\Pi^i}\big[(\mathbb{E}_{\Pi^i}[\xi(X, t)|X_t] - \Phi(X_t)) \cdot g(X_t, t)\big]. \end{aligned} \tag{36}$$

By the definition of the projection, $\Phi(X_t) = \mathbb{E}_{\Pi^i}[\xi(X, t)|X_t]$. Therefore, the inner term is

$$\mathbb{E}_{\Pi^i}\big[(\Phi(X_t) - \Phi(X_t)) \cdot g(X_t, t)\big] = 0. \tag{37}$$

The cross-term vanishes, leaving only the sum of the squared distances, which concludes the proof. □

**Lemma A.6.** Let $\Pi^*, \mathbb{P} \in \mathcal{P}$ such that $\Pi^* \ll \mathbb{P}$. Let $X$ denote the path and $X_t$ the process value at time $t$. Let $\Pi_t^*$ and $\mathbb{P}_t$ denote the marginal distributions at time $t$. Then, $\mathbb{P}_t$-almost surely

$$\frac{\mathrm{d}\Pi_t^*}{\mathrm{d}\mathbb{P}_t}(x) = \mathbb{E}_{\mathbb{P}}\left[\frac{\mathrm{d}\Pi^*}{\mathrm{d}\mathbb{P}}(X)\bigg|X_t = x\right]. \tag{38}$$

*Proof.* Let $h : \mathbb{R}^d \to \mathbb{R}$ be an arbitrary bounded measurable test function. We compute the expectation $\mathbb{E}_{\Pi^*}[h(X_t)]$ in two distinct ways.

1. Via the marginal measure $\Pi_t^*$: Using the definition of the Radon-Nikodym derivative on the marginal space $\mathbb{R}^d$, we have

$$\mathbb{E}_{\Pi^*}[h(X_t)] = \int_{\mathbb{R}^d} h(x)\,\mathrm{d}\Pi_t^*(x) = \int_{\mathbb{R}^d} h(x)\frac{\mathrm{d}\Pi_t^*}{\mathrm{d}\mathbb{P}_t}(x)\,\mathrm{d}\mathbb{P}_t(x). \tag{39}$$

2. Via the path measure $\Pi^*$: Using the Radon-Nikodym derivative $L = \frac{\mathrm{d}\Pi^*}{\mathrm{d}\mathbb{P}}$ on the full path space and the tower property of conditional expectation, we get

$$\mathbb{E}_{\Pi^*}[h(X_t)] = \mathbb{E}_{\mathbb{P}}\left[h(X_t)\frac{\mathrm{d}\Pi^*}{\mathrm{d}\mathbb{P}}(X)\right] \tag{40a}$$

$$= \mathbb{E}_{\mathbb{P}}\left[\mathbb{E}_{\mathbb{P}}\left[h(X_t)\frac{\mathrm{d}\Pi^*}{\mathrm{d}\mathbb{P}}(X)\Big|X_t\right]\right] \tag{40b}$$

$$= \mathbb{E}_{\mathbb{P}}\left[h(X_t)\,\mathbb{E}_{\mathbb{P}}\left[\frac{\mathrm{d}\Pi^*}{\mathrm{d}\mathbb{P}}(X)\Big|X_t\right]\right]. \tag{40c}$$

Note that we pulled $h(X_t)$ out of the inner expectation because it is $\sigma(X_t)$-measurable. We can rewrite (40) as an integral over the marginal measure $\mathbb{P}_t$, i.e.

$$\mathbb{E}_{\Pi^*}[h(X_t)] = \int_{\mathbb{R}^d} h(x)\,\mathbb{E}_{\mathbb{P}}\left[\frac{\mathrm{d}\Pi^*}{\mathrm{d}\mathbb{P}}(X)\Big|X_t = x\right]\,\mathrm{d}\mathbb{P}_t(x). \tag{41}$$

Comparing (39) and (41), we have

$$\int_{\mathbb{R}^d} h(x)\frac{\mathrm{d}\Pi_t^*}{\mathrm{d}\mathbb{P}_t}(x)\,\mathrm{d}\mathbb{P}_t(x) = \int_{\mathbb{R}^d} h(x)\,\mathbb{E}_{\mathbb{P}}\left[\frac{\mathrm{d}\Pi^*}{\mathrm{d}\mathbb{P}}(X)\Big|X_t = x\right]\,\mathrm{d}\mathbb{P}_t(x). \tag{42}$$

Since this equality holds for any bounded measurable $h$, the integrands must be equal $\mathbb{P}_t$-almost everywhere. $\square$

# B. Proofs

In this section we state the proofs of the statements from the main part of the paper and add some additional statements as well.

## B.1. Proofs for general couplings

**Proposition B.1** (Denoising score identity). *For any path measure $\Pi^* \in \mathcal{R}(\mathbb{P})$ with coupling $\Pi_{0,T}^*$, the score of the marginal density $\Pi_t^*$ satisfies the Denoising Score Identity (DSI)*

$$\nabla_x \log \Pi_t^*(x) = \mathbb{E}_{\Pi_{0,T|t}^*}\left[\nabla_{X_t} \log \mathbb{P}_{t|0,T}(X_t|X_0, X_T)|X_t = x\right]. \tag{43}$$

*Proof.* We consider the marginal density $\Pi_t^*(x)$, which for a general coupling $\Pi_{0,T}^*(x_0, x_T)$, is given by

$$\Pi_t^*(x) = \int_{\mathbb{R}^{2d}} \mathbb{P}_{t|0,T}(x|x_0, x_T)\Pi_{0,T}^*(x_0, x_T)\mathrm{d}x_0\mathrm{d}x_T. \tag{44}$$

We differentiate $\Pi_t^*(x)$ with respect to $x$. With Assumption (iii)b, we can interchange differentiation and integration, such that the gradient operates directly on the transition kernel $\mathbb{P}_{t|0,T}$:

$$\nabla_x \Pi_t^*(x) = \int_{\mathbb{R}^{2d}} \nabla_x \mathbb{P}_{t|0,T}(x|x_0, x_T)\Pi_{0,T}^*(x_0, x_T)\mathrm{d}x_0\mathrm{d}x_T. \tag{45}$$

Using the log-derivative identity $\nabla_x \mathbb{P}_{t|0,T} = \mathbb{P}_{t|0,T}\nabla_x \log \mathbb{P}_{t|0,T}$, we rewrite the gradient as

$$\nabla_x \Pi_t^*(x) = \int_{\mathbb{R}^{2d}} \left(\nabla_x \log \mathbb{P}_{t|0,T}(x|x_0, x_T)\right)\mathbb{P}_{t|0,T}(x|x_0, x_T)\Pi_{0,T}^*(x_0, x_T)\mathrm{d}x_0\mathrm{d}x_T. \tag{46}$$

To recover the score $\nabla_x \log \Pi_t^*(x)$, we divide both sides by the marginal density $\Pi_t^*(x)$. By Bayes' theorem, the posterior

density of the boundaries given the current state is given as

$$\Pi_{0,T|t}^*(x_0, x_T|x) = \frac{\mathbb{P}_{t|0,T}(x|x_0, x_T)\Pi_{0,T}^*(x_0, x_T)}{\Pi_t^*(x)}. \tag{47}$$

Substituting this identity into the integral expression (46), we get:

$$\nabla_x \log \Pi_t^*(x) = \int_{\mathbb{R}^{2d}} \nabla_x \log \mathbb{P}_{t|0,T}(x|x_0, x_T)\Pi_{0,T|t}^*(x_0, x_T|x)\mathrm{d}x_0\mathrm{d}x_T \tag{48a}$$

$$= \mathbb{E}_{\Pi_{0,T|t}^*}\left[\nabla_{X_t} \log \mathbb{P}_{t|0,T}(X_t|X_0, X_T)|X_t = x\right]. \tag{48b}$$

This concludes the proof. $\square$

*Proof of Proposition 2.5.* Applying Doob's $h$-transforms (Rogers & Williams, 2000), conditioning the base process on a terminal state $X_T$ yields the (path-dependent) drift $\sigma(t)\nabla_{X_t} \log \mathbb{P}_{T|t}(X_T|X_t)$. Using the Markovian projection, we identify the optimal forward control

$$u^*(x, t) = \sigma(t)\mathbb{E}_{\Pi_{T|t}^*}[\nabla_{X_t} \log \mathbb{P}_{T|t}(X_T|X_t)|X_t = x]. \tag{49}$$

Further, by Nelson's identity, it holds that

$$u^*(x, t) + v^*(x, t) = \sigma(t)\nabla_x \log \Pi_t^*(x). \tag{50}$$

We apply Proposition B.1 for $\nabla_x \log \Pi_t^*$:

$$\nabla_x \log \Pi_t^*(x) = \mathbb{E}_{\Pi_{0,T|t}^*}\left[\nabla_{X_t} \log \mathbb{P}_{t|0,T}(X_t|X_0, X_T)\Big|X_t = x\right]. \tag{51}$$

For the Markovian reference process $\mathbb{P}$, the bridge transition density factorizes as $\mathbb{P}_{t|0,T}(x|x_0, x_T) \propto \mathbb{P}_{t|0}(x|x_0)\mathbb{P}_{T|t}(x_T|x)$. Taking the gradient of the log-density yields

$$\nabla_x \log \mathbb{P}_{t|0,T}(x|X_0, X_T) = \nabla_x \log \mathbb{P}_{t|0}(x|X_0) + \nabla_x \log \mathbb{P}_{T|t}(X_T|x), \tag{52}$$

and substituting into (51), we obtain

$$\sigma(t)\nabla_x \log \Pi_t^*(x) = \sigma(t)\mathbb{E}_{\Pi_{0|t}^*}[\nabla_{X_t} \log \mathbb{P}_{t|0}(X_t|X_0)|X_t = x] + \sigma(t)\mathbb{E}_{\Pi_{T|t}^*}[\nabla_{X_t} \log \mathbb{P}_{T|t}(X_T|X_t)|X_t = x] \tag{53a}$$

$$= \sigma(t)\mathbb{E}_{\Pi_{0|t}^*}[\nabla_{X_t} \log \mathbb{P}_{t|0}(X_t|X_0)|X_t = x] + u^*(x, t). \tag{53b}$$

By Nelson relation, we get the expression for the backward drift $v^*(x, t)$. $\square$

*Proof of Proposition 2.6.* By Proposition B.1, the marginal density $\Pi_t^*(x) = \int_{\mathbb{R}^{2d}} \Pi_{0,T}^*(x_0, x_T)\mathbb{P}_{t|0,T}(x|x_0, x_T)\mathrm{d}x_0\mathrm{d}x_T$ is given by

$$\nabla_x \log \Pi_t^*(x) = \mathbb{E}_{\Pi_{0,T|t}^*}\left[\nabla_x \log \mathbb{P}_{t|0,T}(X_t|X_0, X_T)|X_t = x\right]. \tag{54}$$

For the reference process $\mathrm{d}X_t = \sigma(t)\mathrm{d}B_t$, the transition density is Gaussian, namely

$$\mathbb{P}_{t|0,T}(X_t|X_0, X_T) = \mathcal{N}\left(X_t; (1 - \gamma(t))X_0 + \gamma(t)X_T, \ \kappa(T)\gamma(t)(1 - \gamma(t))I\right), \tag{55}$$

see also Appendix E.5. We can therefore compute

$$\nabla_{X_t} \log \mathbb{P}_{t|0,T}(X_t|X_0, X_T) = \frac{1}{1 - \gamma(t)}\nabla_{X_0} \log \mathbb{P}_{t|0,T}(X_t|X_0, X_T) = \frac{1}{\gamma(t)}\nabla_{X_T} \log \mathbb{P}_{t|0,T}(X_t|X_0, X_T). \tag{56}$$

We integrate by parts and apply Assumption (iv) to eliminate the boundary terms:

$$\int_{\mathbb{R}^{2d}} \nabla_{x_0}\mathbb{P}_{t|0,T}(x_t|x_0, x_T)\Pi_{0,T}^*(x_0, x_T)\mathrm{d}x_0\mathrm{d}x_T = -\int_{\mathbb{R}^{2d}} \mathbb{P}_{t|0,T}(x_t|x_0, x_T)\nabla_{x_0}\Pi_{0,T}^*(x_0, x_T)\mathrm{d}x_0\mathrm{d}x_T. \tag{57}$$

Using the log-derivative identity $\nabla_{x_0}\Pi_{0,T}^*(x_0, x_T) = \Pi_{0,T}^*(x_0, x_T)\nabla_{x_0} \log \Pi_{0,T}^*(x_0, x_T)$, we get

$$\mathbb{E}_{\Pi_{0,T|t}^*}\left[\nabla_{X_t} \log \mathbb{P}_{t|0,T}(X_t|X_0, X_T)|X_t = x\right] = \mathbb{E}_{\Pi_{0,T|t}^*}\left[\frac{1}{1 - \gamma(t)}\nabla_{X_0} \log \Pi_{0,T}^*(X_0, X_T)|X_t = x\right]. \tag{58}$$

By symmetry, applying this to $X_T$ yields

$$\mathbb{E}_{\Pi^*_{0,T|t}}\left[\nabla_{X_t}\log\mathbb{P}_{t|0,T}(X_t|X_0,X_T)|X_t=x\right]=\mathbb{E}_{\Pi^*_{0,T|t}}\left[\frac{1}{\gamma(t)}\nabla_{X_T}\log\Pi^*_{0,T}(X_0,X_T)|X_t=x\right]. \tag{59}$$

Since both (58) and (59) equal the same marginal score $\nabla_x\log\Pi^*_t$, any convex combination is also an unbiased estimator, which concludes the proof. $\quad\square$

*Proof of Proposition 2.7.* We start with Nelson's identity (Proposition A.2)

$$u^*(x,t)=-v^*(x,t)+\sigma(t)\nabla_x\log\Pi^*_t(x). \tag{60}$$

By Proposition 2.5 the backward drift is given by

$$v^*(x,t)=\sigma(t)\mathbb{E}_{\Pi^*_{0|t}}[\nabla_{X_t}\log\mathbb{P}_{t|0}(X_t|X_0)|X_t=x]. \tag{61}$$

Next, we substitute the marginal score $\nabla_x\log\Pi^*_t(x)$ using the generalized TSI (Proposition 2.6), namely

$$\nabla_x\log\Pi^*_t(x)=\mathbb{E}_{\Pi^*_{0,T|t}}\left[\frac{1-c(t)}{1-\gamma(t)}\nabla_{X_0}\log\Pi^*_{0,T}(X_0,X_T)+\frac{c(t)}{\gamma(t)}\nabla_{X_T}\log\Pi^*_{0,T}(X_0,X_T)\Big|X_t=x\right]. \tag{62}$$

Substituting the expressions for $v^*$ and $\nabla\log\Pi^*_t(x)$ back into Nelson's identity, we obtain

$$u^*(x,t)=-\sigma(t)\mathbb{E}_{\Pi^*}[\nabla\log\mathbb{P}_{t|0}(X_t|X_0)|X_t=x]$$
$$+\sigma(t)\mathbb{E}_{\Pi^*}\left[\frac{1-c(t)}{1-\gamma(t)}\nabla_{X_0}\log\Pi^*_{0,T}(X_0,X_T)+\frac{c(t)}{\gamma(t)}\nabla_{X_T}\log\Pi^*_{0,T}(X_0,X_T)|X_t=x\right] \tag{63a}$$

$$=\sigma(t)\mathbb{E}_{\Pi^*}\left[-\nabla\log\mathbb{P}_{t|0}(X_t|X_0)+\frac{1-c(t)}{1-\gamma(t)}\nabla_{X_0}\log\Pi^*_{0,T}(X_0,X_T)+\frac{c(t)}{\gamma(t)}\nabla_{X_T}\log\Pi^*_{0,T}(X_0,X_T)|X_t=x\right]. \tag{63b}$$

Identifying the term inside the expectation as $\xi(X,t)$, we arrive at the result. $\quad\square$

## B.2. Proofs for Schrödinger bridges

We consider the Schrödinger Bridge (SB) problem with reference measure $\mathbb{P}$ induced by the SDE $\mathrm{d}X_t=\sigma(t)\mathrm{d}B_t$. This problem is defined as the entropy-regularized optimal transport problem

$$\min_{\mathbb{P}^u\in\mathcal{P}}D_{\mathrm{KL}}(\mathbb{P}^u|\mathbb{P})\quad\text{subject to}\quad\mathbb{P}^u_0=\Pi^*_0,\ \mathbb{P}^u_T=\Pi^*_T. \tag{64}$$

This is equivalent to the stochastic optimal control problem

$$\min_{u\in\mathcal{U}}\mathbb{E}\left[\int_0^T\tfrac{1}{2}\|u(X_t,t)\|^2\,\mathrm{d}t\right]\quad\text{s.t.}\quad\begin{cases}X_0\sim\Pi^*_0=p_{\mathrm{prior}},\\ X_T\sim\Pi^*_T=p_{\mathrm{target}},\end{cases} \tag{65}$$

under the controlled dynamics

$$\mathrm{d}X_t=\sigma(t)u(X_t,t)\mathrm{d}t+\sigma(t)\mathrm{d}B_t. \tag{66}$$

The following derivations of the Schrödinger system and its associated optimal controls are standrad in the literature, and we provide them here primarily for the reader's convenience.

Using Girsanov's theorem and the Fokker-Planck equation, the evolution of the marginal density $\Pi^*_t(x)$ can be enforced via the Lagrange multiplier $\psi(x,t)$:

$$\mathcal{L}(\Pi^*,u,\psi)=\int_0^T\int_{\mathbb{R}^d}\left(\tfrac{1}{2}\|u(x,t)\|^2\Pi^*_t(x)+\psi(x,t)\left(\partial_t\Pi^*_t(x)+\sigma(t)\nabla\cdot(\Pi^*_t(x)u(x,t))-\tfrac{1}{2}\sigma^2(t)\,\mathrm{Tr}(\nabla^2\Pi^*_t(x))\right)\right)\mathrm{d}x\,\mathrm{d}t. \tag{67}$$

**Lemma B.2.** The optimal marginal density $\Pi^*_t$ and the optimal control $u^*$ satisfy:

$$\begin{cases}\partial_t\psi+\tfrac{1}{2}\sigma^2\,\mathrm{Tr}(\nabla^2\psi)+\tfrac{1}{2}\sigma^2\|\nabla\psi\|^2=0,\\ \partial_t\Pi^*_t+\sigma\nabla\cdot(\Pi^*_t u^*)=\tfrac{1}{2}\sigma^2\,\mathrm{Tr}(\nabla^2\Pi^*_t),\end{cases} \tag{68}$$

where the optimal control is given by $u^*=\sigma\nabla\psi_t$.

*Proof.* Using integration by parts (where boundary terms vanish by Assumption (iv)), we can rewrite the term containing the divergence inside the Lagrangian:

$$\int_{\mathbb{R}^d} \psi(x,t)\sigma(t)\nabla \cdot (\Pi_t^*(x)u(x,t))\mathrm{d}x = -\int_{\mathbb{R}^d} \sigma(t)\Pi_t^*(x)u(x,t) \cdot \nabla\psi(x,t)\mathrm{d}x. \tag{69}$$

The terms in the Lagrangian involving the control $u$ are

$$\int_0^T \int_{\mathbb{R}^d} \Pi_t^*(x) \left(\tfrac{1}{2}\|u(x,t)\|^2 - \sigma(t)u(x,t) \cdot \nabla\psi(x,t)\right) \mathrm{d}x\mathrm{d}t. \tag{70}$$

Taking the functional derivative with respect to $u$ and setting it to zero, we get

$$\frac{\delta\mathcal{L}}{\delta u} = \Pi_t^*(u - \sigma\nabla\psi) = 0. \tag{71}$$

Hence, we obtain the optimal control

$$u^* = \sigma\nabla\psi. \tag{72}$$

Substituting $u^*$ into the Lagrangian yields

$$\mathcal{L} = \int_0^T \int_{\mathbb{R}^d} (\tfrac{1}{2}\sigma^2(t)\|\nabla\psi(x,t)\|^2\Pi_t^*(x) + \psi(x,t)\partial_t\Pi_t^*(x)$$
$$+ \sigma^2(t)\psi(x,t)\nabla \cdot (\Pi_t^*(x)\nabla\psi(x,t)) - \tfrac{1}{2}\sigma^2(t)\psi(x,t)\,\mathrm{Tr}(\nabla^2\Pi_t^*(x)))\mathrm{d}x\,\mathrm{d}t. \tag{73}$$

We isolate $\Pi_t^*$ in every term. For that, we use integration by parts on the last three terms:

$$\int_0^T \psi(x,t)\partial_t\Pi_t^*(x)\mathrm{d}t = -\int_0^T \Pi_t^*(x)\partial_t\psi(x,t)\mathrm{d}t, \tag{74}$$

$$\int_{\mathbb{R}^d} \sigma^2(t)\psi(x,t)\nabla \cdot (\Pi_t^*(x)\nabla\psi(x,t))\mathrm{d}x = -\int_{\mathbb{R}^d} \sigma^2(t)\Pi_t^*(x)(\nabla\psi(x,t)) \cdot \nabla\psi(x,t)\mathrm{d}x \tag{75}$$

$$= -\int_{\mathbb{R}^d} \sigma^2(t)\Pi_t^*(x)\|\nabla\psi(x,t)\|^2\mathrm{d}x, \tag{76}$$

$$\int_{\mathbb{R}^d} \sigma^2(t)\psi(x,t)\,\mathrm{Tr}(\nabla^2\Pi_t^*(x))\mathrm{d}x = \int_{\mathbb{R}^d} \sigma^2(t)\Pi_t^*(x)\,\mathrm{Tr}(\nabla^2\psi(x,t))\mathrm{d}x, \tag{77}$$

where the boundary terms vanish by Assumption (iv). Substituting these back into the Lagrangian gives

$$\mathcal{L} = \int_0^T \int_{\mathbb{R}^d} \Pi_t^*(x) \left(\tfrac{1}{2}\sigma^2(t)\|\nabla\psi(x,t)\|^2 - \partial_t\psi(x,t) - \sigma^2(t)\|\nabla\psi(x,t)\|^2 - \tfrac{1}{2}\sigma^2(t)\,\mathrm{Tr}(\nabla^2\psi(x,t))\right) \mathrm{d}x\mathrm{d}t \tag{78a}$$

$$= \int_0^T \int_{\mathbb{R}^d} \Pi_t^*(x) \left(-\partial_t\psi(x,t) - \tfrac{1}{2}\sigma^2(t)\|\nabla\psi(x,t)\|^2 - \tfrac{1}{2}\sigma^2(t)\,\mathrm{Tr}(\nabla^2\psi(x,t))\right) \mathrm{d}x\mathrm{d}t. \tag{78b}$$

Taking the functional derivative with respect to $\Pi_t^*$ and setting it to zero gives the first PDE in (68) (namely the Hamilton-Jacobi-Bellman equation). If we substitute the optimal control $u^* = \sigma\nabla\psi$ into the Fokker-Planck equation of the general controlled process, we obtain the second PDE:

$$\partial_t\Pi_t^* + \sigma\nabla \cdot (\Pi_t^*u^*) = \tfrac{1}{2}\sigma^2\,\mathrm{Tr}(\nabla^2\Pi_t^*). \tag{79}$$

$\square$

**Lemma B.3.** The nonlinear system of PDEs in (68) can be linearized via the Hopf-Cole transformation to satisfy the linear system

$$\begin{cases} \partial_t\varphi = -\tfrac{1}{2}\sigma^2\Delta\varphi, \\ \partial_t\widehat{\varphi} = \tfrac{1}{2}\sigma^2\Delta\widehat{\varphi}, \end{cases} \tag{80}$$

subject to the coupled boundary conditions

$$\begin{cases} \varphi_0\widehat{\varphi}_0 = \Pi_0^*, \\ \varphi_T\widehat{\varphi}_T = \Pi_T^*. \end{cases} \tag{81}$$

*Proof.* Define the mapping $(\Pi_t^*, \psi) \mapsto (\varphi, \widehat{\varphi})$ with

$$\varphi_t = \exp(\psi_t), \tag{82}$$

$$\widehat{\varphi}_t = \Pi_t^* \exp(-\psi_t) = \frac{\Pi_t^*}{\varphi_t}. \tag{83}$$

From $\psi = \log \varphi$, we calculate the derivatives

$$\nabla \psi = \frac{\nabla \varphi}{\varphi}, \quad \Delta \psi = \frac{\Delta \varphi}{\varphi} - \frac{\|\nabla \varphi\|^2}{\varphi^2}, \quad \partial_t \psi = \frac{\partial_t \varphi}{\varphi}. \tag{84}$$

Substituting these into the second PDE from (68) yields

$$\partial_t \psi + \tfrac{1}{2}\sigma^2 \operatorname{Tr}(\nabla^2 \psi) + \tfrac{1}{2}\sigma^2 \|\nabla \psi\|^2 = 0 \tag{85a}$$

$$\Leftrightarrow \quad \frac{\partial_t \varphi}{\varphi} + \tfrac{1}{2}\sigma^2 \left( \frac{\Delta \varphi}{\varphi} - \frac{\|\nabla \varphi\|^2}{\varphi^2} \right) + \tfrac{1}{2}\sigma^2 \frac{\|\nabla \varphi\|^2}{\varphi^2} = 0. \tag{85b}$$

Multiplying by $\varphi$, we obtain

$$\partial_t \varphi + \tfrac{1}{2}\sigma^2 \Delta \varphi = 0, \tag{86}$$

which is the first PDE in (81). For $\widehat{\varphi}$, we use the product relation $\Pi_t^* = \varphi \widehat{\varphi}$ and substitute it into the Fokker-Planck equation of the optimal density:

$$\partial_t(\varphi\widehat{\varphi}) + \sigma^2 \nabla \cdot (\varphi\widehat{\varphi}\nabla \log \varphi) = \tfrac{1}{2}\sigma^2 \Delta(\varphi\widehat{\varphi}). \tag{87}$$

Applying the product rule and using $u^* = \sigma^2 \frac{\nabla \varphi}{\varphi}$, we get

$$\widehat{\varphi}\partial_t \varphi + \varphi \partial_t \widehat{\varphi} + \sigma^2 \nabla \cdot (\widehat{\varphi}\nabla \varphi) = \tfrac{1}{2}\sigma^2 (\widehat{\varphi}\Delta \varphi + 2\nabla \varphi \cdot \nabla \widehat{\varphi} + \varphi\Delta \widehat{\varphi}). \tag{88}$$

Expanding the divergence term $\nabla \cdot (\widehat{\varphi}\nabla \varphi) = \nabla \widehat{\varphi} \cdot \nabla \varphi + \widehat{\varphi}\Delta \varphi$ and substituting $\partial_t \varphi = -\tfrac{1}{2}\sigma^2 \Delta \varphi$, the terms involving $\Delta \varphi$ and $\nabla \varphi \cdot \nabla \widehat{\varphi}$ cancel out, leaving

$$\varphi \partial_t \widehat{\varphi} = \tfrac{1}{2}\sigma^2 \varphi \Delta \widehat{\varphi}. \tag{89}$$

Dividing by $\varphi$ yields the second PDE. The boundary constraints follow since $\Pi_t^* = \varphi \hat{\varphi}$. $\qquad \square$

*Proof of Proposition 2.9.* By Lemma B.2, the optimal control is $u^*(x, t) = \sigma(t)\nabla \psi_t(x)$. Using the Hopf-Cole transformation $\psi = \log \varphi$, taking the gradient yields $u^* = \sigma \nabla \log \varphi$ and by symmetry, $v^* = \sigma \nabla \log \widehat{\varphi}$. We further have the following equations for $\varphi$ and $\widehat{\varphi}$:

$$\partial_t \varphi + \tfrac{1}{2}\sigma \Delta \varphi = 0, \tag{90}$$

$$\partial_t \widehat{\varphi} - \tfrac{1}{2}\sigma \Delta \widehat{\varphi} = 0. \tag{91}$$

By the Feynman-Kac formula, the solution to the PDE can be represented as the conditional expectation of the terminal value,

$$\varphi_t(x) = \mathbb{E}_{\mathbb{P}}[\varphi_T(X_T)|X_t = x] = \int_{\mathbb{R}^d} \varphi_T(x_T)\mathbb{P}_{T|t}(x_T|x)\mathrm{d}x_T, \tag{92}$$

and similarly for the forward equation,

$$\widehat{\varphi}_t(x) = \int_{\mathbb{R}^d} \mathbb{P}_{t|0}(x|x_0)\widehat{\varphi}_0(x_0)\mathrm{d}x_0. \tag{93}$$

It remains to show the factorization of the optimal coupling $\Pi_{0,T}^*$. Conditioned on the initial state $X_0$, the optimal conditional path measure $\Pi_{|0}^*$ is related to the reference path measure $\mathbb{P}_{|0}$ via the Radon-Nikodym derivative (Léonard, 2013):

$$\frac{\mathrm{d}\Pi_{|0}^*}{\mathrm{d}\mathbb{P}_{|0}}(X) = \frac{\varphi_T(X_T)}{\varphi_0(X_0)}. \tag{94}$$

By the Markov property of the reference process, the conditional transition $\Pi_{T|0}^*$ is the reference transition $\mathbb{P}_{T|0}$ weighted

by the potential ratio:

$$\Pi^*_{T|0}(x_T|x_0) = \mathbb{P}_{T|0}(x_T|x_0)\frac{\varphi_T(x_T)}{\varphi_0(x_0)}. \tag{95}$$

The joint distribution $\Pi^*_{0,T}$ is the product of this transition density and the initial marginal $\Pi^*_0$. Using the boundary condition $\Pi^*_0 = \varphi_0\widehat{\varphi}_0$ from the Schrödinger system, we can compute

$$\Pi^*_{0,T}(x_0, x_T) = \Pi^*_0(x_0)\Pi^*_{T|0}(x_T|x_0) \tag{96a}$$

$$= \varphi_0(x_0)\widehat{\varphi}_0(x_0)\mathbb{P}_{T|0}(x_T|x_0)\frac{\varphi_T(x_T)}{\varphi_0(x_0)} \tag{96b}$$

$$= \widehat{\varphi}_0(x_0)\mathbb{P}_{T|0}(x_T|x_0)\varphi_T(x_T), \tag{96c}$$

which concludes the statement. $\qquad\square$

**Proposition B.4.** *Consider the Schrödinger Bridge problem associated with the reference process $\mathrm{d}X_t = \sigma(t)\mathrm{d}B_t$. Let $\Pi^* \in \mathcal{R}(\mathbb{P})$ be the unique bridge measure with marginals $\Pi^*_0$ and $\Pi^*_T$ The optimal forward drift $u^*$ and backward drift $v^*$ are characterized by the Schrödinger potentials $(\varphi, \widehat{\varphi})$ and satisfy the identities*

$$u^*(x,t) = \sigma(t)\mathbb{E}_{\Pi^*_{T|t}}\left[\nabla_{X_T}\log\frac{\Pi^*_T(X_T)}{\widehat{\varphi}_T(X_T)}\Big| X_t = x\right], \tag{97}$$

$$v^*(x,t) = \sigma(t)\mathbb{E}_{\Pi^*_{0|t}}\left[\nabla_{X_0}\log\frac{\Pi^*_0(X_0)}{\varphi_0(X_0)}\Big| X_t = x\right]. \tag{98}$$

*Proof.* For the optimal forward control $u^*$ holds that

$$u^*(x,t) = \sigma(t)\nabla_x\log\varphi_t(x) = \sigma(t)\frac{\nabla_x\varphi_t(x)}{\varphi_t(x)}. \tag{99}$$

The potential $\varphi_t(x)$ evolves backward in time according to (20), yielding

$$\varphi_t(x) = \int_{\mathbb{R}^d}\mathbb{P}_{T|t}(x_T|x)\varphi_T(x_T)\mathrm{d}x_T. \tag{100}$$

For the reference process $\mathrm{d}X_t = \sigma(t)\mathrm{d}B_t$, the transition density $\mathbb{P}_{T|t}(x_T|x)$ depends only on the difference $(x_T - x)$. Explicitly, we have $\mathbb{P}_{T|t}(x_T|x) = \mathcal{N}(x_T; x, \Sigma_{T|t})$. This implies the identity

$$\nabla_x\mathbb{P}_{T|t}(x_T|x) = -\nabla_{x_T}\mathbb{P}_{T|t}(x_T|x). \tag{101}$$

We differentiate the integral form of $\varphi$ with respect to $x$:

$$\nabla_x\varphi_t(x) = \int_{\mathbb{R}^d}\nabla_x\mathbb{P}_{T|t}(x_T|x)\varphi_T(x_T)\mathrm{d}x_T \tag{102a}$$

$$= \int_{\mathbb{R}^d}\left(-\nabla_{x_T}\mathbb{P}_{T|t}(x_T|x)\right)\varphi_T(x_T)\mathrm{d}x_T \tag{102b}$$

$$= \int_{\mathbb{R}^d}\mathbb{P}_{T|t}(x_T|x)\nabla_{x_T}\varphi_T(x_T)\mathrm{d}x_T. \tag{102c}$$

Use the log-derivative trick $\nabla\varphi = \varphi\nabla\log\varphi$, we get

$$\nabla_x\varphi_t(x) = \int_{\mathbb{R}^d}\mathbb{P}_{T|t}(x_T|x)\varphi_T(x_T)\nabla_{x_T}\log\varphi_T(x_T)\mathrm{d}x_T. \tag{103}$$

We substitute this back into the expression for $u^*$ and divide by $\varphi_t(x)$. We identify the bridge density $\Pi^*_{T|t}$ using the reciprocal relation for Schrödinger Bridges:

$$\Pi^*_{T|t}(x_T|x) = \frac{\mathbb{P}_{T|t}(x_T|x)\varphi_T(x_T)}{\varphi_t(x)}. \tag{104}$$

Thus we get

$$u^*(x,t) = \sigma(t) \int_{\mathbb{R}^d} \underbrace{\frac{\mathbb{P}_{T|t}(x_T|x)\varphi_T(x_T)}{\varphi_t(x)}}_{=\Pi^*_{T|t}(x_T|x)} \nabla_{x_T} \log \varphi_T(x_T) \mathrm{d}x_T \tag{105a}$$

$$= \sigma(t)\mathbb{E}_{\Pi^*_{T|t}}\left[\nabla_{X_T} \log \varphi_T(X_T)|X_t = x\right]. \tag{105b}$$

At time $T$, the Schrödinger system boundary condition is $\Pi^*_T(x) = \varphi_T(x)\widehat{\varphi}_T(x)$. Solving for $\varphi_T(x)$, we get

$$\varphi_T(x) = \frac{\Pi^*_T(x)}{\widehat{\varphi}_T(x)}. \tag{106}$$

Substituting this into the expectation yields the result

$$u^*(x,t) = \sigma(t)\mathbb{E}_{\Pi^*_{T|t}}\left[\nabla \log \frac{\Pi_T(X_T)}{\widehat{\varphi}_T(X_T)}\Big|X_t = x\right]. \tag{107}$$

The proof for $v^*$ is analogous but proceeds forward in time. $\qquad\square$

**Proposition B.5** (Path-dependent drift for Schrödinger bridges). *Under the assumptions of Proposition 2.6, the path-dependent drift $\xi$ corresponding to the non-Markovian SDE (10) that induces the target measure $\Pi^* = \Pi^{\mathrm{SB}}_{0,T}\mathbb{P}_{|0,T}$ is given by*

$$\sigma(t)^{-1}\xi(X,t) = \frac{\gamma(t) - c(t)}{\gamma(t)(1 - \gamma(t))}\nabla_{X_0} \log \mathbb{P}_{T|0}(X_T|X_0) + \frac{\gamma(t) - c(t)}{1 - \gamma(t)}\nabla_{X_0} \log \frac{\Pi^*_0(X_0)}{\varphi_0(X_0)} + \frac{c(t)}{\gamma(t)}\nabla_{X_T} \log \frac{\Pi^*_T(X_T)}{\widehat{\varphi}_T(X_T)}.$$

*Proof.* Let us start from the general identity for $\xi$, stated in Proposition 2.7,

$$\sigma^{-1}(t)\xi(X,t) = -\nabla_{X_t} \log \mathbb{P}_{t|0}(X_t|X_0) + \frac{1 - c(t)}{1 - \gamma(t)}\nabla_{X_0} \log \Pi^*_{0,T}(X_0,X_T) + \frac{c(t)}{\gamma(t)}\nabla_{X_T} \log \Pi^*_{0,T}(X_0,X_T). \tag{108}$$

As stated in Proposition 2.9, for a Schrödinger Bridge, the optimal coupling factorizes as

$$\Pi^*_{0,T}(X_0,X_T) = \mathbb{P}_{T|0}(X_T|X_0)\widehat{\varphi}_0(X_0)\varphi_T(X_T). \tag{109}$$

Hence, we can compute

$$\nabla_{X_0} \log \Pi^*_{0,T}(X_0,X_T) = \nabla_{X_0} \log \mathbb{P}_{T|0}(X_T|X_0) + \nabla_{X_0} \log \widehat{\varphi}_0(X_0), \tag{110}$$

$$\nabla_{X_T} \log \Pi^*_{0,T}(X_0,X_T) = \nabla_{X_T} \log \mathbb{P}_{T|0}(X_T|X_0) + \nabla_{X_T} \log \varphi_T(X_T). \tag{111}$$

Plugging this into (108) yields

$$\sigma^{-1}(t)\xi(X,t) = -\nabla_{X_t} \log \mathbb{P}_{t|0}(X_t|X_0) + \frac{1 - c(t)}{1 - \gamma(t)}\nabla_{X_0} \log \mathbb{P}_{T|0}(X_T|X_0) + \frac{c(t)}{\gamma(t)}\nabla_{X_T} \log \mathbb{P}_{T|0}(X_T|X_0)$$
$$+ \frac{1 - c(t)}{1 - \gamma(t)}\nabla_{X_0} \log \widehat{\varphi}_0(X_0) + \frac{c(t)}{\gamma(t)}\nabla_{X_T} \log \varphi_T(X_T). \tag{112}$$

For the reference process $\mathrm{d}X_t = \sigma(t)\mathrm{d}B_t$, the transition density is $\mathbb{P}_{T|0}(X_T|X_0) = \mathcal{N}(X_T; X_0, \kappa(T)I)$. Hence,

$$\nabla_{X_T} \log \mathbb{P}_{T|0}(X_T|X_0) = -\nabla_{X_0} \log \mathbb{P}_{T|0}(X_T|X_0), \tag{113}$$

and therefore

$$\frac{1 - c(t)}{1 - \gamma(t)}\nabla_{X_0} \log \mathbb{P}_{T|0}(X_T|X_0) + \frac{c(t)}{\gamma(t)}\nabla_{X_T} \log \mathbb{P}_{T|0}(X_T|X_0) = \frac{\gamma(t) - c(t)}{\gamma(t)(1 - \gamma(t))}\nabla_{X_0} \log \mathbb{P}_{T|0}(X_T|X_0). \tag{114}$$

Further, using Proposition 2.5, we have

$$v^*(x,t) = \sigma(t)\mathbb{E}_{\Pi^*_{0|t}}[\nabla_{X_t} \log \mathbb{P}_{t|0}(X_t|X_0)|X_t = x], \tag{115}$$

and from Proposition B.4

$$v^*(x,t) = \sigma(t)\mathbb{E}_{\Pi^*_{0|t}}\left[\nabla_{X_0} \log \frac{\Pi^*_0(X_0)}{\varphi_0(X_0)}\Big|X_t = x\right] = \sigma(t)\mathbb{E}_{\Pi^*_{0|t}}\left[\nabla_{X_0} \log \widehat{\varphi}_0(X_0)\Big|X_t = x\right]. \tag{116}$$

We can therefore compute

$$\mathbb{E}_{\Pi^*_{0|t}}\Big[-\nabla_{X_t}\log\mathbb{P}_{t|0}(X_t|X_0)+\frac{1-c(t)}{1-\gamma(t)}\nabla_{X_0}\log\widehat{\varphi}_0(X_0)|X_t=x\Big]=\mathbb{E}_{\Pi^*_{0|t}}\Big[\frac{\gamma(t)-c(t)}{1-\gamma(t)}\nabla_{X_0}\log\widehat{\varphi}_0(X_0)|X_t=x\Big].$$

(117)

Finally, combining (112), (114) and (117), and noting again the Markovian projection formula from Lemma 2.3, we obtain

$$\sigma^{-1}(t)\xi(X,t)=\frac{\gamma(t)-c(t)}{\gamma(t)(1-\gamma(t))}\nabla_{X_0}\log\mathbb{P}_{T|0}(X_T|X_0)+\frac{\gamma(t)-c(t)}{1-\gamma(t)}\nabla_{X_0}\log\widehat{\varphi}_0(X_0)+\frac{c(t)}{\gamma(t)}\nabla_{X_T}\log\varphi_T(X_T).$$

(118)

$\square$

## B.3. Proofs for Schrödinger half bridges

Under the memoryless condition, a Schrödinger Half Bridge can be formulated as a special case of a Schrödinger Bridge. Recall that we have the following minimization problem for SHBs:

$$\min_{u\in\mathcal{U}}\mathbb{E}_{\mathbb{P}^u}\left[\int_0^T\tfrac{1}{2}\|u(X_t,t)\|^2\mathrm{d}t\right]\quad\text{s.t.}\quad X_T\sim\Pi^*_T=p_{\text{target}}.$$

(119)

This optimization yields the optimal joint coupling $\Pi^*_{0,T}=\Pi^*_T\mathbb{P}_{0|T}$. By the memoryless condition, the backward transition of the reference process becomes independent of the terminal state and thus $\mathbb{P}_{0|T}=\mathbb{P}_0$. Consequently, we also have $\Pi^*_0=\mathbb{P}_0$.

Since both the initial and terminal marginals are determined, we can identify the corresponding potentials for the Schrödinger system $\Pi^*_t=\varphi_t\hat{\varphi}_t$.

- **Initial potentials:** To satisfiy $\Pi^*_0=\varphi_0\hat{\varphi}_0=\mathbb{P}_0$, we have the initial potentials as $\hat{\varphi}_0=\mathbb{P}_0$ and $\varphi_0=1$.

- **Terminal potentials:** Under the reference dynamics, the forward potential evolves exactly as the reference marginal, yielding $\hat{\varphi}_T=\mathbb{P}_T$. To satisfy the terminal constraint $\Pi^*_T=\varphi_T\hat{\varphi}_T$, the terminal backward potential must therefore be $\varphi_T=\Pi^*_T/\mathbb{P}_T$.

**Lemma B.6** (Path-dependent drift for Schrödinger Half Bridges). Let $\mathrm{d}X_t=\sigma(t)\mathrm{d}B_t$ be the reference process with path measure $\mathbb{P}$. Consider the target path measure $\Pi^*\in\mathcal{R}(\mathbb{P})$ defined by the Schrödinger Half Bridge coupling $\Pi^*_{0,T}=\Pi^*_T(X_T)\mathbb{P}_{0|T}(X_0|X_T)$. The unique optimal control $u^*$ is given by

$$u^*(x,t)=\sigma(t)\mathbb{E}_{\Pi^*}\left[\xi(X,t)|X_t=x\right],$$

(120)

where

$$\sigma(t)^{-1}\xi(X,t)=\frac{c(t)-\gamma(t)}{\gamma(t)}\nabla_{X_0}\log\mathbb{P}_0(X_0)+\frac{c(t)}{\gamma(t)}\nabla_{X_T}\log\frac{\Pi^*_T(X_T)}{\mathbb{P}_T(X_T)}+\frac{\gamma(t)-c(t)}{\gamma(t)(1-\gamma(t))}\nabla_{X_0}\log\mathbb{P}_{0|T}(X_0|X_T)$$

(121)

for any $c=c(t)\in[0,1]$.

*Proof.* For a general Schrödinger Bridge, the path-dependent drift $\xi$ is given by Proposition B.5:

$$\sigma(t)^{-1}\xi(X,t)=\frac{\gamma(t)-c(t)}{\gamma(t)(1-\gamma(t))}\nabla_{X_0}\log\mathbb{P}_{T|0}(X_T|X_0)+\frac{\gamma(t)-c(t)}{1-\gamma(t)}\nabla_{X_0}\log\frac{\Pi^*_0(X_0)}{\varphi_0(X_0)}+\frac{c(t)}{\gamma(t)}\nabla_{X_T}\log\frac{\Pi^*_T(X_T)}{\widehat{\varphi}_T(X_T)}.$$

(122)

Substituting the known couplings for Schrödinger Half Bridges yields

$$\sigma(t)^{-1}\xi(X,t)=\frac{\gamma(t)-c(t)}{\gamma(t)(1-\gamma(t))}\nabla_{X_0}\log\mathbb{P}_{T|0}(X_T|X_0)+\frac{\gamma(t)-c(t)}{1-\gamma(t)}\nabla_{X_0}\log\mathbb{P}_0(X_0)+\frac{c(t)}{\gamma(t)}\nabla_{X_T}\log\frac{\Pi^*_T(X_T)}{\mathbb{P}_T(X_T)}.$$

(123)

We use Bayes' theorem to expand the score of the forward transition: $\nabla_{X_0}\log\mathbb{P}_{T|0}(X_T|X_0)=\nabla_{X_0}\log\mathbb{P}_{0|T}(X_0|X_T)-$

$\nabla_{X_0} \log \mathbb{P}_0(X_0)$. After substituting and grouping the terms of $\nabla_{X_0} \log \mathbb{P}_0(X_0)$ together, we have

$$\sigma(t)^{-1}\xi(X,t) = \frac{c(t)-\gamma(t)}{\gamma(t)}\nabla_{X_0}\log\mathbb{P}_0(X_0) + \frac{c(t)}{\gamma(t)}\nabla_{X_T}\log\frac{\Pi_T^*(X_T)}{\mathbb{P}_T(X_T)} + \frac{\gamma(t)-c(t)}{\gamma(t)(1-\gamma(t))}\nabla_{X_0}\log\mathbb{P}_{0|T}(X_0|X_T). \tag{124}$$

$\square$

*Proof of Proposition 2.8.* This directly follows from Lemma B.6 setting $c(t) = \gamma(t)\ \forall t \in [0,T]$. $\square$

### B.4. Proofs for independent couplings

*Proof of Proposition 2.11.* This follows directly from Proposition 2.7, which for the general case states

$$\xi(X,t) = -\sigma(t)\left(\nabla_x\log\mathbb{P}_{t|0}(X_t|X_0) + \frac{1-c(t)}{1-\gamma(t)}\nabla_{X_0}\log\Pi_{0,T}^*(X_0,X_T) + \frac{c(t)}{\gamma(t)}\nabla_{X_T}\log\Pi_{0,T}^*(X_0,X_T)\right). \tag{125}$$

For independent couplings, we have $\Pi_{0,T}^*(x_0,x_T) = \Pi_0^*(x_0) \otimes \Pi_T^*(x_T)$ and can calculate

$$\nabla_{X_0}\log\Pi_{0,T}^*(X_0,X_T) = \nabla_{X_0}\log\Pi_0^*(X_0), \tag{126}$$

$$\nabla_{X_T}\log\Pi_{0,T}^*(X_0,X_T) = \nabla_{X_T}\log\Pi_T^*(X_T). \tag{127}$$

Substituting these gradients back into the expression for $\xi$ concludes the proof. $\square$

### B.5. Proofs for damped fixed-point iterates

*Proof of Proposition 2.13.* The operator $\Phi(u_i)$ is defined as the Markovian projection of the drift $\xi$ under the measure $\Pi^i$:

$$\Phi(u_i)(x,t) = \mathbb{E}_{\Pi^i}[\xi(X,t)|X_t = x]. \tag{128}$$

Let $\mathcal{J}$ be the objective functional

$$\mathcal{J}(u) = \mathbb{E}_{\Pi^i}\left[\int_0^T \left(\frac{1}{2}\|\xi(X,t) - u(X_t,t)\|^2 + \frac{\eta}{2}\|u_i(X_t,t) - u(X_t,t)\|^2\right)\mathrm{d}t\right]. \tag{129}$$

Using the bias-variance decomposition (see Lemma A.5), the first term of the objective can be decomposed as:

$$\mathbb{E}_{\Pi^i}[\|\xi(X,t) - u(X_t,t)\|^2|X_t = x] = \mathbb{E}_{\Pi^i}[\|\xi(X,t) - \Phi(u_i)(X_t,t)\|^2|X_t = x] + \|\Phi(u_i)(x,t) - u(x,t)\|^2. \tag{130}$$

Since the first term does not depend on $u$, minimizing the original objective functional (129) is equivalent to minimizing

$$\int_0^T \left(\mathbb{E}_{\Pi^i}\left[\frac{1}{2}\|\Phi(u_i)(X_t,t) - u(X_t,t)\|^2 + \frac{\eta}{2}\|u_i(X_t,t) - u(X_t,t)\|^2\right]\right)\mathrm{d}t. \tag{131}$$

We minimize the integrand pointwise for almost every $(x,t)$ by taking the variational derivative with respect to $u$ and set it to zero, i.e.

$$\delta_u\left(\frac{1}{2}\|\Phi(u_i) - u\|^2 + \frac{\eta}{2}\|u_i - u\|^2\right) = -(\Phi(u_i) - u) - \eta(u_i - u) \overset{!}{=} 0. \tag{132}$$

Rearranging terms to isolate $u$ gives

$$u = \frac{1}{1+\eta}\Phi(u_i) + \frac{\eta}{1+\eta}u_i. \tag{133}$$

By substituting the definition of the damping parameter $\eta = \frac{1-\alpha}{\alpha}$, we find that $\frac{1}{1+\eta} = \alpha$ and $\frac{\eta}{1+\eta} = 1 - \alpha$. Thus, the minimizer exactly recovers the damped fixed-point update rule:

$$u_{i+1}(x,t) = \alpha\,\Phi(u_i)(x,t) + (1-\alpha)u_i(x,t). \tag{134}$$

$\square$

**B.6. Proofs for control variates**

*Proof of Proposition E.1.* We seek to minimize the conditional variance term from the decomposition with respect to $c(t)$. Let us isolate the integrand:

$$J(c) = \mathbb{E}_{\Pi_t^*}\left[\mathrm{Var}_{\Pi_{|t}^*}\left(\sigma^{-1}(t)\xi^c(X,t|X_t)\right)\right] = \mathbb{E}_{\Pi^*}\left[\|\sigma^{-1}(t)\xi^c(X,t) - \mathbb{E}_{\Pi^*}[\sigma^{-1}(t)\xi^c(X,t)|X_t]\|^2\right]. \tag{135}$$

Substituting our definition $\sigma^{-1}(t)\xi^c = G_0(X) - G_v(X) - c(t)(G_0(X) - G_T(X))$ into the expected conditional variance yields

$$J(c) = \mathbb{E}_{\Pi^*}\left[\left\|\left(G_0(X) - G_v(X) - c(t)(G_0(X) - G_T(X))\right) - \mathbb{E}_{\Pi^*}[G_0(X) - G_v(X) - c(t)(G_0(X) - G_T(X))|X_t]\right\|^2\right]. \tag{136}$$

Crucially, because $G_0$ and $G_T$ are both unbiased estimators of the marginal score, their conditional expectations are equal. Therefore, the conditional expectation of their difference evaluates to zero:

$$\mathbb{E}_{\Pi^*}[G_0(X) - G_T(X)|X_t] = \mathbb{E}_{\Pi^*}[G_0(X)|X_t] - \mathbb{E}_{\Pi^*}[G_T(X)|X_t] = 0. \tag{137}$$

This simplifies our objective to

$$J(c) = \mathbb{E}_{\Pi^*}\left[\left\|(G_0(X) - G_v(X) - \mathbb{E}_{\Pi^*}[G_0(X) - G_v(X)|X_t]) - c(t)(G_0(X) - G_T(X))\right\|^2\right]. \tag{138}$$

Expanding the squared norm using the inner product, and applying our definitions for scalar variance and covariance, we obtain a quadratic function in $c(t)$, namely

$$J(c) = \mathbb{E}_{\Pi^*}\left[\|G_0(X) - G_v(X) - \mathbb{E}_{\Pi^*}[G_0(X) - G_v(X)|X_t]\|^2\right]$$
$$- 2c(t)\mathbb{E}_{\Pi^*}\left[(G_0(X) - G_v(X) - \mathbb{E}_{\Pi^*}[G_0(X) - G_v(X)|X_t]) \cdot (G_0(X) - G_T(X))\right] + c(t)^2\,\mathrm{Var}(G_0 - G_T). \tag{139}$$

By the law of total expectation, and recalling that $\mathbb{E}_{\Pi^*}[G_0(X) - G_T(X)|X_t] = 0$, the cross-term simplifies to exactly the covariance between $(G_0 - G_v)$ and $(G_0 - G_T)$:

$$\mathbb{E}_{\Pi^*}\left[(G_0(X) - G_v(X) - \mathbb{E}_{\Pi^*}[G_0(X) - G_v(X)|X_t]) \cdot (G_0(X) - G_T(X))\right] \tag{140a}$$
$$= \mathrm{Cov}(G_0 - G_v, G_0 - G_T) - \mathbb{E}_{\Pi^*}\left[(\mathbb{E}_{\Pi^*}[G_0(X) - G_v(X)|X_t]) \cdot (\mathbb{E}_{\Pi^*}[G_0(X) - G_T(X)|X_t])\right] \tag{140b}$$
$$= \mathrm{Cov}(G_0 - G_v, G_0 - G_T). \tag{140c}$$

To find the minimum, we differentiate $J(c)$ with respect to the scalar $c(t)$ and set the derivative to zero:

$$\frac{\partial J(c)}{\partial c(t)} = -2\,\mathrm{Cov}(G_0 - G_v, G_0 - G_T) + 2c(t)\,\mathrm{Var}(G_0 - G_T) = 0. \tag{141}$$

Solving for $c(t)$ yields $c^*(t) = \frac{\mathrm{Cov}(G_0 - G_v, G_0 - G_T)}{\mathrm{Var}(G_0 - G_T)}$. Finally, using the bilinearity of covariance to expand the numerator and denominator produces the full expression in the proposition. $\square$

*Proof of Proposition E.2.* The left term on the right-hand side of the bias-variance decomposition in (194) is independent of $c$ due to $\nabla_c \mathbb{E}_{\Pi^*}[\xi^c(X,t)|X_t] = 0$. Minimizing the total objective $\mathbb{E}_{\Pi^*}\left[\int_0^T \frac{1}{2}\|u(X_t,t) - \xi^c(X,t)\|^2 dt\right]$ with respect to $c$ is therefore equivalent to minimizing the variance penalty $\mathbb{E}_{\Pi^*}\left[\int_0^T \frac{1}{2}\|\xi^c(X,t) - \mathbb{E}[\xi^c(X,t)|X_t]\|^2 dt\right]$. Using the equality in (195) concludes the proof. $\square$

**B.7. Proofs for the alternative characterization of the non-Markovian drift $\xi$**

Before proceeding with the proof of Proposition E.3, we provide two auxiliary results.

**Proposition B.7** (Transition score decompositions). *Let $\mathbb{P}$ be the reference process induced by scaled Brownian motion $dX_t = \sigma(t)dB_t$. Define $\kappa(t) := \int_0^t \sigma^2(s)ds$ and $\gamma(t) := \frac{\kappa(t)}{\kappa(T)}$. The scores of the forward transition densities can be*

*decomposed in terms of the bridge score and a boundary term as follows:*

$$\nabla_{X_t} \log \mathbb{P}_{T|t}(X_T|X_t) = \frac{X_T - X_0}{\kappa(T)} + \gamma(t)\nabla_{X_t} \log \mathbb{P}_{t|0,T}(X_t|X_0, X_T), \tag{142}$$

$$\nabla_{X_t} \log \mathbb{P}_{t|0}(X_t|X_0) = (1 - \gamma(t))\nabla_{X_t} \log \mathbb{P}_{t|0,T}(X_t|X_0, X_T) - \frac{X_T - X_0}{\kappa(T)}. \tag{143}$$

*Proof of Proposition B.7.* For the reference process $X_t = \int_0^t \sigma(s)\mathrm{d}B_s$, the forward and bridge transition densities are Gaussian. Their respective variances are given by $\mathrm{Var}(X_T|X_t) = \kappa(T) - \kappa(t) = \kappa(T)(1 - \gamma(t))$, $\mathrm{Var}(X_t|X_0) = \kappa(t) = \kappa(T)\gamma(t)$, and $\mathrm{Var}(X_t|X_0, X_T) = \kappa(T)\gamma(t)(1 - \gamma(t))$.

The score of the bridge density with respect to $X_t$ is therefore

$$\nabla_{X_t} \log \mathbb{P}_{t|0,T}(X_t|X_0, X_T) = \frac{-X_t + (1 - \gamma(t))X_0 + \gamma(t)X_T}{\kappa(T)\gamma(t)(1 - \gamma(t))}. \tag{144}$$

To prove (142), we compute the score of the forward density $\mathbb{P}_{T|t}(X_T|X_t)$ as

$$\nabla_{X_t} \log \mathbb{P}_{T|t}(X_T|X_t) = \frac{X_T - X_t}{\kappa(T)(1 - \gamma(t))}. \tag{145}$$

By adding and subtracting $X_0$ in the numerator, we obtain

$$\frac{X_T - X_t}{\kappa(T)(1 - \gamma(t))} = \frac{(X_T - X_0)(1 - \gamma(t)) + (X_T - X_0)\gamma(t) + X_0 - X_t}{\kappa(T)(1 - \gamma(t))} \tag{146a}$$

$$= \frac{X_T - X_0}{\kappa(T)} + \gamma(t)\left(\frac{-X_t + (1 - \gamma(t))X_0 + \gamma(t)X_T}{\kappa(T)\gamma(t)(1 - \gamma(t))}\right). \tag{146b}$$

Substituting (144) into the second term yields (142).

To prove (143), we compute the score of the forward density $\mathbb{P}_{t|0}(X_t|X_0)$ as

$$\nabla_{X_t} \log \mathbb{P}_{t|0}(X_t|X_0) = \frac{X_0 - X_t}{\kappa(T)\gamma(t)}. \tag{147}$$

Similarly, by adding and subtracting $\gamma(t)X_T$ in the numerator, we find

$$\frac{X_0 - X_t}{\kappa(T)\gamma(t)} = \frac{-X_t + (1 - \gamma(t))X_0 + \gamma(t)X_0 + \gamma(t)X_T - \gamma(t)X_T}{\kappa(T)\gamma(t)} \tag{148a}$$

$$= \frac{-X_t + (1 - \gamma(t))X_0 + \gamma(t)X_T}{\kappa(T)\gamma(t)} - \frac{\gamma(t)(X_T - X_0)}{\kappa(T)\gamma(t)} \tag{148b}$$

$$= (1 - \gamma(t))\left(\frac{-X_t + (1 - \gamma(t))X_0 + \gamma(t)X_T}{\kappa(T)\gamma(t)(1 - \gamma(t))}\right) - \frac{X_T - X_0}{\kappa(T)}. \tag{148c}$$

Substituting (144) into the first term yields (143). □

**Proposition B.8** (Optimal forward backward drifts)**.** *Let $\mathbb{P}$ be the reference process induced by scaled Brownian motion $\mathrm{d}X_t = \sigma(t)\mathrm{d}B_t$. Define $\kappa(t) \coloneqq \int_0^t \sigma^2(s)\mathrm{d}s$ and $\gamma(t) \coloneqq \frac{\kappa(t)}{\kappa(T)}$. The optimal forward and backward drifts $u^*, v^*$ satisfy*

$$u^*(x,t) = \sigma(t)\gamma(t)\nabla_x \log \Pi_t^*(x) - \mathbb{E}_{\Pi_{0,T}^*}\left[\nabla_{X_T} \log \mathbb{P}_{T|0}(X_T|X_0)|X_t = x\right], \tag{149}$$

$$v^*(x,t) = \sigma(t)(1 - \gamma(t))\nabla_x \log \Pi_t^*(x) - \mathbb{E}_{\Pi_{0,T}^*}\left[\nabla_{X_0} \log \mathbb{P}_{T|0}(X_T|X_0)|X_t = x\right], \tag{150}$$

*and therefore $u^* + v^* = \sigma\nabla \log \Pi^*$ since $\nabla_{X_T} \log \mathbb{P}_{T|0} = -\nabla_{X_0} \log \mathbb{P}_{T|0}$.*

*Proof of Proposition B.8.* The proof directly follows from Proposition B.7 when using that $\nabla_x \log \Pi_t^*(x) = \mathbb{E}_{\Pi_{0,T|t}^*}[\nabla_{X_t} \log \mathbb{P}_{t|0,T}(X_t|X_0, X_T)|X_t = x]$ and $\frac{X_T - X_0}{\kappa(T)} = -\nabla_{X_T} \log \mathbb{P}_{T|0}(X_T|X_0) = \nabla_{X_0} \log \mathbb{P}_{T|0}(X_T|X_0)$. □

*Proof of Proposition E.3.* The statement directly follows by combining the generalized target score identity and (149). □

### B.8. Proofs for stochastic interpolants

*Proof of Proposition D.1.* Note that

$$\rho_t(x_t) = \int_{\mathbb{R}^{2d}} \rho_{0,t,T}(x_0, x_t, x_T) \mathrm{d}x_0 \mathrm{d}x_T = \int_{\mathbb{R}^{2d}} \rho_{t|0,T}(x_t|x_0, x_T)\nu(x_0, x_T) \mathrm{d}x_0 \mathrm{d}x_T. \tag{151}$$

Using that $x_t = I_t$ we have

$$\rho_{t|0,T}(x_t|x_0, x_T) = \mathcal{N}\left(x_t|\alpha(t)x_0 + \beta(t)x_T, \gamma(t)^2 I\right), \tag{152}$$

and therefore

$$\nabla_{x_t}\rho_{t|0,T}(x_t|x_0, x_T) = -\tfrac{1}{\alpha(t)}\nabla_{x_0}\rho_{t|0,T}(x_t|x_0, x_T) = -\tfrac{1}{\beta(t)}\nabla_{x_T}\rho_{t|0,T}(x_t|x_0, x_T). \tag{153}$$

It follows that

$$\nabla_{x_t}\log\rho_t(x_t) = \frac{1}{\rho_t(x_t)}\int_{\mathbb{R}^{2d}}\nabla_{x_t}\rho_{t|0,T}(x_t|x_0, x_T)\nu(x_0, x_T)\mathrm{d}x_0\mathrm{d}x_T \tag{154a}$$

$$= \frac{1}{\rho_t(x_t)}\int_{\mathbb{R}^{2d}}\left(-\tfrac{1}{\alpha(t)}\nabla_{x_0}\rho_{t|0,T}(x_t|x_0, x_T)\right)\nu(x_0, x_T)\mathrm{d}x_0\mathrm{d}x_T. \tag{154b}$$

Applying integration by parts with respect to $x_0$, and assuming that the boundary terms vanish, we obtain

$$\nabla_{x_t}\log\rho_t(x_t) = \frac{1}{\rho_t(x_t)}\int_{\mathbb{R}^{2d}}\tfrac{1}{\alpha(t)}\rho_{t|0,T}(x_t|x_0, x_T)\nabla_{x_0}\nu(x_0, x_T)\mathrm{d}x_0\mathrm{d}x_T \tag{155a}$$

$$= \frac{1}{\rho_t(x_t)}\int_{\mathbb{R}^{2d}}\tfrac{1}{\alpha(t)}\rho_{t|0,T}(x_t|x_0, x_T)\nu(x_0, x_T)\nabla_{x_0}\log\nu(x_0, x_T)\mathrm{d}x_0\mathrm{d}x_T, \tag{155b}$$

where we used the log-derivative identity $\nabla_{x_0}\nu(x_0, x_T) = \nu(x_0, x_T)\nabla_{x_0}\log\nu(x_0, x_T)$.

By Bayes' theorem, the posterior of the boundaries given the interpolant is $\rho_{0,T|t}(x_0, x_T|x_t) = \frac{\rho_{t|0,T}(x_t|x_0,x_T)\nu(x_0,x_T)}{\rho_t(x_t)}$. Substituting this into the integral yields the conditional expectation:

$$\nabla_{x_t}\log\rho_t(x_t) = \mathbb{E}\left[\tfrac{1}{\alpha(t)}\nabla_{X_0}\log\nu(X_0, X_T)\Big|I_t = x_t\right]. \tag{156}$$

By a symmetric argument, utilizing $-\tfrac{1}{\beta(t)}\nabla_{x_T}\rho_{t|0,T}(x_t|x_0, x_T)$ instead and integrating by parts with respect to $x_T$, we also have

$$\nabla_{x_t}\log\rho_t(x_t) = \mathbb{E}\left[\tfrac{1}{\beta(t)}\nabla_{X_T}\log\nu(X_0, X_T)\Big|I_t = x_t\right]. \tag{157}$$

Since both expressions evaluate to the exact same marginal score, any convex combination scaled by an arbitrary scalar function $c(t)$ is also an unbiased estimator. Multiplying the first expectation by $1 - c(t)$, multiplying the second by $c(t)$, and summing them together concludes the proof:

$$\nabla_{x_t}\log\rho_t(x_t) = \mathbb{E}\left[\tfrac{1-c(t)}{\alpha(t)}\nabla_{X_0}\log\nu(X_0, X_T) + \tfrac{c(t)}{\beta(t)}\nabla_{X_T}\log\nu(X_0, X_T)\Big|I_t = x_t\right]. \tag{158}$$

$$\square$$

## C. Further related works

**Diffusion-based sampling.** Sampling from unnormalized densities has traditionally relied on Markov chain Monte Carlo (MCMC) methods. Early learnable diffusion-based approaches utilized Schrödinger-Föllmer diffusions (Zhang & Chen, 2022; Richter, 2021; Vargas et al., 2023b) or adaptations of denoising diffusion models to the energy-based setting (Berner et al., 2024; Vargas et al., 2023a; Doucet et al., 2022; Huang et al., 2023). From the perspective of stochastic optimal control, many of these approaches can be viewed as finding controls that minimize divergences between forward and backward processes (Richter & Berner, 2024; Vargas et al., 2024; Blessing et al., 2025b). A series of recent works improved the objectives, training schemes, and exploration of diffusion-based samplers (Zhang et al., 2024; He et al., 2024; OuYang et al., 2024; Erives et al., 2025; Wang et al., 2025; Yoon et al., 2025; Sanokowski et al., 2025b; Gritsaev et al., 2025; Shi et al., 2024; Berner et al., 2025; Kim et al., 2024; 2025; Chen et al., 2025; Zhang et al., 2025a; Blessing et al., 2025c; Zhang et al., 2025b; Sendera et al., 2024; Sun et al., 2024; Gruhlke et al., 2026). However, while effective, these methods typically require to keep SDE simulations in memory for the gradient computation, which is computationally expensive.

To address this, Havens et al. (2025) introduced *adjoint sampling* (AS), a framework that leverages the adjoint method to

*Table 4.* Characterization of diffusion-based sampling methods, based on their ability to use arbitrary prior distributions $p_{\text{prior}}$, whether they require importance weights or multiple stages during optimization.

| Method | Flexible Prior | No Importance Weights | No Multi-stage Optimization |
|---|---|---|---|
| iEFM (Woo & Ahn, 2024) | ✓ | ✗ | ✓ |
| BNEM (OuYang et al., 2024) | ✗ | ✗ | ✓ |
| DiKL (He et al., 2024) | ✗ | ✓ | ✗ |
| iDEM (Akhound-Sadegh et al., 2024) | ✗ | ✗ | ✓ |
| AS (Havens et al., 2025) | ✗ | ✓ | ✓ |
| IWSM (Wang et al., 2025) | ✗ | ✗ | ✓ |
| EWFM (Dern et al., 2025) | ✓ | ✗ | ✓ |
| ASBS (Liu et al., 2025) | ✓ | ✓ | ✗ |
| NAAS (Choi et al., 2025) | ✗ | ✓ | ✗ |
| RFM (Li et al., 2026) | ✓ | ✗ | ✓ |
| BMS (Ours) | ✓ | ✓ | ✓ |

enable memory-efficient gradient estimation. This allowed them to scale diffusion-based sampling to amortized conformer generation across many molecular system. However, AS relies on a "memoryless" condition, restricting the reference process and prior distributions to specific forms (e.g., Dirac priors). This condition necessitates large diffusion coefficients to achieve good performance, which can lead to instability.

**Boltzmann generators.** Boltzmann generators (BGs) (Noé et al., 2019) refer to machine learned models that approximate the Boltzmann density of a physical system, i.e.

$$p_{\text{target}}(x) \propto \exp\left(-\frac{E(x)}{k_B \tau}\right), \tag{159}$$

where $k_B$ is the Boltzmann constant and $\tau$ the temperature of the system. While BGs have been considered for various physical systems, such as condensed matter systems (Schebek et al., 2025; Hoffmann et al., 2026), our discussion focuses on molecular BGs.

Learning molecular Boltzmann generators purely from energy evaluations has been explored with some form of prior knowledge. For instance, many methods rely on internal or mixed coordinate representation (Stimper et al., 2022; Midgley et al., 2022; Schopmans & Friederich, 2025b; von Klitzing et al., 2025; Guo et al., 2025a; Choi et al., 2025), or enhance sampling techniques using e.g. knowledge of collective variables (Nam et al., 2025; Xie et al., 2026). While using such prior knowledge simplifies the problem, it typically prevents from learning a sampler that amortizes sampling across different systems, often referred to as transferable BGs (Klein & Noé, 2024; Tan et al., 2025), which is one of the most promising directions of neural samplers to gain an advantage over classical Monte Carlo methods. In contrast, training on Cartesian coordinates allows for amortization, by e.g., conditioning the model on the molecular graph (Klein & Noé, 2024). Other methods rely on molecular dynamics (MD) samples from a higher temperature (Dibak et al., 2022; Wahl et al., 2025; Schopmans & Friederich, 2025b; Rissanen et al., 2025; Akhound-Sadegh et al., 2025b) or use are used in combination with classical Monte Carlo methods (Grenioux & Noble, 2026). To the best of our knowledge, our method is the first (concurrently to (Havens et al., 2026)) to achieve reasonable results on learning molecular BGs on Cartesian coordinates purely from energy.

# D. Connections to existing literature

## D.1. Connections to proximal point methods

In this section, we establish a formal connection between our proposed damping scheme and proximal point algorithms (Parikh et al., 2014; Chen et al., 2022) defined on path space.

**Proximal point methods on path space.** The proximal point algorithm is a standard method in convex optimization for finding the minimum of a proper, lower semi-continuous convex function $\ell : \mathcal{H} \to \mathbb{R} \cup \{+\infty\}$ on a Hilbert space $\mathcal{H}$ (Parikh et al., 2014). A standard proximal update takes the form:

$$x_{i+1} = \text{prox}_{\tau\ell}(x_i) := \underset{x \in \mathcal{H}}{\arg\min}\left\{\ell(x) + \frac{1}{2\tau}\|x - x_i\|^2\right\}, \tag{160}$$

where $\tau > 0$ is a step-size parameter and the term $\frac{1}{2\tau}\|x - x_i\|^2$ acts as a penalty that restricts the update from deviating too far from the previous iterate $x_i$.

When lifting this concept to the space of probability measures (specifically, measures on path space $\mathcal{P}$), the Euclidean distance is typically replaced by a statistical divergence, such as the Kullback-Leibler (KL) divergence. For a functional $\mathcal{L} : \mathcal{P} \to \mathbb{R}$, a proximal point iteration on path space can be formulated as:

$$\mathbb{P}_{i+1} = \arg\min_{\mathbb{P} \in \mathcal{P}} \left\{\mathcal{L}(\mathbb{P}) + \eta D(\mathbb{P}, \mathbb{P}_i)\right\}, \tag{161}$$

where $\eta$ is the inverse step size (or regularization strength). This formulation is central to many recent advances in variational inference and sampling (Hertrich & Gruhlke, 2024; Guo et al., 2025a), where the goal is to iteratively transport a reference measure towards a target measure while controlling the displacement cost.

One can also formulate the proximal point scheme (161) as an optimization over the set of admissible drifts, $\mathcal{U}$, thereby implicitly restricting the optimization over Markov measures $\mathbb{P}^u \in \mathcal{M}$:

$$u_{i+1} = \arg\min_{u \in \mathcal{U}} \left\{\mathcal{L}(u) + \eta D(\mathbb{P}^u, \mathbb{P}^{u_i})\right\}, \tag{162}$$

where $\mathcal{L} : \mathcal{U} \to \mathbb{R}$ with slight abuse of notation. Note that Guo et al. (2025a) recently used an instance of the update scheme (162) with $\mathcal{L}(\mathbb{P}^u) = D_{\mathrm{KL}}(\mathbb{P}^u \| \mathbb{P}^*)$ and $D(\mathbb{P}^u, \mathbb{P}^{u_i}) = D_{\mathrm{KL}}(\mathbb{P}^u \| \mathbb{P}^{u_i})$ where $\mathbb{P}^* \in \mathcal{P}$ is the target measure.

**Proximal point fixed-point diffusion matching.** For the fixed point iteration, the objective functional is given by

$$\mathcal{L}(u) \coloneqq \mathbb{E}_{\Pi^i} \left[\int_0^T \tfrac{1}{2}\|\xi(X, t) - u(X_t, t)\|^2 \mathrm{d}t\right]. \tag{163}$$

While in principle we can choose any divergence as a regularization term, it is convenient to consider $D_{\mathrm{KL}}(\Pi^i | \mathbb{P}^u)$ as the expectation is also taken with respect to $\Pi^i$, giving the proximal update

$$u_{i+1} = \arg\min_{u \in \mathcal{U}} \mathcal{L}(u) + \eta D_{\mathrm{KL}}(\Pi^i | \mathbb{P}^u), \tag{164}$$

with $\Pi^i = \mathrm{proj}_{\mathcal{R}(\mathbb{P})}(\mathbb{P}^{u_i})$. Using Proposition 2.5 and replacing $\Pi^*$ with $\Pi^i$ we have

$$\mathrm{proj}_{\mathcal{M}}(\Pi^i) \coloneqq \arg\min_{u \in \mathcal{U}} D_{\mathrm{KL}}(\Pi^i | \mathbb{P}^u) = \mathbb{E}_{\Pi^i_{T|t}} \left[\sigma(t)\nabla_{X_t} \log \mathbb{P}_{T|t}(X_T | X_t)\Big| X_t = x\right]. \tag{165}$$

Setting the first variation of $\mathcal{L}(u) + \eta D_{\mathrm{KL}}(\Pi^i | \mathbb{P}^u)$ with respect to $u$ to zero and solving for $u$ gives the first order optimality condition

$$u_{i+1} = \tfrac{\eta}{1+\eta}\mathrm{proj}_{\mathcal{M}}(\Pi^i) + \tfrac{1}{1+\eta}\Phi(u_i). \tag{166}$$

Note that in contrast to the damped fixed-point iteration, the numerical optimization of (164) does not require storing the parameters of the previous drift $u_i$. We leave a numerical comparison for future work.

### D.2. Connections to stochastic interpolants

We note that our Bridge Matching Sampler can be interpreted as a data-free version of celebrated stochastic interpolant methods (Albergo et al., 2025), in which no samples from the target measure are available and only an unnormalized target density is given. In the classical data-based setting, one introduces an interpolant $I$ connecting samples from the prior with samples from the target distribution and exploits the identity $b(x, t) = \mathbb{E}\left[\dot{I}_t | X_t = x\right]$ for the drift of the associated dynamical system, where the dot refers to the time-derivative. This representation can be directly compared with (13) and may likewise be interpreted as a form of *Markovianization*. A key distinction is that, in the data-based setting, one can sample directly from the target path measure (9), which leads to a more straightforward training procedure and avoids the need for fixed-point iterations. In contrast, in the data-free setting considered here, one must derive a suitable regression formula that incorporates information about the target distribution solely through its density.

In the following, we provide a brief overview of stochastic interpolants. To facilitate readability for readers familiar with the stochastic interpolant literature, we adopt notation closely aligned with that of Albergo et al. (2025). For technical details and precise assumptions, we also refer the reader to Albergo et al. (2025).

**Stochastic interpolants.** Define the stochastic interpolant

$$I_t = \alpha(t)X_0 + \beta(t)X_T + \gamma(t)Z, \quad (X_0, X_T) \sim \nu, \quad Z \sim \mathcal{N}(0, I), \tag{167}$$

where $\alpha_0 = \beta_T = 1$, $\alpha_T = \beta_0 = 0$, and $\gamma_0 = \gamma_T = 0$ ensure the boundary conditions $I_0 = X_0$, and $I_T = X_T$. Here $\nu$ denotes the coupling with

$$p_{\text{prior}}(x_0) = \int_{\mathbb{R}^d} \nu(x_0, x_T) \, dx_T, \quad \text{and} \quad p_{\text{target}}(x_T) = \int_{\mathbb{R}^d} \nu(x_0, x_T) \, dx_0. \tag{168}$$

Further, we assume that $(X_0, X_T)$ and $Z$ are independent. The time evolution of the interpolant marginals can be described by the ordinary differential equation (ODE)

$$\frac{dX_t}{dt} = b(X_t, t) \tag{169}$$

where $b$ satisfies the continuity equation

$$\partial_t \rho + \nabla \cdot (b\rho) = 0. \tag{170}$$

Alternatively, the marginals can be represented by the forward and backward SDEs:

$$dX_t = \underbrace{\left(b(X_t, t) + \sigma_t^2 \nabla \log \rho(X_t, t)\right)}_{b_{\text{F}}(X_t, t)} dt + \sqrt{2}\sigma_t dB_t \tag{171}$$

$$dX_t = \underbrace{\left(b(X_t, t) - \sigma_t^2 \nabla \log \rho(X_t, t)\right)}_{b_{\text{B}}(X_t, t)} dt + \sqrt{2}\sigma_t \overleftarrow{d}B_t \tag{172}$$

which correspond to the Fokker-Planck equations:

$$\partial_t \rho + \nabla \cdot (b_{\text{F}}\rho) = \sigma_t^2 \Delta \rho, \quad \rho(0) = p_{\text{prior}} \tag{173}$$

$$\partial_t \rho + \nabla \cdot (b_{\text{B}}\rho) = -\sigma_t^2 \Delta \rho, \quad \rho(T) = p_{\text{target}}. \tag{174}$$

By applying conditional expectations, the vector field and the score are given by:

$$b(x, t) = \mathbb{E}[\dot{I}_t | I_t = x] \quad \text{and} \quad s(x, t) := \nabla_x \log \rho(x, t) = -\frac{1}{\gamma(t)} \mathbb{E}[Z | I_t = x], \tag{175}$$

which motivates the standard least-squares matching objectives:

$$b = \arg\min_{\hat{b}} \mathbb{E}\left[\|\hat{b}(I_t, t) - \dot{I}_t\|^2\right] \quad \text{and} \quad s = \arg\min_{\hat{s}} \mathbb{E}\left[\|\hat{s}(I_t, t) + \frac{1}{\gamma(t)} Z\|^2\right] \tag{176}$$

where the expectation is taken independently over $(X_0, X_T) \sim \nu$ and $Z \sim \mathcal{N}(0, I)$ with

$$\dot{I}_t = \dot{\alpha}(t)X_0 + \dot{\beta}(t)X_T + \dot{\gamma}(t)Z, \tag{177}$$

and the dot refers to the time-derivative.

**Stochastic interpolants via generalized target score identity.** To elucidate the connection between the stochastic interpolant and our framework, we start by proposing a generalized target score identity that is adapted to the SI setting and whose proof can be found in Appendix B.8.

**Proposition D.1** (Generalized target score identity for stochastic interpolants)**.** *Consider the stochastic interpolant defined in* (167)*, with* $(X_0, X_T) \sim \nu$ *and* $Z \sim \mathcal{N}(0, I)$ *independent. Let* $\rho(\cdot, t)$ *denote the density of* $I_t$ *and let* $c : [0, T] \to \mathbb{R}$ *be an arbitrary scalar function. Then,*

$$\nabla \log \rho(x, t) = \mathbb{E}\left[\frac{1 - c(t)}{\alpha(t)} \nabla_{X_0} \log \nu(X_0, X_T) + \frac{c(t)}{\beta(t)} \nabla_{X_T} \log \nu(X_0, X_T) \Big| I_t = x\right]. \tag{178}$$

Proposition D.1 generalizes the standard score representation of stochastic interpolants by expressing the marginal score as an arbitrary convex combination of the prior and target log-density contributions. The the scalar $c(t)$ acts as a control variate that can be tuned to reduce the variance of the score estimator. Plugging (178) into the standard least-squares score-matching objective yields the regression problem

$$s = \arg\min_{\hat{s}} \mathbb{E}\left[\|\hat{s}(I_t, t) - \frac{1 - c(t)}{\alpha(t)} \nabla_{X_0} \log \nu(X_0, X_T) - \frac{c(t)}{\beta(t)} \nabla_{X_T} \log \nu(X_0, X_T)\|^2\right]. \tag{179}$$

Similarly, plugging the generalized identity into the forward drift representation $b_{\text{F}} = b + \sigma(t)^2 \nabla \log \rho$ leads to the following matching objective for the forward drift

$$b_{\text{F}} = \arg\min_{\hat{b}_{\text{F}}} \mathbb{E}\left[\|\hat{b}_{\text{F}}(I_t, t) - \dot{I}_t - \sigma(t)^2 \left(\frac{1 - c(t)}{\alpha(t)} \nabla_{X_0} \log \nu(X_0, X_T) - \frac{c(t)}{\beta(t)} \nabla_{X_T} \log \nu(X_0, X_T)\right)\|^2\right]. \tag{180}$$

**Fixed-point diffusion matching for stochastic interpolants.** The objectives above optimize the score and forward drift under the true (unknown) coupling $\nu$. Following the same fixed-point philosophy as in our main framework, we replace $\nu$ by the coupling $\nu^i$ induced by the current model and iteratively refine it. At iteration $i$, we denote by $\nu^i$ the joint distribution of $(X_0, X_T)$ obtained from simulating the current drift, and we solve the regression problem

$$b_{\mathrm{F}}^{i+1} = \underset{\hat{b}_{\mathrm{F}}}{\arg\min}\, \mathbb{E}_{\nu^i}\left[\|\hat{b}_{\mathrm{F}}(I_t, t) - \dot{I}_t - \sigma(t)^2 \left(\tfrac{1-c(t)}{\alpha(t)}\nabla_{X_0}\log\nu(X_0, X_T) - \tfrac{c(t)}{\beta(t)}\nabla_{X_T}\log\nu(X_0, X_T)\right)\|^2\right]. \tag{181}$$

The stochastic interpolant framework allows for a more flexible choice of interpolants where $\alpha, \beta$, and $\gamma$ provide additional degrees of freedom compared to the interpolant obtained using the framework presented in this paper (see (221)).

### D.3. Connections to Flow Sampling

Flow Sampling was introduced by Havens et al. (2026) concurrently to our work. Here, we highlight its connections to our method and the stochastic interpolant framework introduced in Appendix D.2.

Consider the stochastic interpolant defined in (167) with $\gamma \equiv 0$ and an independent coupling such that

$$\nabla_{X_0}\log\nu_{0,T}(X_0, X_T) = \nabla_{X_0}\log p_{\mathrm{prior}}(X_0) \quad \text{and} \quad \nabla_{X_T}\log\nu_{0,T}(X_0, X_T) = \nabla_{X_T}\log p_{\mathrm{target}}(X_T). \tag{182}$$

Using the fixed-point equation for stochastic interpolants (181) gives

$$b_{\mathrm{F}}^{i+1} = \underset{\hat{b}_{\mathrm{F}}}{\arg\min}\, \mathbb{E}_{\nu^i}\left[\|\hat{b}_{\mathrm{F}}(I_t, t) - \dot{I}_t - \sigma(t)^2 \left(\tfrac{1-c(t)}{\alpha(t)}\nabla_{X_0}\log p_{\mathrm{prior}}(X_0) - \tfrac{c(t)}{\beta(t)}\nabla_{X_T}\log p_{\mathrm{target}}(X_T)\right)\|^2\right]. \tag{183}$$

Setting $c = 1$, $\alpha(t) = t$, $\beta(t) = 1 - t$ and $\sigma(t)^2 = \bar{\gamma}\alpha(t)$ for some $\bar{\gamma} \in \mathbb{R}$ recovers the Flow Sampling objective

$$b_{\mathrm{F}} = \underset{\hat{b}_{\mathrm{F}}}{\arg\min}\, \mathbb{E}_{\nu^i}\left[\|\hat{b}_{\mathrm{F}}(X_t, t) - (X_T - X_0 + \bar{\gamma}\nabla_{X_T}\log p_{\mathrm{target}}(X_T))\|^2\right]. \tag{184}$$

Hence, both Flow Sampling and the Bridge Matching Sampler can be seen as special cases of fixed-point diffusion matching for stochastic interpolants.

## E. Further algorithmic details

### E.1. Importance sampling and probability flow ordinary differential equation

**Importance sampling in path space.** Our primary goal is to estimate the expectation of an observable $\Omega \in C(\mathbb{R}^d, \mathbb{R})$ under a target distribution, denoted as $\mathbb{E}_{X_T \sim p_{\mathrm{target}}}[\Omega(X_T)]$. Since the optimized process $\mathbb{P}^u$ serves as an approximation to the true target dynamics (i.e., $\mathbb{P}_T^u \approx p_{\mathrm{target}}$), we can correct for the resulting bias by employing importance sampling. While importance sampling is classically applied to probability densities, it extends naturally to path space (Berner et al., 2024), allowing for the reweighting of entire continuous trajectories of a stochastic process. We can express the target expectation as an integral over the proposal measure $\mathbb{P}^u$:

$$\mathbb{E}_{X_T \sim p_{\mathrm{target}}}[\Omega(X_T)] = \mathbb{E}_{\mathbb{P}^u}\left[\frac{\mathrm{d}\overleftarrow{\mathbb{P}}^v}{\mathrm{d}\mathbb{P}^u}(X)\,\Omega(X_T)\right]. \tag{185}$$

Here, the Radon-Nikodym derivative (RND), $w(X) := \frac{\mathrm{d}\overleftarrow{\mathbb{P}}^v}{\mathrm{d}\mathbb{P}^u}(X)$, is given by an extension of Girsanov's theorem (Nüsken & Richter, 2021; Vargas et al., 2024; Richter & Berner, 2024):

$$w(X) = \frac{p_{\mathrm{target}}(X_T)}{p_{\mathrm{prior}}(X_0)}\exp\left(\int_0^T \sigma^{-1}v(X_t, t)\cdot\overleftarrow{\mathrm{d}}X_t - \int_0^T \sigma^{-1}u(X_t, t)\cdot\mathrm{d}X_t - \frac{1}{2}\int_0^T \left(\|v\|^2 - \|u\|^2\right)(X_t, t)\mathrm{d}t\right). \tag{186}$$

Practical implementation faces two distinct challenges. First, we typically only have access to $p_{\mathrm{target}}$ up to an intractable normalization constant $\mathcal{Z}$. We therefore resort to self-normalized importance sampling (Tokdar & Kass, 2010):

$$\frac{\mathbb{E}_{\mathbb{P}^u}[\widetilde{w}(X)\,\Omega(X_T)]}{\mathbb{E}_{\mathbb{P}^u}[\widetilde{w}(X)]}, \tag{187}$$

where $\widetilde{w} = w\mathcal{Z}$. The second challenge is that evaluating the RND requires access to both the forward and the backward drift $u, v$, respectively. However, our proposed fixed-point iteration yields only $u$, making the direct computation of (186) impossible. To resolve this, we leverage Nelson's identity (16) to formulate a regression problem for learning $v \approx v^*$ and

the score $s \approx \sigma \nabla \log \Pi^*$ separately, instead of $u$:

$$\mathcal{L}(v, s) := \mathbb{E}_{\Pi^i} \left[ \int_0^T \tfrac{1}{2} \| \xi^v(X, t) - v(X_t, t) \|^2 dt \right] + \mathbb{E}_{\Pi^i} \left[ \int_0^T \tfrac{1}{2} \| \xi^s(X, t) - s(X_t, t) \|^2 dt \right], \tag{188}$$

with

$$v^*(x, t) = \mathbb{E}_{\Pi^*} \left[ \xi^v(X, t) | X_t = x \right] \quad \text{and} \quad \sigma(t) \nabla \log \Pi_t^*(x) = \mathbb{E}_{\Pi^*} \left[ \xi^s(X, t) | X_t = x \right], \tag{189}$$

where $\xi^v = \sigma \nabla \log \mathbb{P}_{t|0}$ and $\xi^s$ is inferred via the generalized target score identity. Implementation-wise, it is possible to employ a shared neural backbone to approximate $v$ and $s$ simultaneously, potentially improving parameter efficiency. However, we leave the exploration of such architectural optimizations for future work. Finally, the forward process (1) can be simulated as

$$dX_t = \sigma(t) (s - v)(X_t, t) dt + \sigma(t) dB_t, \quad X_0 \sim p_{\text{prior}}. \tag{190}$$

**Probability flow ODE.** An alternative approach to importance sampling is to leverage the Probability Flow ODE (PF-ODE) (Maoutsa et al., 2020; Song et al., 2020; Chen et al., 2021). This deterministic process shares the same marginal densities $(\Pi_t^*)_{t \in [0, T]}$ as the optimally controlled stochastic process but follows a smooth trajectory defined by

$$dX_t = \underbrace{\left( \sigma u^* - \tfrac{1}{2} \sigma^2 \nabla \log \Pi_t^* \right)(X_t, t)}_{:= f(X_t, t)} dt. \tag{191}$$

By leveraging Nelson's identity (16), we can express the drift term $f$ in equivalent forms involving the backward control $v$, i.e.,

$$\sigma u^* - \tfrac{1}{2} \sigma^2 \nabla \log \Pi_t^* = -\sigma v^* + \tfrac{1}{2} \sigma^2 \nabla \log \Pi_t^* = \tfrac{1}{2} \sigma (u^* - v^*), \tag{192}$$

suggesting that one needs two quantities among $u, v, \nabla \log \Pi^*$ to simulate (191). Consequently, one can use the objective (188) to learn $v$ and the score $\nabla \log \Pi^*$ separately. A key advantage of the ODE formulation is the ability to compute exact log-likelihoods via the *instantaneous change of variables* formula (Chen et al., 2018). The evolution of the log-density is governed by the divergence of the drift field:

$$\log \mathbb{P}_T^u(X_T) = \log p_{\text{prior}}(X_0) - \int_0^T \nabla \cdot f(X_t, t) \, dt, \tag{193}$$

where $\nabla \cdot f = \text{Tr}(\nabla f)$ denotes the divergence. While this allows for exact density estimation, computing the trace of the Jacobian $\nabla f$ is computationally expensive for high-dimensional problems. To mitigate this cost, one can trade exactness for efficiency by employing unbiased estimators, such as Hutchinson's trace estimator (Hutchinson, 1989). Having access to $\mathbb{P}_T^u$ allows for importance sampling via (187) with $w = p_{\text{target}} / \mathbb{P}_T^u$.

### E.2. Variance reduction via control variates

The efficiency of learning the optimal drift $u^*$ is limited by the stochasticity of the target drift $\xi$. To formalize this, we observe that the matching objective decomposes into two distinct components via Lemma A.5:

$$\mathbb{E}_{\Pi^*} \left[ \int_0^T \tfrac{1}{2} \| u(X_t, t) - \xi(X, t) \|^2 dt \right] = \underbrace{\mathbb{E}_{\Pi^*} \left[ \int_0^T \tfrac{1}{2} \| u(X_t, t) - \mathbb{E}_{\Pi^*} \left[ \xi(X, t) | X_t \right] \|^2 dt \right]}_{\text{bias}} + \underbrace{\mathbb{E}_{\Pi^*} \left[ \int_0^T \tfrac{1}{2} \| \xi(X, t) - \mathbb{E}_{\Pi^*} \left[ \xi(X, t) | X_t \right] \|^2 dt \right]}_{\text{variance}}. \tag{194}$$

While the variance term,

$$\mathbb{E}_{\Pi^*} \left[ \int_0^T \tfrac{1}{2} \| \xi(X, t) - \mathbb{E}_{\Pi^*} \left[ \xi(X, t) | X_t \right] \|^2 dt \right] = \int_0^T \tfrac{1}{2} \mathbb{E}_{\Pi_t^*} \left[ \text{Var}_{\Pi_{|t}^*} \left( \xi(X, t | X_t) \right) \right] dt, \tag{195}$$

does not change the optimal drift $u^*$, it adds noise to the empirical gradient. To minimize its contribution, we first recall the definition of $\xi$:

$$\sigma^{-1}(t) \xi^c(X, t) = \frac{1 - c(t)}{1 - \gamma(t)} \nabla_{X_0} \log \Pi_{0,T}^*(X_0, X_T) + \frac{c(t)}{\gamma(t)} \nabla_{X_T} \log \Pi_{0,T}^*(X_0, X_T) - \nabla_{X_t} \log \mathbb{P}_{t|0}(X_t | X_0), \tag{196}$$

where we make the dependence of $\xi$ on $c$ explicit. In what follows, we show that $c$ can be used to reduce the empirical variance.

**Control variate interpretation of $c$.** We start by making the following definitions:

$$G_0(X) := \frac{1}{1-\gamma(t)}\nabla_{X_0}\log\Pi_{0,T}^*, \quad G_T(X) := \frac{1}{\gamma(t)}\nabla_{X_T}\log\Pi_{0,T}^*, \quad G_v(X) := \nabla_{X_t}\log\mathbb{P}_{t|0}. \quad (197)$$

Using these definitions, we can express the scaled target field as $\sigma^{-1}\xi^c = (1-c)G_0 + cG_T - G_v$. Crucially, both $G_0$ and $G_T$ are unbiased estimators of the true score $\nabla_x\log\Pi_t^*(x)$ when conditioned on $X_t$. This implies:

$$\mathbb{E}_{\Pi^*}\left[\sigma^{-1}(t)\xi^c(X,t)\,|\,X_t = x\right] = \mathbb{E}_{\Pi^*}\left[(1-c(t))G_0 + c(t)G_T - G_v\,|\,X_t = x\right] \quad (198)$$

$$= \underbrace{\mathbb{E}_{\Pi^*}\left[G_0 - G_v\,|\,X_t = x\right]}_{=\sigma^{-1}(t)u^*(x,t)} - c(t)\underbrace{\mathbb{E}_{\Pi^*}\left[G_0 - G_T\,|\,X_t = x\right]}_{=0}. \quad (199)$$

The term $\Delta G := G_0 - G_T$ acts as a zero-mean control variate. We can therefore optimize $c$ to minimize the variance contribution in (194) by solving:

$$c^* = \arg\min_c \int_0^T \tfrac{1}{2}\mathbb{E}_{\Pi_t^i}\left[\mathrm{Var}_{\Pi_{|t}^*}\left(\xi^c(X,t|X_t)\right)\right]\mathrm{d}t. \quad (200)$$

**Proposition E.1** (Optimal scalar-valued control variate). *For any time $t \in [0,T]$, noting that the estimators are zero-mean, we define the scalar variance (equivalent to the trace of the covariance matrix) as $\mathrm{Var}(A) := \mathbb{E}_{\Pi^*}[\|A\|^2]$ and the scalar cross-covariance as $\mathrm{Cov}(A,B) := \mathbb{E}_{\Pi^*}[(A)\cdot(B)]$. The optimal control variate $c^*(t)$ that minimizes the conditional variance of $\xi^c$ is given by*

$$c^*(t) = \frac{\mathrm{Var}(G_0) - \mathrm{Cov}(G_0, G_T) - \mathrm{Cov}(G_0, G_v) + \mathrm{Cov}(G_T, G_v)}{\mathrm{Var}(G_0) + \mathrm{Var}(G_T) - 2\mathrm{Cov}(G_0, G_T)}. \quad (201)$$

**Joint optimization of the drift and control variate.** In practice, evaluating the exact variances and covariances in the optimal schedule $c^*(t)$ is intractable. However, we can circumvent this limitation and still achieve variance reduction by jointly learning the drift $u$ and the schedule $c$. This approach is formally justified by the following proposition, which demonstrates that minimizing the fixed-point diffusion matching objective with respect to $c$ exactly minimizes the conditional variance of the target.

**Proposition E.2.** *Let $\xi^c$ be defined as in (196). It holds that*

$$c^* = \arg\min_c \mathbb{E}_{\Pi^*}\left[\int_0^T \tfrac{1}{2}\|u(X_t,t) - \xi^c(X,t)\|^2\mathrm{d}t\right] = \arg\min_c \int_0^T \tfrac{1}{2}\mathbb{E}_{\Pi_t^*}\left[\mathrm{Var}_{\Pi_{|t}^*}\left(\xi^c(X,t|X_t)\right)\right]\mathrm{d}t. \quad (202)$$

Because $c(t)$ strictly weights a zero-mean control variate, the conditional expectation of our target – and therefore the optimal drift $u^*$ – remains perfectly invariant to its value. Looking at the decomposition of our main matching loss, this implies that any gradient taken with respect to $c$ exclusively targets the irreducible conditional variance of $\xi^c$. Consequently, following (Ko & Geffner, 2025; Kahouli et al., 2025; Young et al., 2026) we can parameterize the control variate as a learnable function $c(t)$ (e.g., a lightweight neural network) and jointly train it alongside the drift network $u$, i.e.,

$$u^*, c^* = \arg\min_{u,c} \mathbb{E}_{\Pi^*}\left[\int_0^T \tfrac{1}{2}\|u(X_t,t) - \xi^c(X,t)\|^2\mathrm{d}t\right], \quad (203)$$

using the exact same objective. By treating the variance reduction as an end-to-end learning problem, the model can dynamically discover the optimal interpolation schedule during optimization.

Note that (198) only holds if the expectation is taken with respect to $\Pi^*$. However, when performing our proposed fixed-point iteration (15), we only have access to samples from $\Pi^i$. It is therefore unclear if a joint optimization (203) with samples from $\Pi^i$ leads to variance reduction. We leave the numerical investigation for future work. For completeness, we provide a parameterization of the control variate $c$ that avoids numerical instabilities at the boundaries $t=0$ and $t=T$ below.

While (203) assumes access to the true coupling $\Pi^*$, our fixed-point iteration (15) relies on samples from the intermediate distribution $\Pi^i$. The extent to which joint optimization preserves its variance-reduction properties under this distributional shift remains an open question, which we leave for future numerical study. For completeness, we detail below a parameterization of $c$ designed to prevent numerical instabilities at the boundaries $t=0$ and $t=T$.

**Parameterization of the learned control variate.** A critical consideration when learning $c$ is the numerical stability of the target drift $\xi^c$ at the temporal boundaries. As defined in (196), the terms weighting the boundary score estimators are scaled

by $(1 - \gamma(t))^{-1}$ and $\gamma(t)^{-1}$. Under standard interpolation schemes where $\gamma(0) = 0$ and $\gamma(T) = 1$, these fractions introduce division-by-zero singularities at $t = 0$ and $t = T$, respectively. To prevent the empirical gradient from exploding, the control variate must strictly satisfy the boundary conditions $c(0) = 0$ and $c(T) = 1$. This ensures the numerators vanish at the boundaries, allowing the limits to remain finite. To enforce this constraint by design while still retaining full expressivity, we parameterize the learnable schedule as

$$c^{\phi}(t) = \gamma(t) + \gamma(t)(1 - \gamma(t))\mathrm{NN}^{\phi}(t), \tag{204}$$

where $\mathrm{NN}^{\phi}(t)$ is a neural network with learnable parameters $\phi$. In this formulation, the base term $\gamma(t)$ provides a stable linear-equivalent baseline, while the multiplicative envelope $\gamma(t)(1 - \gamma(t))$ acts as a gating mechanism. It zeroes out the network's contribution exactly at the boundaries, satisfying $c^{\phi}(0) = 0$ and $c^{\phi}(T) = 1$. Substituting $c^{\phi}(t)$ back into the coefficients of (196) yields

$$\frac{c^{\phi}(t)}{\gamma(t)} = 1 + (1 - \gamma(t))\mathrm{NN}^{\phi}(t) \quad \text{and} \quad \frac{1 - c^{\phi}(t)}{1 - \gamma(t)} = 1 - \gamma(t)\mathrm{NN}^{\phi}(t), \tag{205}$$

and therefore (196) becomes

$$\sigma^{-1}(t)\xi^{\phi}(X, t) = \left(1 - \gamma(t)\mathrm{NN}^{\phi}(t)\right) \nabla_{X_0} \log \Pi_{0,T}^{*}(X_0, X_T) \tag{206}$$

$$+ \left(1 + (1 - \gamma(t))\mathrm{NN}^{\phi}(t)\right) \nabla_{X_T} \log \Pi_{0,T}^{*}(X_0, X_T) - \nabla_{X_t} \log \mathbb{P}_{t|0}(X_t|X_0). \tag{207}$$

Because both simplified coefficients are well-defined and bounded for all $t \in [0, T]$, this parameterization completely circumvents the boundary explosion, ensuring stable joint optimization of $u_{\theta}$ and $c_{\phi}$.

### E.3. Variance reduction via damping

To show that the damped fixed-point iteration proposed in Section 2.5 results in a variance reduction, we can rewrite the closed-form solution for $u_{i+1}$ as the solution to a modified, unconstrained regression problem by combining the two separate regression problems in Proposition 2.13, which yields.

$$u_{i+1} = \arg\min_{u \in \mathcal{U}} \mathbb{E}_{\Pi^i} \left[ \int_0^T \tfrac{1}{2}\|\xi_i^{\eta}(X, t) - u(X_t, t)\|^2 \mathrm{d}t \right], \quad \text{with} \quad \xi_i^{\eta}(X, t) = \tfrac{\eta}{1+\eta} u_i(X_t, t) + \tfrac{1}{1+\eta}\xi(X, t). \tag{208}$$

Applying the same bias-variance decomposition as in (194) to this new objective, the variance component that dictates the noise in the empirical gradients becomes

$$\mathbb{E}_{\Pi^i} \left[ \int_0^T \tfrac{1}{2}\|\xi_i^{\eta}(X, t) - \mathbb{E}_{\Pi^i}\left[\xi_i^{\eta}(X, t)|X_t\right]\|^2 \mathrm{d}t \right] = \int_0^T \tfrac{1}{2}\mathbb{E}_{\Pi_t^i} \left[ \mathrm{Var}_{\Pi_{|t}^i} \left(\xi_i^{\eta}(X, t|X_t)\right) \right] \mathrm{d}t. \tag{209}$$

Crucially, because the previous network state $u_i(X_t, t)$ is fully deterministic conditioned on the marginal state $X_t$, its conditional variance is zero. The conditional variance of the modified target therefore simplifies to

$$\mathrm{Var}_{\Pi_{|t}^i} \left(\xi_i^{\eta}(X, t|X_t)\right) = \mathrm{Var}_{\Pi_{|t}^i} \left(\tfrac{\eta}{1+\eta} u_i(X_t, t) + \tfrac{1}{1+\eta}\xi(X, t|X_t)\right) = \left(\tfrac{1}{1+\eta}\right)^2 \mathrm{Var}_{\Pi_{|t}^i} \left(\xi(X, t|X_t)\right). \tag{210}$$

Substituting this back into the integrated variance term yields a direct relationship to the original variance:

$$\mathbb{E}_{\Pi^i} \left[ \int_0^T \tfrac{1}{2}\|\xi_i^{\eta}(X, t) - \mathbb{E}_{\Pi^i}\left[\xi_i^{\eta}(X, t)|X_t\right]\|^2 \mathrm{d}t \right] = \left(\tfrac{1}{1+\eta}\right)^2 \int_0^T \tfrac{1}{2}\mathbb{E}_{\Pi_t^i} \left[ \mathrm{Var}_{\Pi_{|t}^i} \left(\xi(X, t|X_t)\right) \right] \mathrm{d}t. \tag{211}$$

For any strictly positive penalty $\eta > 0$, the scaling factor $\left(\tfrac{1}{1+\eta}\right)^2$ is strictly less than 1. Therefore, we can guarantee that:

$$\int_0^T \tfrac{1}{2}\mathbb{E}_{\Pi_t^i} \left[ \mathrm{Var}_{\Pi_{|t}^i} \left(\xi(X, t|X_t)\right) \right] \mathrm{d}t > \int_0^T \tfrac{1}{2}\mathbb{E}_{\Pi_t^i} \left[ \mathrm{Var}_{\Pi_{|t}^i} \left(\xi_i^{\eta}(X, t|X_t)\right) \right] \mathrm{d}t. \tag{212}$$

This demonstrates that the regularized objective inherently shrinks the conditional variance of the target by anchoring the highly stochastic path drift $\xi(X, t)$ to the deterministic prior prediction $u_i(X_t, t)$. As the penalty parameter $\eta$ increases, the variance is more aggressively reduced, albeit at the cost of heavily biasing the update towards the old model.

### E.4. Loss weighting and numerical stability

In this section, we analyze the variance properties of the matching objective and derive a weighting schedule to mitigate numerical instabilities near $t = 0$. We introduce a time-dependent weighting function $\omega(t) > 0$ to the primary matching objective. This weighting balances the contribution of the loss over time without altering the optimal solution $u^* \in \mathcal{U}$ at the critical points of the objective. The weighted loss is given by

$$\mathcal{L}(u) = \mathbb{E}_\Pi \left[ \int_0^T \omega(t) \left\| u(X_t, t) - \xi(X, t) \right\|^2 \mathrm{d}t \right]. \tag{213}$$

The target vector field $\xi(X, t)$ typically includes the score of the forward transition kernel $\nabla \log \mathbb{P}_{t|0}(X_t|X_0)$, which dominates the variance as $t \to 0$. For a general Gaussian reference process with variance schedule $\kappa(t) = \int_0^t \sigma^2(s)\mathrm{d}s$, the transition kernel is given by $\mathbb{P}_{t|0}(X_t|X_0) = \mathcal{N}(X_t; X_0, \kappa(t)I)$. Consequently, the scaled score term is

$$\sigma(t)\nabla_{X_t} \log \mathbb{P}_{t|0}(X_t|X_0) = \sigma(t)\frac{X_0 - X_t}{\kappa(t)}. \tag{214}$$

Substituting the interpolant $X_t = \frac{\kappa(T)-\kappa(t)}{\kappa(T)}X_0 + \frac{\kappa(t)}{\kappa(T)}X_T + \left(B_{\kappa(t)} - \frac{\kappa(t)}{\kappa(T)}B_{\kappa(T)}\right)$ into the score expression allows us to isolate the stochastic component. Note that the stochastic part corresponds to a Brownian bridge with variance $\frac{\kappa(t)(\kappa(T)-\kappa(t))}{\kappa(T)}$ which can be represented as $\sqrt{\frac{\kappa(t)(\kappa(T)-\kappa(t))}{\kappa(T)}}\varepsilon$, where $\varepsilon \sim \mathcal{N}(0, I)$. The expansion of the score term yields

$$\sigma(t)\nabla_{X_t} \log \mathbb{P}_{t|0}(X_t|X_0) = \underbrace{\frac{\sigma(t)}{\kappa(T)}(X_0 - X_T)}_{\text{drift}} - \underbrace{\sigma(t)\sqrt{\frac{\kappa(T) - \kappa(t)}{\kappa(T)\kappa(t)}}\varepsilon}_{\text{noise}}. \tag{215}$$

We now analyze the variance of the noise term, denoted by $\Sigma^2_{\text{noise}}(t)$, as $t \to 0$. Since $\kappa(0) = 0$, the term $\kappa(t)$ in the denominator causes a singularity. Approximating $\kappa(T) - \kappa(t) \approx \kappa(T)$ for small $t$, the variance behaves as

$$\Sigma^2_{\text{noise}}(t) = \sigma^2(t)\frac{\kappa(T) - \kappa(t)}{\kappa(T)\kappa(t)} \approx \frac{\sigma^2(t)}{\kappa(t)}. \tag{216}$$

Assuming $\sigma(t)$ is roughly constant near $t = 0$, $\kappa(t) \approx \sigma^2 t$, implying that the variance explodes as $\mathcal{O}(1/t)$. To enforce numerical stability, we select $\omega(t)$ to counteract this variance, ensuring the effective loss contribution remains bounded. The optimal choice, often referred to as the *noise-prediction* weighting, is the inverse of the asymptotic variance factor:

$$\omega(t) = \frac{\kappa(t)}{\sigma^2(t)}. \tag{217}$$

Applying this weighting to the loss function neutralizes the singularity. By absorbing the weight into the norm, we obtain a numerically stable parameterization, namely

$$\mathcal{L}(u) = \mathbb{E}_\Pi \left[ \int_0^T \left\| \frac{\sqrt{\kappa(t)}}{\sigma(t)}u(X_t, t) - \frac{\sqrt{\kappa(t)}}{\sigma(t)}\xi(X, t) \right\|^2 \mathrm{d}t \right] \tag{218a}$$

$$= \mathbb{E}_\Pi \left[ \int_0^T \left\| \widehat{u}(X_t, t) - \left( \frac{\sqrt{\kappa(t)}}{\kappa(T)}(X_0 - X_T) - \sqrt{1 - \frac{\kappa(t)}{\kappa(T)}}\varepsilon + \sqrt{\kappa(t)}\nabla \log \Pi_t^*(X_t) \right) \right\|^2 \mathrm{d}t \right], \tag{218b}$$

with $\widehat{u}(x, t) = \frac{\sqrt{\kappa(t)}}{\sigma(t)}u(x, t)$. Here, the coefficient of the noise $\varepsilon$ becomes $\sqrt{1 - \kappa(t)/\kappa(T)}$, which approaches $1$ as $t \to 0$, resulting in a stable, unit-variance objective throughout training. However, while the weighting $\omega(t)$ stabilizes training, it necessitates rescaling the drift of the forward SDE in (1) to recover the drift $u$. Substituting $u(x, t) = \frac{\sigma(t)}{\sqrt{\kappa(t)}}\widehat{u}(x, t)$ back into the forward process gives

$$\mathrm{d}X_t = \frac{\sigma^2(t)}{\sqrt{\kappa(t)}}\widehat{u}(X_t, t)\mathrm{d}t + \sigma(t)\mathrm{d}B_t. \tag{219}$$

*Table 5.* Overview: our framework considers target path measures of the form $\Pi^* = \Pi^*_{0,T}\mathbb{P}_{|0,T}$ with different couplings $\Pi^*_{0,T}$. We provide expressions for $\xi$ that satisfy $u^*(x,t) = \sigma(t)\mathbb{E}_{\Pi^*}[\xi(X,t)|X_t = x]$ and are used for the proposed fixed-point iteration scheme in Algorithm 1.

| Coupling | | Formula for $\sigma(t)^{-1}\xi(X)$ |
|---|---|---|
| Bridge dep. | $\Pi^*_{0,T}$ | $\frac{1-c(t)}{1-\gamma(t)}\nabla_{X_0}\log\Pi^*_{0,T}(X_0,X_T) + \frac{c(t)}{\gamma(t)}\nabla_{X_T}\log\Pi^*_{0,T}(X_0,X_T) - \nabla_{X_t}\log\mathbb{P}_{t|0}(X_t|X_0)$ |
| SHB | $\Pi^*_{0,T} = \mathbb{P}_0 \otimes \Pi^*_T$ | $\frac{\gamma(t)-c(t)}{1-\gamma(t)}\nabla_{X_0}\log\mathbb{P}_0(X_0) + \frac{c(t)}{\gamma(t)}\nabla_{X_T}\log\frac{\Pi^*_T(X_T)}{\mathbb{P}_T(X_T)} + \frac{\gamma(t)-c(t)}{\gamma(t)(1-\gamma(t))}\nabla_{X_0}\log\mathbb{P}_{0|T}(X_0|X_T)$ |
| SB | $\Pi^*_{0,T} = \Pi^{SB}_{0,T}$ | $\frac{\gamma(t)-c(t)}{1-\gamma(t)}\nabla_{X_0}\log\frac{\Pi^*_0(X_0)}{\varphi_0(X_0)} + \frac{c(t)}{\gamma(t)}\nabla_{X_T}\log\frac{\Pi^*_T(X_T)}{\widehat{\varphi}_T(X_T)} + \frac{\gamma(t)-c(t)}{\gamma(t)(1-\gamma(t))}\nabla_{X_0}\log\mathbb{P}_{T|0}(X_T|X_0)$ |
| Bridge indep. | $\Pi^*_{0,T} = \Pi^*_0 \otimes \Pi^*_T$ | $\frac{1-c(t)}{1-\gamma(t)}\nabla_{X_0}\log\Pi^*_0(X_0) + \frac{c(t)}{\gamma(t)}\nabla_{X_T}\log\Pi^*_T(X_T) - \nabla_{X_t}\log\mathbb{P}_{t|0}(X_t|X_0)$ |

This formulation reveals a numerical challenge near the boundary $t = 0$. Assuming $\sigma(t) \approx \sigma$ is constant near the origin, we have $\kappa(t) \approx \sigma^2 t$. Consequently, the effective drift coefficient scales as

$$\frac{\sigma^2(t)}{\sqrt{\kappa(t)}} \approx \frac{\sigma^2}{\sigma\sqrt{t}} \propto \frac{1}{\sqrt{t}}. \tag{220}$$

This $t^{-1/2}$ singularity implies that the drift velocity diverges as $t \to 0$. Standard fixed-step solvers (e.g., Euler-Maruyama) may exhibit instability or large discretization errors if evaluated strictly at $t = 0$. In practice, this is mitigated by starting the integration at a small cutoff time, e.g., $t = 10^{-3}$.

### E.5. Reference process

In this work, we use a reference measure $\mathbb{P}$ induced by scaled Brownian motion. The dynamics are governed by the driftless SDE $dX_t = \sigma(t)dB_t$, where $\sigma(t)$ is a time-dependent noise schedule. To characterize the path properties, we define the cumulative variance schedule as $\kappa(t) := \int_0^t \sigma^2(s)ds$. When conditioned on fixed boundary values $X_0 = x_0$ and $X_T = x_T$, the process admits a closed-form representation known as a diffusive interpolant (Albergo et al., 2025):

$$X_t = \frac{\kappa(T) - \kappa(t)}{\kappa(T)}x_0 + \frac{\kappa(t)}{\kappa(T)}x_T + \left(B_{\kappa(t)} - \frac{\kappa(t)}{\kappa(T)}B_{\kappa(T)}\right). \tag{221}$$

Using $\gamma(t) := \frac{\kappa(t)}{\kappa(T)}$, we recover the simplified notation

$$X_t = (1 - \gamma(t))x_0 + \gamma(t)x_T + \mathcal{B}^\gamma_t, \tag{222}$$

where $\mathcal{B}^\gamma_t$ is a Gaussian process with mean zero and covariance $\kappa(T)\gamma(t)(1-\gamma(t))I$. This analytic form allows for the direct simulation of bridge states $X_t \sim \mathbb{P}_{t|0,T}(\cdot|x_0, x_T)$ without requiring numerical integration of the SDE. The relevant transition densities are given by

$$\mathbb{P}_{t|0,T}(x_t|x_0, x_T) = \mathcal{N}\big(x_t|(1-\gamma(t))x_0 + \gamma(t)x_T, \ \kappa(T)\gamma(t)(1-\gamma(t))I\big), \tag{223}$$

$$\mathbb{P}_{t|0}(x_t|x_0) = \mathcal{N}(x_t|x_0, \ \kappa(t)I), \tag{224}$$

$$\mathbb{P}_{T|t}(x_T|x_t) = \mathcal{N}(x_T|x_t, \ \kappa(T)(1-\gamma(t))I). \tag{225}$$

### E.6. Alternative characterization of the non-Markovian drift $\xi$

Here, we present an alternative characterization of the path-dependent non-Markovian drift $\xi$ compared to the one in Proposition 2.7. The expression is given in Proposition E.3.

**Proposition E.3** (Alternative characterization of $\xi$). *Let $\mathbb{P}$ be the reference process induced by scaled Brownian motion $dX_t = \sigma(t)dB_t$. Define $\kappa(t) := \int_0^t \sigma^2(s)ds$ and $\gamma(t) := \frac{\kappa(t)}{\kappa(T)}$. The path dependent drift $\xi$ satisfies*

$$\xi(X,t) = \sigma(t)\left(\frac{(1-c(t))\gamma(t)}{1-\gamma(t)}\nabla_{X_0}\log\Pi^*_{0,T}(X_0,X_T) + c(t)\nabla_{X_T}\log\Pi^*_{0,T}(X_0,X_T) - \nabla_{X_T}\log\mathbb{P}_{T|0}(X_T|X_0)\right), \tag{226}$$

*with $u^*(x,t) = \sigma(t)\mathbb{E}_{\Pi^*}[\xi(X)|X_t = x]$.*

---

**Algorithm 2** Damped Fixed-Point Diffusion Matching (detailed version)

---

**Require:** Neural network $u_\theta$ with parameters $\theta$, diffusion schedule $\sigma$, target path measure $\Pi^*$, buffer size $K$, Number of outer steps $I$, number of gradient steps $M$ per outer step, damping $\eta$

    Initialize $i = 0$

    **for** $i = 0, \ldots, I - 1$ **do**

        Set $u_i = u_\theta$ (detached)

        *Simulation:* Simulate $K$ trajectories $\left\{ X^{(k)} \sim \mathbb{P}^{u_i} \right\}_{k=1}^{K}$ using $u_i$

        *Coupling:* Let $\Pi_{0,T}^i = \begin{cases} \mathbb{P}_{0,T}^{u_i}, & \text{ASBS (Liu et al., 2025)} \\ \mathbb{P}_0 \otimes \mathbb{P}_T^{u_i}, & \text{AS (Havens et al., 2025)} \\ p_{\text{prior}} \otimes \mathbb{P}_T^{u_i}, & \text{BMS (ours)} \end{cases}$

        *Buffer:* Initialize buffer $\mathcal{B} = \left\{ (X_0^{(k)}, X_T^{(k)}) \sim \Pi_{0,T}^i \right\}_{k=1}^{K}$

        **for** $m = 1, \ldots, M$ **do**

            Estimate $\mathcal{L}(\theta) = \mathbb{E}_{(X_0, X_T) \sim \mathcal{B}, \, X_t \sim \mathbb{P}_{t|0,T}} \left[ \frac{1}{2} \|\xi(X,t) - u_\theta(X_t, t)\|^2 + \frac{\eta}{2} \|u_i(X_t, t) - u_\theta(X_t, t)\|^2 \right]$

            Perform a gradient-descent step on $\mathcal{L}(\theta)$

        **end for**

    **end for**

    **Return** control $u_\theta$ with $\mathbb{P}_t^{u_\theta} \approx \Pi_t^*$ for all $t \in [0, T]$

---

The main difference between Proposition E.3 and Proposition 2.7 is that the former does not suffer from the numerical instability at $t = 0$ that arises through the term $\mathbb{P}_{t|0}$ as explained in Appendix E.4. We leave the numerical evaluation of this alternative expression for future work.

### E.7. Detailed algorithmic description

We provide a detailed description of our framework in Algorithm 2.

## F. Experimental setup

### F.1. Gaussian mixture models

**Target density specifications.** We consider a *Gaussian mixture model (GMM)* target of the form

$$p_{\text{GMM}}(x) = \sum_{k=1}^{K} \pi_i \, \mathcal{N}(x|\mu_i, \Sigma_i), \tag{227}$$

where $\mu_i \in \mathbb{R}^d$, $\Sigma_i \in \mathbb{R}^{d \times d}$, $\pi_i \geq 0$, and $\sum_{k=1}^{K} \pi_i = 1$. For our experiments, we set $\pi_i = 1/K$, $\Sigma_i = I$. The means $\mu_i$ are uniformly sampled from $[-K, K]^d$.

**Evaluation criteria.**

The *mode TVD* metric is proposed by Blessing et al. (2025a) and inspired by recent work (Blessing et al., 2024; Grenioux et al., 2025). It uses the ground truth mixing weights $\pi_i$, $k \in \{1, \ldots, K\}$, along with a partition $\{S_1, \ldots, S_i\}$ of $\mathbb{R}^d$, where each region $S_i \subset \mathbb{R}^d$ corresponds to the $k$-th mode of the target distribution given as

$$S_i = \{x \in \mathbb{R}^d | \arg\max_j \pi_j \mathcal{N}(x|\mu_j, \Sigma_j) = k\}. \tag{228}$$

We estimate the empirical mixing weights using

$$\widehat{\pi}_i = \frac{\mathbb{E}_{\mathbb{P}^u} \left[ \mathbb{1}_{S_i}(X_T) \right]}{\sum_{k'=1}^{K} \mathbb{E}_{\mathbb{P}^u} \left[ \mathbb{1}_{S_{k'}}(X_T) \right]}. \tag{229}$$

Using these estimates, we compute the total variation distance (TVD) between the empirical and true mode weights as

$$\text{mode TVD} = \sum_{k=1}^{K} |\pi_i - \widehat{\pi}_i|, \tag{230}$$

where we approximate the expectation in (229) using $2k$ samples. Note that mode TVD $\in [0, 1]$ where a value of 0 indicates that the model learned the same mixture weights as the target and therefore did not suffer from any mode-collapse.

sliced TVD ↓

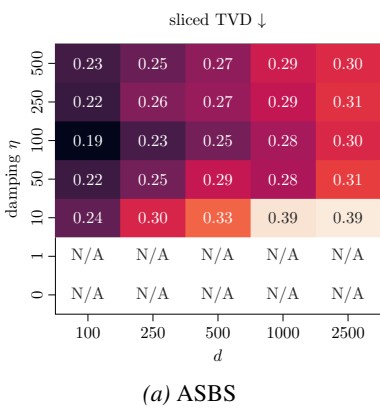

sliced TVD ↓

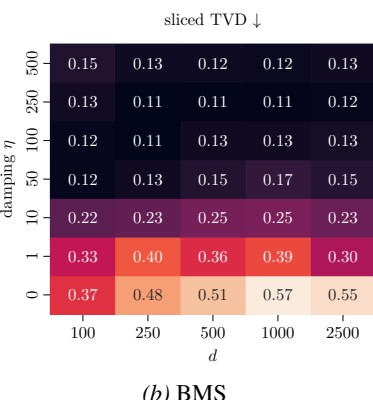

*(a)* ASBS

*(b)* BMS

*Figure 6.* Comparison of sliced TVD values on Gaussian mixture models across varying dimensions $d$ and damping values $\eta$. Figure 6a shows that ASBS exhibits instability at low damping values (N/A) and performance degradation as dimensionality increases. Figure 6b demonstrates that our method maintains stable training and consistently avoids mode collapse up to $d = 2500$, significantly scaling beyond the $d = 50$ range common in recent literature. All values are averaged over three random seeds.

Additionally, we use *sliced TVD* to assess the accuracy of the density fitting by comparing empirical densities against the analytically available marginals. For a set of model samples $\{x_n\}_{n=1}^N$ and target samples $\{y_m\}_{m=1}^M$, we project the data onto $P$ random unit vectors $\{\theta_p\}_{p=1}^P$ sampled uniformly from the unit sphere $\mathbb{S}^{d-1}$. For each projection, we compute the 1D total variation distance between the resulting histograms:

$$\text{sliced TVD} = \frac{1}{P} \sum_{p=1}^P \text{TVD}(\text{hist}(\{x_n \cdot \theta_p\}), \text{hist}(\{y_m \cdot \theta_p\})). \tag{231}$$

In our experiments, we use $P = 100$ projections and 50 bins to estimate these 1D distributions. Lastly, we compare the Wasserstein-2 ($W_2$) optimal transport distance between $2k$ model and target samples using the POT library (Flamary et al., 2021).

**Algorithm specifications.** For the all GMM experiments we used the same setting for ASBS and BMS: A residual network with 6 layers and 512 hidden units and Fourier time embedding is used. The source distribution is modeled as a normal distribution which is adapted to the support of the target by setting $\mathcal{N}(0, K^2)$. For optimization, the Adam optimizer (Kingma, 2014) with a learning rate of $10^{-4}$ and (value) gradient clipping at 1 is used. Furthermore, we leverage a batchsize of 1024 and a buffer size of $30k$. We perform $1k$ outer steps where each outer step entails $1k$ gradient steps on the current buffer. The diffusion process is integrated using the Euler-Maruyama method with 100 discretization steps. For the GMM experiments, we used a constant noise scheduler, i.e. $\sigma(t) = \sigma$. For both ASBS and BMS, we tested values in $\{1.5, 2.5, 4\}$ and found that $\sigma = 2.5$ works best for both methods. In our attempts to compare to adjoint sampling (Havens et al., 2025) we required larger values for $\sigma$ in order to roughly cover the support of the target. Particularly, we set $\sigma = K$, as the means are sampled from $[-K, K]^d$. We conjecture that these large $\sigma$ values caused the instabilities in training. For all algorithms, we used $c = \gamma$ when computing $\xi$.

### F.2. $n$-body identical particle systems

**Target density specifications.** Following the configurations in Köhler et al. (2020) and Akhound-Sadegh et al. (2024), we evaluate our method on two primary particle systems defined by their potential energy functions $E$ as

$$p_{\text{target}}(x) = \frac{e^{-E(x)}}{\mathcal{Z}}. \tag{232}$$

The *Double Well Potential (DW-4)* models a system of $n = 4$ particles in $\mathbb{R}^2$ ($d = 8$). The energy is defined by a pairwise distance potential

$$E_{\text{DW}}(x) = \frac{1}{\tau} \sum_{i<j} \left[ a(d_{ij} - d_0) + b(d_{ij} - d_0)^2 + c(d_{ij} - d_0)^4 \right], \tag{233}$$

where $d_{ij} = \|x_i - x_j\|_2$. We set parameters $a = 0, b = -4, c = 0.9$, and temperature $\tau = 1$.

The *Lennard-Jones (LJ)* potential describes the intermolecular interactions of $n$ particles in $\mathbb{R}^3$, where we consider systems

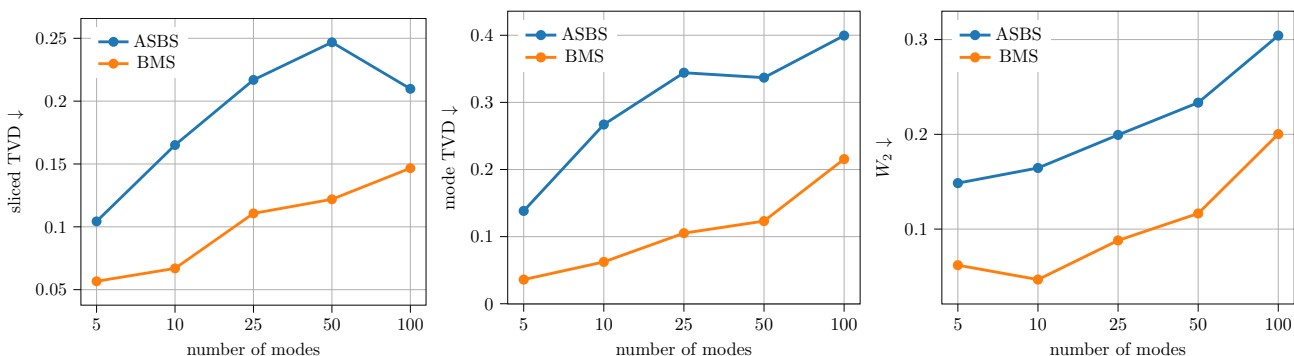

*Figure 7.* Comparison of *mode TVD*, *sliced TVD* and Wasserstein-2 $W_2$ on a Gaussian mixture model target in $d = 100$ with different number of modes. All values are averaged over three random seeds.

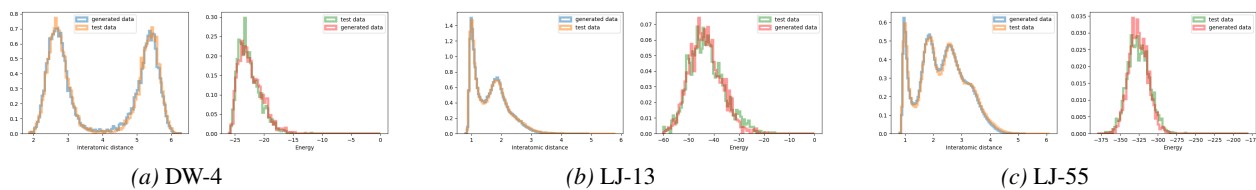

*(a)* DW-4                         *(b)* LJ-13                         *(c)* LJ-55

*Figure 8.* Visualization of interatomic distances and energy histograms of different $n$-body identical particle systems. The blue and red curves are generated from our BMS method, and compared to the orange and green curves obtained from a long run MCMC.

of $n = 13$ particles ($d = 39$) and $n = 55$ particles ($d = 165$). The total energy $E_{\text{Tot}}(x)$ is the sum of the LJ pairwise potential and a harmonic oscillator potential $E_{\text{osc}}$ used to prevent system drift

$$E_{\text{Tot}}(x) = \frac{\varepsilon}{\tau} \sum_{i<j} \left[ \left( \frac{r_m}{d_{ij}} \right)^{12} - 2 \left( \frac{r_m}{d_{ij}} \right)^6 \right] + c_{\text{osc}} \sum_i \|x_i - x_{\text{COM}}\|^2, \tag{234}$$

where $x_{\text{COM}}$ denotes the center of mass. Consistent with Köhler et al. (2020) and Akhound-Sadegh et al. (2024), we use $r_m = 1, \tau = 1, \varepsilon = 1$, and an oscillator scale $c_{\text{osc}} = 1.0$.

**Evaluation criteria.** For the evaluation criteria, we refer to Havens et al. (2025) (Appendix E.4 Reported Metrics) for detailed information on the evaluation criteria.

**Algorithm specifications.** We integrated our method into the code base of Liu et al. (2025) (see github.com/facebookresearch/adjoint_samplers) and re-used their hyperparameter settings and model architecture. As such, we refer to Liu et al. (2025) for a detailed specification of architectures and parameters. For tuning our method, we varied the scale of the initial distribution $\mathcal{N}(0, \sigma_{\text{init}}^2 I)$ and $\sigma_{\max}$ of the geometric noise scheduler, i.e.,

$$\sigma(t) = \sigma_{\min} \left( \frac{\sigma_{\max}}{\sigma_{\min}} \right)^{1-t} \sqrt{2 \log \frac{\sigma_{\max}}{\sigma_{\min}}}. \tag{235}$$

Specifically, we tested $\sigma_{\text{init}} \in \{0.5, 1, 2\}$ and $\sigma_{\max} \in \{0.5, 1, 1.5, 2\}$. We obtained the following parameter, which led to the best performance: For DW-4 we got $\sigma_{\text{init}} = 2; \sigma_{\max} = 1.5$, for LJ-13 $\sigma_{\text{init}} = 1; \sigma_{\max} = 1$ and LJ-55 $\sigma_{\text{init}} = 0.5; \sigma_{\max} = 0.5$. Lastly, for all algorithms, we used $c = \gamma$ when computing $\xi$. The results reported are without using damping, i.e., $\eta$ is set to zero.

### F.3. Molecular benchmarks: Peptide systems

**Target density specifications.** The target density for the Alanine dipeptide and tetrapeptide systems follow the configuration in Nam et al. (2025) and are defined over the configuration space of their 22 and 42 constituent atoms, respectively. In this work, the model is trained directly on the raw Cartesian coordinates of the molecule purely from energy evaluations. The distribution of molecular states is governed by the Boltzmann distribution, which defines the unnormalized target density as

$$p_{\text{target}}(x) \propto \exp \left( -\frac{E(x)}{k_B \tau} \right), \tag{236}$$

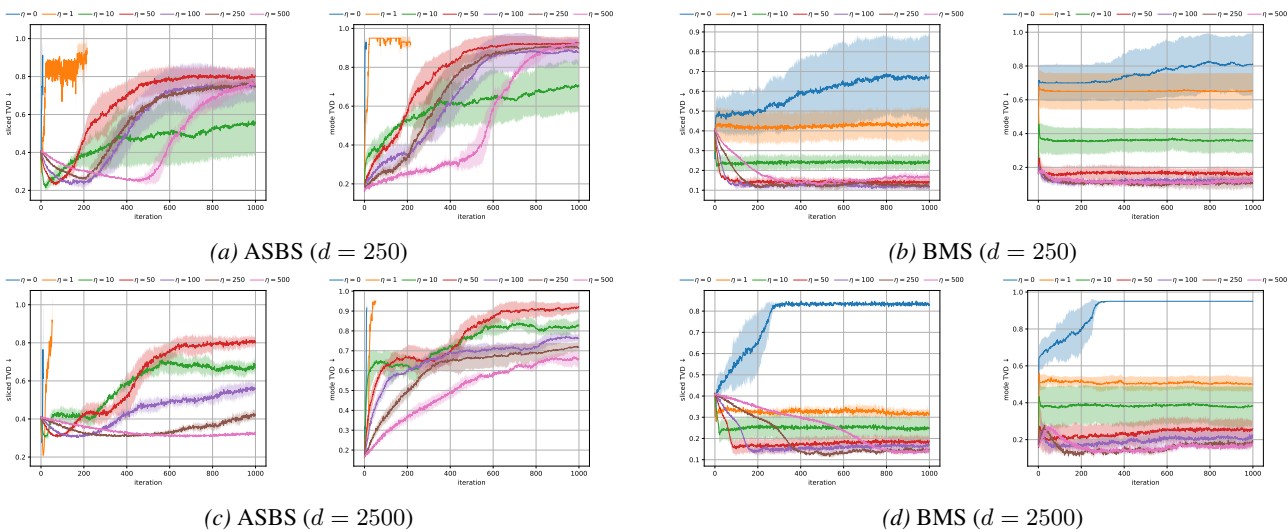

*Figure 9.* Comparison of *mode TVD* and *sliced TVD* on Gaussian mixture models across varying damping values $\eta$ for $d = 250$ and $d = 2500$ visualized over training iterations. All values are averaged over three random seeds.

where $E(x)$ represents the potential energy of the molecular configuration, $k_B$ is the Boltzmann constant, and $\tau$ is the thermodynamic temperature which is set to $\tau = 300K$. The potential energy of the system uses the Amber ff96 force field (Kollman et al., 1997) with GBSA-OBC implicit solvation (Onufriev et al., 2004) and uses OpenMM (Eastman et al., 2023). Moreover, we follow Nam et al. (2025) to constrain the chirality of the generated molecules.

**Evaluation criteria.** To quantify the performance of our sampler on the Alanine dipeptide and tetrapeptide systems, we evaluate the learned density against ground-truth Molecular Dynamics (MD) data using Ramachandran plots, which serve as a standard visualization in structural biology to represent the joint distribution of the backbone dihedral angles $\phi$ and $\psi$ (Noé et al., 2019). To measure the local backbone structural fidelity, we compared the probability distributions of the backbone dihedral angles, $\phi$ and $\psi$. For each residue $i$, we computed the 2D empirical probability density functions, $P_i(\phi, \psi)$ for the reference MD data and $P_i^{\mathrm{MD}}(\phi, \psi)$ for the generated samples. These distributions were discretized onto a $50 \times 50$ grid spanning $[-\pi, \pi] \times [\pi, \pi]$. The similarity between the two distributions for the $i$th residue was quantified using the Jensen-Shannon divergence, defined as

$$D_{\mathrm{JS}}^i(P_i | P_i^{\mathrm{MD}}) = \frac{1}{2} D_{\mathrm{KL}}(P_i | M_i) + \frac{1}{2} D_{\mathrm{KL}}(P_i^{\mathrm{MD}} | M_i), \tag{237}$$

where $M_i = \frac{1}{2}(P_i + P_i^{\mathrm{MD}})$ is the mixture distribution, and $D_{\mathrm{KL}}$ denotes the Kullback-Leibler divergence. The final reported metric is the average Jensen-Shannon divergence across all $N_{\mathrm{res}}$ residues:

$$(\phi, \psi) - D_{\mathrm{JS}} := \frac{1}{N_{\mathrm{res}}} \sum_{i=1}^{N_{\mathrm{res}}} D_{\mathrm{JS}}(P_i | Q_i). \tag{238}$$

While $D_{\mathrm{JS}}$ measures the divergence of probability distributions, we also evaluated the thermodynamic alignment by comparing the implied Free-Energy Surfaces (FES) of the backbone dihedrals. Using the same $50 \times 50$ discrete probability distributions $P_i(\phi, \psi)$ obtained above, the free energy $\mathrm{F}_i$ for a given bin $b$ is estimated via the Boltzmann inversion

$$\mathrm{F}_i(b) := -k_B T \log(P_i(b) + \varepsilon), \tag{239}$$

where $k_B T \approx 0.596$ kcal/mol (assuming $T = 300$ K) and $\varepsilon = 10^{-10}$ is a small constant added for numerical stability. To account for relative free-energy differences rather than absolute values, each surface was zero-centered by subtracting its global minimum, i.e., $\mathrm{F}_i'(b) = \mathrm{F}_i(b) - \min_b \mathrm{F}_i(b)$. To avoid noise from highly under-sampled, high-energy regions, we restricted our comparison to a valid subspace $\mathcal{M}_i$. A bin was included in $\mathcal{M}_i$ if its probability in *either* the reference or the generated ensemble exceeded $10^{-4}$. The Root-Mean-Squared Error for residue $i$ is then computed as

$$\mathrm{RMSE}_{\Delta \mathrm{F}, i} = \sqrt{\frac{1}{|\mathcal{M}_i|} \sum_{b \in \mathcal{M}_i} \left( \mathrm{F}_{\mathrm{ref}, i}'(b) - \mathrm{F}_{\mathrm{gen}, i}'(b) \right)^2}. \tag{240}$$

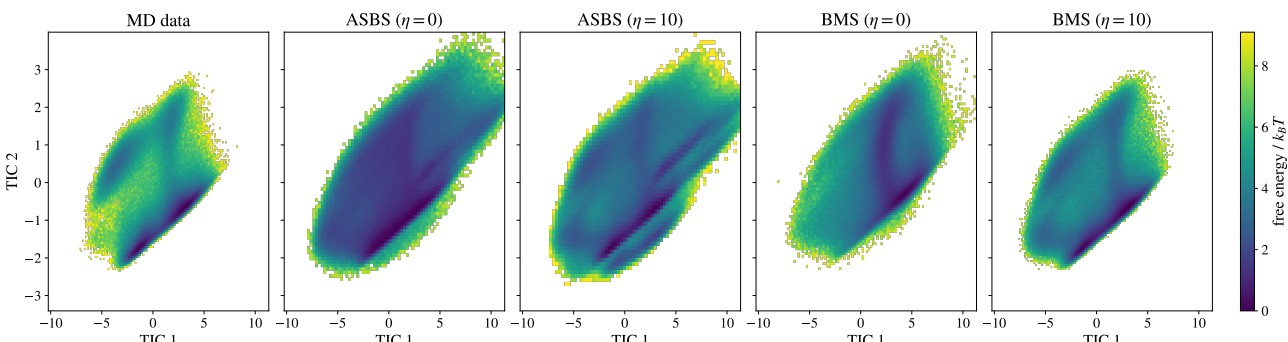

*Figure 10.* TICA plots with $10^6$ samples for Alanine Dipeptide for molecular dynamics (MD) data, ASBS (Liu et al., 2025), and our proposed method, BMS, for damping values $\eta \in \{0, 10\}$.

The overall FES discrepancy is reported as the mean $\text{RMSE}_{\Delta F, i}$ across all residues. To capture global conformational dynamics and long-range structural correlations, we evaluated the generated ensembles in a reduced, kinetically relevant latent space. We employed Time-lagged Independent Component Analysis (TICA), which identifies the slowest collective motions in a molecular system by finding projections that maximize the autocorrelation of the time-series MD data at a given lag time. We refer to Tan et al. (2025) for a more detailed description.

All results for $D_{\text{JS}}$ and $\text{RMSE}_{\Delta F}$ except those presented in Figure 5 are computed using $10^6$ model and MD data samples. The latter are taken from Schopmans & Friederich (2025a). Due to computational constraints, we used $10^4$ samples for computing the TICA-$W_2$ metric. Lastly, the values in Figure 5 are computed during training on the replay buffer and therefore use $\approx 16k$ samples.

**Algorithm specifications.** Our implementation builds on the codebase by Nam et al. (2025) (see github.com/facebookresearch/wt-asbs), where we removed the collective variables as we are interested in the performance without using prior knowledge. We use the same hyperparameter setting for ASBS and our method, as detailed below.

For the model architecture, the PaiNN architecture (Schütt et al., 2021) was used, an E(3)-equivariant graph neural network, modified to include time-conditioning inputs and vector product layers (Schreiner et al., 2023) to break parity symmetry, effectively restricting the model to SE(3)-equivariance (Liao et al., 2023). To improve training stability, equivariant layer normalization is applied. For ALA2 with $\eta = 0$, we follow the setting of (Nam et al., 2025) and employ 4 layers, a hidden dimension of 128 and 250 diffusion steps, as we were facing numerical instabilities when using larger architectures. For $\eta = 10$, we used 6 layers, a hidden dimension of 256, and 50 diffusion steps, training stably. For ALA4, we also use 6 layers, a hidden dimension of 256, and 50 diffusion steps. For the damped version we set $\eta = 50$. A radial cutoff of 8.0 Å is used for graph construction for all methods and systems. For all algorithms, we used $c = \gamma$ when computing $\xi$. The noise schedule is defined by the EDM variance exploding schedule (Karras et al., 2022), i.e.,

$$\sigma(t) = \left[ (1 - t)\sigma_{\max}^{1/\rho} + t\sigma_{\min}^{1/\rho} \right]^\rho, \tag{241}$$

with a minimum noise level $\sigma_{\min} = 0.001\,\text{Å}$, a maximum noise level $\sigma_{\max} = 6.0\,\text{Å}$, and an exponent $\rho = 3.0$. The source distribution is modeled as a normal distribution $\mathcal{N}(0, 1\,\text{Å}^2)$. For optimization, the AdamW optimizer (Loshchilov & Hutter, 2017) with a weight decay of $10^{-3}$. We employ a cosine annealing learning rate schedule, decaying from an initial learning rate of $1 \times 10^{-4}$ to a final learning rate of $3 \times 10^{-5}$. To stabilize training, energy gradients are clipped at a maximum value of 100.0. The training utilizes a replay buffer with a capacity of 16,384 samples. At each outer step, we draw 2,048 samples from the buffer and perform $L = 100$ gradient updates. The diffusion process is integrated using the Euler-Maruyama method with 250 discretization steps. For ALA2 and ALA4, we trained for $5k$ and $8k$ outer steps, respectively. For ASBS, we follow (Nam et al., 2025) and perform 500 corrector matching steps after every 1000 outer steps. Note that we do not count these corrector matching steps towards the outer iterations, meaning that we provide more computational budget to ASBS. For model selection, we always picked the last checkpoint.

## G. Further numerical results

Here, we provide additional results that complement the results in Section 4.

**Gaussian mixture models.** For Gaussian mixture models, we additionally report the sliced total variation distance (TVD);

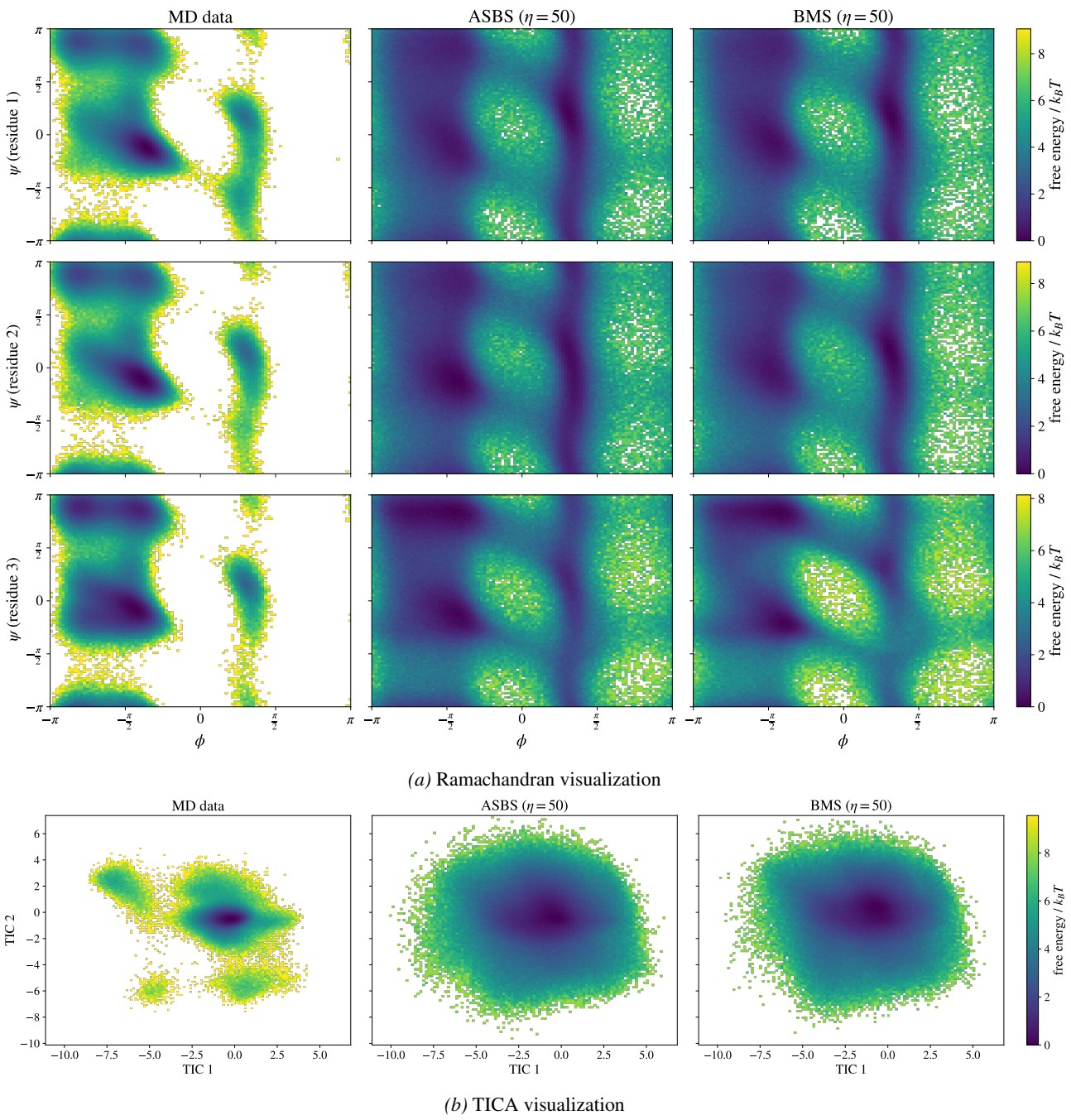

*(a)* Ramachandran visualization

*(b)* TICA visualization

*Figure 11.* Ramachandran (Figure 11a) and TICA (Figure 11b) plots with $10^6$ samples for Alanine Tetrapeptide for molecular dynamics (MD) data, ASBS (Liu et al., 2025), and our proposed method, BMS, for damping values $\eta = 50$.

see Appendix F.1 for further details. The results are shown in Figure 6 and are consistent with the findings in Section 4. Additionally, we test ASBS and our method on an increasing number of modes $K$ sampled from $[-K, K]^d$. The results are shown in Figure 7 and indicate that our method's performance is less degrading when increasing the number of modes.

$n$**-body identical particle systems.** For reference, we provide visualizations of the energy histograms and interatomic distances in Figure 8. Moreover, we visualize the performance over training iterations in Figure 9

**Molecular benchmarks: Peptide systems.** Here, we provide additional visualizations for peptide systems. Specifically, Figure 10 shows the TICA plot for ALA2 and Figure 11 shows the TICA and Ramachandran plots for ALA4.

**Comparison to classical Monte Carlo methods.** We compare our method against three classical Monte Carlo baselines on Gaussian mixture model (GMM) targets across varying dimensions $d \in \{5, 10, 25, 50, 100, 250, 500, 1000, 2500\}$. As baselines, we consider *Annealed Importance Sampling* (AIS) (Neal, 2001), *Sequential Monte Carlo* (SMC) (Del Moral et al., 2006), and *Parallel Tempering Markov chain Monte Carlo* (PT-MCMC) (Swendsen & Wang, 1986). All three methods use Metropolis-adjusted Langevin algorithm (MALA) transitions, using the same number of target energy evaluations as the neural sampling methods (BMS/ASBS). We evaluate performance using *mode TVD* and *Wasserstein-2* ($W_2$) metrics, averaged over three random seeds. Figure 12 summarizes the results: SMC is prone to mode collapse due to resampling, which is avoided when using AIS. PT-MCMC achieves the best performance among the baselines when the MALA step size is chosen properly.

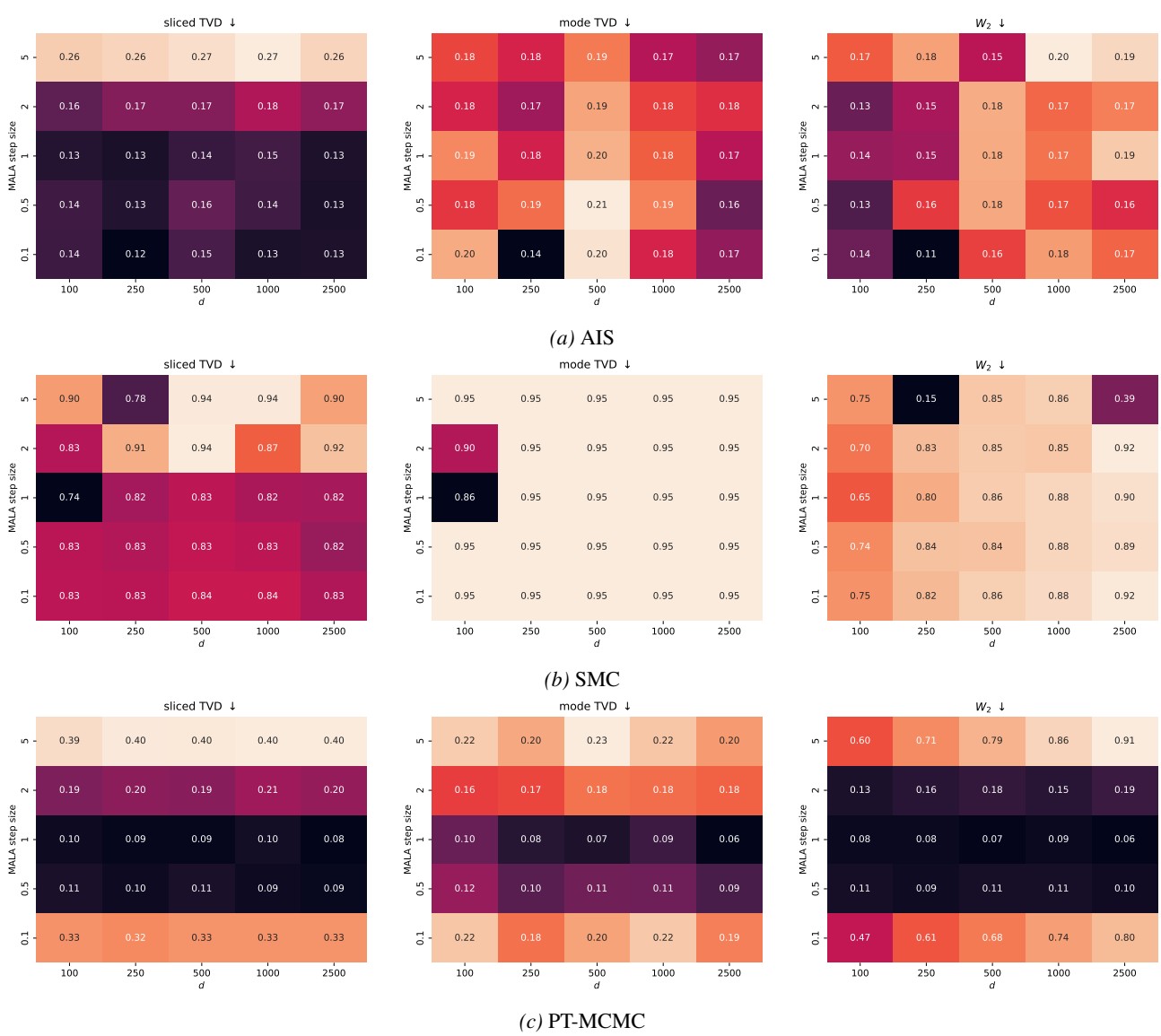

*Figure 12.* Comparison of *mode TVD* and *Wasserstein-2 ($W_2$)* values on Gaussian mixture models across varying dimensions $d$ and Metropolis adjusted Langevin algorithm (MALA) step sizes using the same number of target evaluations as the neural sampling methods. All values are averaged over three random seeds. SMC suffers from mode collapse due to resampling which is avoided when using AIS. PT-MCMC works best if the MALA step size is tuned properly.

