# OpenReview forum: "Bridge Matching Sampler: Scalable Sampling via Generalized Fixed-Point Diffusion Matching"
_ICML.cc/2026/Conference — ICML 2026 regular_

### Official Review · Reviewer_bUR2 · 2026-03-05

**Soundness:** 3
**Presentation:** 4
**Significance:** 3
**Originality:** 4
**Overall Recommendation:** 5
**Confidence:** 3

**Summary:**

This work generalizes and illuminates the theory behind existing "matching" samplers such as the adjoint sampler. The core idea is to Markovianize the non-Markovian dynamics arising from the bridge formulation of the transport problem. In addition, techniques of damping and control variates extend the practical applicability of these methods. Applications including Alanine Dipeptide are considered.

**Compliance With Llm Reviewing Policy:**

Affirmed.

**Key Questions For Authors:**

In the qualitative example of the Alanine Dipeptide system, what are the white patches? It would seem like neither ASBS nor BMS actually works very well at all, assuming that the MD results are a ground truth. Or have I misunderstood the plot?

More broadly, what should my takeaways be here about using diffusion/flow based samplers in place of MCMC/MD methods? In this field, I see suspiciously few quantitative comparisons of MCMC/MD to these ML methods that actually include the training cost, so I have very little sense of whether these are serious competitors to existing methods or just interesting curiosities.

The adjoint sampling paper, which this generalizes, is framed as an SOC problem, but this didn't seem to be the central framing of this work. Is there a way to make the connection?

Is the assumption here that we choose a $mathbb{P}_{0,T}$ that has an analytic form, like the Brownian bridge? Otherwise I wasn't clear on how to sample from it in the general case.

**Limitations:**

yes

**Strengths And Weaknesses:**

I found the paper to be extremely well written, and a very useful clarification of the existing literature on matching methods. The main strength of the paper is an original and illuminating formulation of the problem of finding a control u for the transport problem.

The results are clearly presented. However, given that the goal of sampling here is (usually) to compute expectations, I think a more natural measure of success for the sampler would be to report the computational cost of estimating an expectation (e.g. dihedral angles) within a fixed error of the truth.

In terms of soundness, I did not carefully check each proof, but the approach seemed to be sound.


Algorithm 1 doesn't give clear enough operational details, in my opinion. For example, $\rho_{target}$ is not even mentioned in it, but obviously the algorithm must depend on the unnormalized target distribution, and it's really important for the reader to see exactly how (I presume through \Pi^*?). Currently this requires tracing through the paper. It would be great for the authors to clarify how and where $\rho_{target}$ enters in their rebuttal.

---

> ### Author Rebuttal · Authors · 2026-03-30
>
> We thank the reviewer for taking the time to review our work and for the many helpful comments and suggestions. We hope the following replies address the questions and concerns raised.
>
> ---
>
> **Clarification how $p_{\mathrm{target}}$ enters the algorithm**
>
> We thank the reviewer for the opportunity to clarify this. The distributions $p_{\text{target}}$ and $p_{\text{prior}}$ are set to the terminal and initial time marginals, $\Pi^\*_T$ and $\Pi_0^\*$, respectively. They enter the objective through the path-dependent drift $\xi$, specifically via their score functions (see, e.g., Propositions 2.6 and 2.11).
>
> In the updated manuscript, we have revised Algorithm 1 and Algorithm 2 to explicitly show how the prior and target distributions enter the algorithm.
>
> ---
>
> **Clarification on Alanine Dipeptide results**
>
> The white patches in the figures represent regions of high free energy (very low likelihood) where the reference MD data contains no samples. While both ASBS and BMS show slight overrepresentation in these regions, we believe our results remain significant for the following reasons:
>
> 1. Recent works often rely on pre-training with MD data to learn these systems [1].
> 2. Many approaches require prior knowledge to prevent mode collapse [2].
> 3. Other methods use an internal coordinate representation, which limits transferability between systems (see the next answer for details) [3].
>
> In this context, our results are noteworthy: to the best of our knowledge, we are the first to train ALDP on Cartesian coordinates using only energy evaluations, without any prior knowledge - a setting that is more relevant for practical applications.
>
> Importantly, we found that increasing network capacity and applying damping significantly improves performance. We have updated the results in the revised version to reflect these improvements, which better align the learned density with the MD ground truth. Additionally, we include results for Alanine Tetrapeptide, which roughly doubles the number of atoms compared to Alanine.
>
> ---
>
> **Diffusion/flow-based samplers vs. MCMC/MD methods**
>
> Diffusion- and flow-based samplers represent a very active area of research, compared to MCMC/MD methods, which are long-established approaches for sampling. One notable advantage of diffusion-based samplers is that they target non-asymptotic convergence, whereas MCMC/MD methods can only converge asymptotically - and in practice often do so rather slowly.
>
> While MD may still be more efficient for generating samples from a single system in certain tasks, diffusion-based samplers offer the additional advantage of generalization across different systems (e.g., by conditioning on the molecular graph; see [4]) as well as fast sampling once trained. Classical methods, in contrast, require independent MD simulations for each system and cannot transfer learned information.
>
> We believe that our work represents a significant step toward scalable, transferable Boltzmann generators trained purely from energy evaluations, particularly enabling sampling of very high-dimensional, multimodal target distributions.
>
> ---
>
> **Connection to SOC**
>
> We thank the reviewer for the great suggestion to clarify the connection between our work and the SOC perspective. We added a detailed discussion to the revised manuscript. Here, we provide a brief sketch of the connection.
>
> Let $\Pi^\* \in \mathcal{R}(\mathbb{P})$. SOC problems share the same minimizer as reverse KL minimization, i.e.,
>
> $$
> u^* = \arg\min_u D_{\mathrm{KL}}(\mathbb{P}^u \| \Pi^\*) = \arg\min_u \mathbb{E}_{\mathbb{P}^u} \left[ \int_0^T \frac{1}{2} \|u(X_t, t)\|^2 dt  + \log \frac{\Pi\_{0,T}^*(X\_0, X\_T)}{\mathbb{P}\_{0,T}(X\_0, X\_T)} \right]
> $$
>
> For general couplings and reference processes, it does **not** hold that $\mathbb{P}^\{u^\*}\_T=\Pi^\*\_T = p\_{\mathrm{target}}$. This is because the reverse KL is not marginal preserving. Hence, there is no direct SOC perspective for solving the sampling problem with general $\Pi^* \in \mathcal{R}(\mathbb{P})$.
>
> ---
>
> **Assumption of analytical from of the reference process**
>
> Yes, the path measure $\mathbb{P}$  (induced by the reference process) needs to admit a tractable form of the endpoint-conditioned density $\mathbb{P}_{t|0,T}$ to efficiently sample intermediate states $X_t$ for optimizing the fixed-point loss.
>
> ---
>
> Again, we would like to thank the reviewer for the many helpful questions and their positive feedback. We welcome the opportunity to address any additional concerns or questions they may have.
>
> [1] Akhound-Sadegh et al. "Progressive inference-time annealing of diffusion models for sampling from boltzmann densities.”
>
> [2] Nam et al. "Enhancing diffusion-based sampling with molecular collective variables.”
>
> [3] Schopmans & Friederich. "Temperature-annealed boltzmann generators.”
>
> [4] Klein et al. “Transferable Boltzmann Generators.”

---

> > ### Author Rebuttal · Reviewer_bUR2 · 2026-04-02
> >
> > Thanks, this was clarifying.

---

### Official Review · Reviewer_zWUh · 2026-03-11

**Soundness:** 4
**Presentation:** 4
**Significance:** 2
**Originality:** 3
**Overall Recommendation:** 4
**Confidence:** 4

**Summary:**

This paper deals with the problem of sampling from an unnormalized density in high dimension. It positions itself in the direct line of the adjoint sampling literature by rephrasing sampling as solving a stochastic optimal control problem, with the important computational advantage of not storing intermediate trajectories (unlike variational diffusion-based samplers). The contribution of this paper is twofold:

(i) it provides a very clean and general framework for adjoint sampling, from which previous works (namely AS and ASBS) can be recovered as particular instances;

(ii) it proposes a new way to build the coupling before the reciprocal projection by considering the independent coupling, which allows for arbitrary priors and references and avoids several theoretical or computational constraints of previous coupling choices.

Learning is performed iteratively through a fixed-point procedure, whose stability is improved by introducing a damping factor. The performance of the algorithm is evaluated on various synthetic and real-world high-dimensional sampling problems. Overall, I find the paper clear, well written, and conceptually interesting.

**Compliance With Llm Reviewing Policy:**

Affirmed.

**Final Justification:**

The rebuttal addressed my main concerns. Although, I will not champion the paper for acceptance due to some (but acknowledged) limitations.

**Key Questions For Authors:**

* I find the term "fixed point" potentially slightly misleading. While the optimal control satisfies a fixed-point relation, the associated operator is not shown to be contractive or stable in a formal sense.

* Are the authors aware of [3], which, similarly to Proposition 2.11, proposes a control variate approach for TSM?

* How is the neural network initialized? Initialization seems potentially important for the fixed-point iterations, and many adjoint samplers rely on pretraining.

* What is the practical difference with the following procedure: at step $i$, one uses a control $u_i$ to generate approximate samples from the target, treats these samples as a dataset, and then trains using DSM or TSM (or a mixture) to obtain a new control $u_{i+1}$ for a few epochs and then loop ? In other words, how does this differ from training a diffusion model with periodically refreshed model-generated data?

* In the beginning of the related work section, the paper states that many diffusion-based approaches "require keeping SDE simulations in memory for gradient computation." I believe this statement is somewhat inaccurate or at least too broad given the cited works. While many of the methods computing a divergence between the forward and backward paths indeed require simulating trajectories to compute each gradient step, they typically do not store full SDE paths in memory (the objective is evaluated on-the-fly, leading to a memory cost of order batch size x dimension). Moreover, some of the cited papers do just rely on a buffer to be noised: for example, He et al. (2024) and Zhang et al. (2025a,b) generate noisy samples directly via the bridge; OuYang et al. (2024) and Wang et al. (2024) rely on buffers of terminal samples combined with bridge sampling; and Kim et al. (2024) adopts a hybrid strategy. It would therefore be helpful to refine this paragraph to more accurately reflect the diversity of approaches in the literature.

*Remark*: I would like to clearly state to the AC and to the authors that I appreciate the clarity of the paper and find the idea genuinely interesting. My below-the-bar score is driven solely by the experimental comparison. The tasks themselves are well chosen (with the caveat about Gaussian mixtures mentioned above), but the limited comparison with strong and widely used state-of-the-art samplers makes it difficult to fully assess the practical impact. If this aspect is improved during rebuttal, I would be very willing to raise my score above the bar.

**References**

[1] Saifuddin Syed, Alexandre Bouchard-Côté, Kevin Chern, & Arnaud Doucet. (2025). Optimised Annealed Sequential Monte Carlo Samplers.

[2] Syed, S., Bouchard-Côté, A., Deligiannidis, G., & Doucet, A. (2021). Non-Reversible Parallel Tempering: A Scalable Highly Parallel MCMC Scheme. Journal of the Royal Statistical Society Series B: Statistical Methodology, 84(2), 321-350.

[3] Khaled Kahouli, Romuald Elie, Klaus-Robert Müller, Quentin Berthet, Oliver T. Unke, & Arnaud Doucet. (2025). Control Variate Score Matching for Diffusion Models.

**Limitations:**

See weaknesses and questions

**Strengths And Weaknesses:**

**Strengths** :

* The paper is very well written and easy to follow.

* The results appear sound and carefully justified.

* The contributions are well motivated and significantly improve both the performance and the conceptual understanding of adjoint sampling methods.

* Proposition 2.5 is particularly nice, as it introduces a balance between the TSM and DSM identities via control variates with an arbitrary coupling.

* The experiments are conducted on meaningful real-world problems and include ablation studies.

* The conclusion that the independent coupling performs best retrospectively sheds light on the practical trade-offs of previous works (notably ASBS and its heavier computational requirements).

**Weaknesses** :

* While the experimental tasks are well chosen and carefully executed, the comparison with the broader state of the art is limited. PIS and DDS have been significantly improved (e.g., DIS, SCLD), iDEM is widely recognized as underperforming in practice, and several recent neural samplers have been proposed (PITA, PTSD, Diffusion APT, etc...). The comparison is mostly restricted to adjoint sampling variants. It would be important to compare against practical samplers that are widely used, such as AIS, SMC [1], or Parallel Tempering [2], ideally under a matched energy/gradient evaluation budget.

* Related to the previous point, the computational cost of the method appears quite large, especially in terms of gradient evaluations. A clearer discussion of cost–performance trade-offs would be valuable.

* The damping factor is presented as a key ingredient for stability, but it also appears to be a sensitive hyperparameter. This could raise practical concerns.

* There are no theoretical guarantees that the final samples are drawn from the correct target distribution; rather, they are generated from a trained model. It would be helpful to clarify whether the learned sampler could be used as a proposal in a trustworthy Monte Carlo estimator.

* Although the paper states that it assumes access to the target density up to a normalizing constant, in practice the algorithm only uses the score throughout. Since the score carries less information than the density (for example, mixture distributions with different weights can share the same score field almost surely), this may raise issues in multimodal settings. The mode TVD results in Figures 1 and 5 partially address this, but given the equal weighting and limited number of modes, changes in TVD may reflect mode collapse but not imbalance. Figure 3 already suggests potential mode balance issues. It could be informative to compare the output against Langevin chains run within each mode (including for the peptide experiments), and then compare the aggregated samples (completely blind to the balances) to those of the proposed sampler.

* The experiments always use Brownian motion with a Gaussian prior, which does not fully illustrate the flexibility of the more general bridge formulation introduced in the paper.

---

> ### Author Rebuttal · Authors · 2026-03-30
>
> We thank the reviewer for the helpful comments.
>
> **Important**: Due to the character limit, we could not address all points in detail. Please let us know which aspects to prioritize for a more detailed follow-up.
>
> ---
>
> **Comparison to widely used samplers under a matched target evaluation (TE) budget**
>
> We ran AIS, SMC, and PT-MCMC with MALA transitions, using the same number of TEs:
>
> GMM: SMC suffers from severe mode collapse due to resampling. AIS avoids mode collapse but performs worse than BMS. PT-MCMC performs on par.
>
> particle-sys: MC methods underperform BMS on all systems. We also added results for SCLD along with other neural samplers.
>
> ALDP: Comparing against MC methods under matched TEs is challenging as the evaluation is conducted with $10\^6$ samples, which only leaves ~10 evals per particle for the transport. We highlight this as an additional strength of amortized samplers that can easily generate new samples once trained.
>
> We are happy to provide details in a follow up.
>
> Lastly, while such comparisons are valuable, we emphasize that neural samplers offer key advantages: non-asymptotic convergence and the potential to generalize across systems (e.g., via conditioning), which classical MC methods cannot provide.
>
> ---
>
> **Clearer discussion of cost-performance trade-offs in terms of TEs**
>
> We revised the manuscript by (i) adding a table with TEs, and (ii) including figures showing performance versus the TEs.
>
> GMM: We used $3\times10^7$ TEs while the algorithm (typically coverged after $6 \times 10^6$ TEs; see Fig. 7))
>
> particle-Sys./ALDP: $10^6/10^7$ TEs
>
> By comparison, prior work [1] reports that related methods often require $>10^9$ TEs to learn a GMM in 2d.
>
> ---
>
> **Sensitivity of the damping parameter**
>
> We found the damping parameter to be robust. As shown in Figs. 7b,d, values between 50 and 500 yield consistently strong performance. If set too low, instability appears early in training, enabling quick adjustment through a brief parameter search.
>
> ---
>
> **Method as MC proposal**
>
> Yes, the proposed method can be used as a proposal by leveraging forward-backward paths, e.g., by (post-hoc) learning of $v$. We outlined the details in a dedicated section on importance sampling in the appendix of the revised manuscript.
>
> ---
>
> **Mode balance using score-based methods**
>
> Having a smooth transport between prior and target partially remedies the blindness of the score to the relative weight of well-separated modes, as they share overlap for small timesteps, but we agree it remains a challenge for score-only methods. We have added a dedicated discussion on this limitation to the revised manuscript.
>
> Regarding Fig. 3, we obtained significantly better results by incorporating damping and using larger architectures. We kindly refer to our answer to Reviewer bUR2 for further details.
>
> ---
>
> **Leveraging the flexibility of the framework**
>
> The framework allows for small diffusion coefficients, which we found to markedly improve training stability. In contrast, VE/VP schedules with a Gaussian prior did not yield reasonable results. Although not explored here, this flexibility may enable new applications in the future.
>
> ---
>
> **Contraction of fixed-point iteration**
>
> While the optimal drift satisfies a fixed-point relation, formal proofs of contractivity or convergence remain open (similar to AS/ASBS). However, our experiments consistently show stable convergence, especially when using our damping strategy.
>
> ---
>
> **Connection to Control Variate Score Matching (CVSM)**
>
> Yes, we are aware of (and cited) CVSM. We obtain a slightly more general expression than CVSM by combining the generalized TSI (Prop 2.5) and the DSI (Prop B.1). We have added a discussion about the connection to CVSM (and related works) in the revised manuscript.
>
> ---
>
> **Initializiation of the drift model**
>
> For all experiments, we initialized $u$ as zero (no pretraining).
>
> ---
>
> **Suggested alternative procedure based on TSM/DSM**
>
> The main difference between the reviewer’s proposed approach and ours is that, in general, the optimal $u$ is not equal to the score but rather satisfies Nelson’s relation (see Eq. 16). Therefore, using TSM/DSM is not sufficient as one has to account for the backward drift $v$.
>
> Although the reviewer's suggested TSM/DSM approach works for denoising diffusions (VE/VP) where $v\approx0$, those models required high diffusion coefficients that proved unstable in our tests.
>
> ---
>
> **Too broad characterization of related works**
>
> We revised this section to provide a more granular characterization of cited methods. Specifically, we discuss different loss functions and categorize methods with matching-based objectives by specific requirements, such as $\sigma /p_{\mathrm{prior}}$ flexibility, use of importance weights, and whether they require multi-stage optimization.
>
> ---
>
> [1] He, Jiajun, et al. "No trick, no treat: Pursuits and challenges towards simulation-free training of neural samplers.”

---

> > ### Author Rebuttal · Reviewer_zWUh · 2026-04-03
> >
> > First, I would like to thank the authors for the thorough and thoughtful rebuttal—especially given the tight space constraints. I also apologize for the delay in my response.
> >
> > As a general remark, it would be very helpful to share the new experimental results via an anonymous PDF link (containing only the additional material, not the full revised manuscript), as this would make it easier to assess the updates.
> >
> > Before going point by point, I would also like to comment on the discussion with reviewers d3Fd and bURZ. In particular, I would encourage the authors to be more cautious in positioning the ALDP results: while they are interesting within the neural sampling literature, ALDP is generally considered a relatively simple benchmark for classical Monte Carlo methods.
> >
> > > Comparison to widely used samplers under a matched TE budget
> >
> > Thank you for including comparisons with AIS, SMC, and PT-MCMC.
> >
> > * The reported SMC mode collapse on Gaussian mixtures with a large TE budget is quite surprising. Existing scaling results (e.g., [1]) would suggest that SMC should handle such settings more robustly.
> >
> > * For ALDP, I did not fully understand the explanation regarding the difficulty of matching TE budgets. The argument about amortization is valid, but it would still be useful to better contextualize the comparison.
> >
> > More generally, I agree with the transferability argument for neural samplers. However, the "non-asymptotic" nature is also a limitation, as it comes without guarantees on sampling correctness. As discussed with reviewer bURZ, neural samplers are a very active research direction, but we are still far from replacing classical (annealed) MCMC methods, and carefully monitoring this gap remains important.
> >
> > > Cost–performance trade-offs
> >
> > Thank you for adding TE-based comparisons and plots. That said, the reported budgets still appear quite high. While this may be acceptable if no significantly better baselines exist under similar constraints, it is worth noting that recent large-scale evaluations (e.g., [4]) report strong performance on similar toy problems with $10^5$ evaluations or fewer. Still, I appreciate the added clarity.
> >
> > > Sensitivity of the damping parameter
> >
> > Thank you for the clarification, this addresses my concern.
> >
> > > Method as a Monte Carlo proposal
> >
> > Thank you for clarifying this point and for adding the discussion on importance sampling. This helps mitigate concerns regarding theoretical guarantees.
> >
> > > Mode balance and score-based limitations
> >
> > Thank you for expanding the discussion. I still partially disagree with the explanation. While it is true that diffusion models can recover mode weights due to overlap at intermediate noise levels (which means that the drift of the denoising dynamics knows the weights), this is not guaranteed uniformly across all noise scales.
> >
> > More importantly, my concern is slightly different: since the input to the algorithm is the score, which does not depend on mixture weights, the output should, in principle, be invariant to changes in those weights. This raises the question of what additional benefit the method provides over running independent MCMC chains per mode.
> >
> > That said, I appreciate that this limitation is now explicitly acknowledged.
> >
> > > Flexibility of the framework
> >
> > Thank you, this is a nice point.
> >
> >
> > > Fixed-point convergence
> >
> > Thank you for the clarification. As a suggestion, it could be interesting to empirically study convergence of the learned control toward the optimal control in settings where the latter is known (e.g., Gaussian or Gaussian mixture cases).
> >
> > > CVSM connection
> >
> > Thank you for the clarification, and apologies for missing the reference earlier.
> >
> > > Initialization
> >
> > Clear, thank you.
> >
> > > Alternative TSM/DSM procedure
> >
> > Thank you for the clarification. I still believe that a closely related approach could be obtained by learning $v$ in a stochastic interpolant fashion using model-generated data (even if the loss isn't target-informed). Also, based on your response, it seems that in diffusion-like regimes the approaches may become quite similar, could you confirm or clarify this point?
> >
> > > Related work
> >
> > Thank you for improving this section.
> >
> > Overall, I appreciate the authors' efforts in addressing both my comments and those of other reviewers. The addition of MC comparisons, improved discussion of limitations, and clarification of several points are valuable.
> >
> > Given these updates, I am willing to increase my score to weak accept. That said, I would still appreciate answers to the new questions raised above, as well as access to the updated experimental results (e.g., via an anonymous PDF).
> >
> > [4] Blessing et al. (2024). *Beyond ELBOs: A Large-Scale Evaluation of Variational Methods for Sampling*. ICML.

---

> > > ### Author Response · Authors · 2026-04-08
> > >
> > > We thank the reviewer for the many helpful comments, suggestions, and questions. Below, we provide clarifications and responses to the specific questions raised in the rebuttal acknowledgment.
> > >
> > > ---
> > >
> > > > As a general remark, it would be very helpful to share the new experimental results via an anonymous PDF link
> > > >
> > >
> > > We thank the reviewer for the suggestion. We were previously not aware that link sharing is permitted. Please find the results under the following link: https://anonymous.4open.science/r/ICML2026-rebuttal-819E/
> > >
> > > ---
> > >
> > > > I would encourage the authors to be more cautious in positioning the ALDP results: while they are interesting within the neural sampling literature, ALDP is generally considered a relatively simple benchmark for classical Monte Carlo methods.
> > > >
> > >
> > > We agree and will revise the manuscript to make explicit that our results should be interpreted in the context of the neural sampling literature.
> > >
> > > At the same time, we would like to emphasize that, to the best of our knowledge, no neural sampler that does not rely on reaction coordinates or pretraining has previously achieved comparable performance on ALDP. We therefore believe these results represent a meaningful step toward improved scalability in this setting.
> > >
> > > ---
> > >
> > > > Existing scaling results (e.g., [1]) would suggest that SMC should handle such settings more robustly.
> > > >
> > >
> > > We did not find SMC to be robust for these high-dimensional multimodal targets, as resampling led to severe mode-collapse. This is also a finding that was reported in [1] (”M1) Resampling causes mode collapse in high dimensions”).
> > >
> > > ---
> > >
> > > > For ALDP, I did not fully understand the explanation regarding the difficulty of matching TE budgets.
> > > >
> > >
> > > To get statistically significant results on ALDP we used $10^6$ samples for the evaluation. Transporting $10^6$ samples from prior to target using SMC/AIS under the same TE budget ($10^7$ TEs) leaves only 10 TE per particle, which is typically far from sufficient.
> > >
> > > However, we additionally ran an MD simulation using the same number of TEs and compared to the results we obtained with neural samplers. Please find these results in the link posted above.
> > >
> > > ---
> > >
> > > >
> > > >
> > > >
> > > > […] This raises the question of what additional benefit the method provides over running independent MCMC chains per mode.
> > > >
> > >
> > > We agree that if all modes of the target density were known, then alternative approaches might be preferable. However, the main goal of this work is to learn a sampler without any prior knowledge of the target density.
> > >
> > > ---
> > >
> > > > […] based on your response, it seems that in diffusion-like regimes the approaches may become quite similar, could you confirm or clarify this point?
> > > >
> > >
> > > If the reviewer refers to denoising diffusion models with ‘diffusion-like regimes’, then, yes, both approaches are related. Specifically, denoising diffusions can be reformulated such that they fit in the Schroedinger half-bridge category of our framework with an Ornstein-Uhlenbeck reference process; see [2]. The memorylessness is ensured here by adding sufficient noise such that the prior is (approximately) Gaussian.
> > >
> > > Directly regressing onto the TSI also seems valid for denoising diffusion models. We are currently investigating how this fits into our framework. We thank the reviewer for the comment and will provide further details in the revised version.
> > >
> > > ---
> > >
> > > > As a suggestion, it could be interesting to empirically study convergence of the learned control toward the optimal control in settings where the latter is known (e.g., Gaussian or Gaussian mixture cases).
> > > >
> > >
> > > We thank the reviewer for the great suggestion. We used the closed-form solution for the optimal drift for Gaussian to Gaussian mixtures provided in ([3], Appendix A) to compute the drift error
> > >
> > > $$
> > > \text{drift err} \coloneqq \mathbb{E}_{\mathbb{P}^u}\left[\int_0^T\|u(X_t,t)-u^*(X_t,t)\|^2 \mathrm dt\right].
> > > $$
> > >
> > > We added preliminary results to the link shared above and will continue to investigate the empirical convergence based on this drift error. Note that every component contributes to the drift error, and in $d=100$ are convergence results seem particularly strong.
> > >
> > > ---
> > >
> > > We want to sincerely thank the reviewer for taking the time to review our paper and for the many great suggestions that have greatly improved our manuscript.
> > >
> > > ---
> > >
> > > [1] Blessing et al. (2024). "Beyond ELBOs: A Large-Scale Evaluation of Variational Methods for Sampling."
> > >
> > > [2] Vargas, Francisco, Will Grathwohl, and Arnaud Doucet. "Denoising diffusion samplers.”
> > >
> > > [3] Albergo, Michael, Nicholas M. Boffi, and Eric Vanden-Eijnden. "Stochastic interpolants: A unifying framework for flows and diffusions.”

---

### Official Review · Reviewer_d3Fd · 2026-03-12

**Soundness:** 3
**Presentation:** 3
**Significance:** 1
**Originality:** 2
**Overall Recommendation:** 2
**Confidence:** 3

**Summary:**

This work studies diffusion-model-based sampling from unnormalized target densities, where recent scalable approaches often rely on least-squares “matching” losses but introduce practical compromises (e.g., constrained priors) or suffer from optimization instability. The authors reinterpret these prior methods through the lens of fixed-point iterations derived from Nelson’s relation, and propose a new formulation that subsumes existing techniques while removing key limitations. Concretely, the method learns a stochastic transport map between arbitrary prior and target distributions using a single objective designed to be both scalable and stable. To address failure modes such as mode collapse, a damped fixed-point variant is introduced with an added regularization term intended to preserve diversity. Experiments are reported to show strong scalability and improved mode coverage, yielding state-of-the-art performance on challenging synthetic unnormalized densities and high-dimensional molecular benchmarks.

**Compliance With Llm Reviewing Policy:**

Affirmed.

**Final Justification:**

I still feel more novelty, clarification, and large-scale experiments are needed.

**Key Questions For Authors:**

see weaknesses

**Limitations:**

see weaknesses

**Strengths And Weaknesses:**

Strengths
- The paper is rich in solid theoretical proofs and rigorous measure-theory definitions.

Weaknesses
- My main concern: The core improvement (from Eq 15 -> Eq 24), which is the so-called fix-point updates, seems like an EMA trick to the learned SI models $u(x,t)$. Therefore, it is not that novel to me. The key novel improvement needs to be highlighted more.
- I am not sure what the meaning of "Scalable sampling" is.
- The experimental validation is limited to dimension and scale. Can this approach be extended to high-dimension image dataset?

---

> ### Author Rebuttal · Authors · 2026-03-30
>
> We thank the reviewer for taking the time to review our work. We hope the following replies address the questions and concerns raised.
>
> ---
>
> **Damping as EMA “Trick”**
>
> We agree that the damped update is a simple and intuitive modification, which can be understood as an exponential moving average. While, to our knowledge, this formulation has not been applied in this context before, we do not consider it the primary novelty of our work. It is an extension that stabilizes the training of the much more significant underlying framework.
>
> Moreover, we took the following actions:
>
> - We mentioned the connection between damping and EMA in the revised version
> - We compared damping to EMA in the parameter space which is more common in existing literature. However, damping performed significantly better
> - We added a theoretical justification for the damping: We prove that damping reduces the variance of the matching objective (see eq. (176) for a bias-variance decomposition)
>
>
> ---
>
> **Highlighting key novelties**
>
> To better highlight the fundamental contributions of our work, we have added a dedicated contribution statement to the introduction. Our primary novelties are the following :
>
> 1. **Theoretical:** We present the first framework that applies Markovian projections to the setting where no data from the target distribution is available. In other words, we extend the celebrated concept of *stochastic interpolants* to the significantly more challenging data-free scenario. As a valuable side effect, our framework also allows for the generalization of previously existing samplers (e.g., AS and ASBS).
> 2. **Algorithmic:** We introduce the Bridge Matching Sampler (BMS), a scalable method that enables stochastic transport between arbitrary prior and target distributions using a single, stable objective. Compared to previous attempts, we reach significantly improved performance in higher dimensions. To the best of our knowledge, there is no diffusion-based sampler which can scale to $d=2500$ on multimodal tasks.
> 3. **Practical:** We propose a damped iteration scheme that significantly stabilizes training and mitigates mode collapse in high-dimensional settings, allowing us to tackle sampling problems that were previously unattainable.
>
> We believe the paper's strength lies in the synergy between our new theoretical framework and the empirical results in practically relevant examples as well as in very high dimensions. To the best of our knowledge, no other method achieves comparable results on ALDP when trained on Cartesian coordinates using only the system’s energy. Many related approaches rely on additional prior knowledge, such as MD simulations at high temperatures, knowledge of collective variables, or internal coordinate representations, which substantially reduce the task complexity. Furthermore, since the submission, we have made significant improvements over the results presented in the submission including a) new evaluation criteria, including further discrepancies computed on the Ramachandran and TICA plots, and b) added results for Alanine Tetrapeptide, which has roughly double the number of atoms. We included these results in the revised version of the manuscript.
>
> ---
>
> **The meaning of "Scalable sampling"**
>
> “Sampling” refers to the task of sampling from an unnormalized density without access to a dataset, which is quite different from the more common task of learning a generative model with samples from a dataset. As many algorithms for training diffusion and flow models for sampling from an unnormalized density are quite restricted in terms of problem dimensionalities that can be tackled, the term "scalable sampling" refers to the property that a method can scale to much larger problem dimensionalities and is also used frequently in related literature; see e.g. [1, 2].
>
> ---
>
> **Extension to a high-dimensional image dataset**
>
> Connecting to the previous answer, the goal of this work is not to learn a generative model from a dataset, but rather from an unnormalized density. Training from high-dimensional image datasets falls into the former category and is therefore different from the setting considered in this work. However, the dimensionalities that are considered in this work in the setting of sampling from unnormalized densities are much higher than those considered in previous work. For reference, e.g. [2], [3], and [4] have been published very recently and consider problems of size $d <200$, whereas our work considers problems with up to 2500 dimensions.
>
> ---
>
> **References:**
>
> [1] Akhound-Sadegh et al. "Iterated denoising energy matching for sampling from Boltzmann densities.” ICML 2024
>
> [2] Havens et al. "Adjoint sampling: Highly scalable diffusion samplers via adjoint matching.” ICML 2025
>
> [3] Liu et al. "Adjoint Schrödinger Bridge Sampler.” NeurIPS 2025 (Oral)
>
> [4] Guo et al. "Proximal diffusion neural sampler.” ICLR 2026

---

> > ### Author Rebuttal · Reviewer_d3Fd · 2026-04-04
> >
> > Thanks for your reply. I still feel more novelty, clarification, and large-scale experiments are needed.

---

> > > ### Author Response · Authors · 2026-04-07
> > >
> > > Dear Reviewer d3Fd,
> > >
> > > We thank the reviewer for the additional feedback. However, we believe there remain fundamental misunderstandings regarding the problem setting and the standards of the sampling community that we would like to clarify.
> > >
> > > **Problem Setting (Densities vs. Datasets):** The request for "high-dimensional image datasets" suggests a confusion between **Generative Modeling** (learning from data samples) and **Neural Sampling** (generating samples from a known unnormalized density). Our work falls into the latter; therefore, image datasets are not a standard or applicable benchmark for the methodology presented.
> > >
> > > **Experimental Scale:** We have demonstrated our method on the most high-dimensional problems currently available in neural sampling literature. Expecting "large-scale" experiments equivalent to those in Computer Vision is a category error, as the computational constraints and objectives of sampling from complex densities are fundamentally different.
> > >
> > > We are concerned that the current **Confidence Score (3)** does not reflect these discrepancies in domain-specific expertise. While we welcome critical feedback on our technical proofs or methodology, we respectfully submit that the current critique is based on expectations from a different field of research.
> > >
> > > If these clarifications do not change your evaluation of the work's quality, we kindly suggest that a confidence score of **1 or 2** would more accurately reflect the alignment between the review’s requirements and the paper’s actual research domain.

---

### Official Review · Reviewer_kis2 · 2026-03-13

**Soundness:** 3
**Presentation:** 1
**Significance:** 1
**Originality:** 3
**Overall Recommendation:** 3
**Confidence:** 2

**Summary:**

This paper tackles the problem of learning Boltzmann generators without relying on sampling data.
The general setting is to have the target distribution in closed  form except for the normalization
constant. The goal is then  to see if one can construct a diffusion model from an easy to sample prior distribution with the additional
sophistication  that both the forward and backward processes are learned (If I understand well).
As a result in this framework two controls for the Langevin equation have to be learned and a constraint is imposed on the learning that the forward and backward distributions should coincide.

**Compliance With Llm Reviewing Policy:**

Affirmed.

**Key Questions For Authors:**

- I don't understand how eq (23) is used in practice? Is P_{t\vert 0} given  or is it learned
(which is what I thought it was supposed to be but I am not sure now) or is this simply the time scheduling seemingly encoded into
some mysterious function c(t) and gamma(t)?

- I don't actually get the starting principle which is why is (3)-(4) a desirable setting. Why not for instance looking for an optimal transport between P_0 and P_T? basic premise are not sufficiently discussed to my viewpoint.

- In practice what is the loss which is used? I see a loss for the Markov projection but is that all?

- Gaussian mixture experiment seems miningless to me  since the target distribution is given and easy to sample. What is P_0 by the way in this experiment?

- What is P_0 for the Alanine experiment?

- Alinine seems to be kind of very small scale, how many parameters? results seems poor compared to other methods.

- Actually the metrics used in the Alanine experiments are not comparable to those used in other state of the art papers on this kind of problems (see e.g. von Klitzing et al. ICLR 2026).
Aren't there any possible way to make things in this domain quantitatively comparable?

**Limitations:**

no

My suggestions:
- it would be worth to put this work in better perspective with other methods.
- if possible illustrate the method in a context where sample based methods are inapplicable.
- a possible alternative presentation: start from what would be ideally learned to perform the task and then the approximations which are made in practice (projection onto a Markov process)

**Strengths And Weaknesses:**

This paper  pertains to a domain research  I am not familiar with and I found this paper quite complex. It appeared to me as a puzzle with many pieces scattered around and I could not get the high level picture to be willing to fully enter the details. As a result  I will be able to ask only naive questions (see below). The strong point seems to be that the method does not require samples from the target distribution  which are usually difficult to sample but uses only the un-normalized density function i.e. the Boltzmann energy and seems to perform comparatively better than methods of the same family. Experiments on Gaussian mixtures show good performances at high dimensions.
On the other end it looks like this kind of methods lag far behind in terms of performance compared to state of the art methods making use of at least some sampling data, when applied to the Alanine study case example of the experimental section.
So considering the complexity of the approach I would imagine it to be  of actual interest only in special cases not considered in the experimental part. At least a  broader discussion on the merits of  this approach and how it is situated in the vaster realm of Boltzmann generators would maybe help to motivate the reader. One of the main result seems to be Equation (23) which looks indeed to have a convenient and tractable form, but coming after a lengthy and tortuous development to achieve a quite intuitive result:
compared to Markovian diffusion model, the difference seems only to include terms corresponding to the end points, the last one being seemingly the usual score, If I did not misunderstood?

---

> ### Author Rebuttal · Authors · 2026-03-30
>
> We thank the reviewer for the helpful comments.
>
> **Important:** Due to the character limit, we could not address all points in detail. Please let us know which aspects to prioritize for a more detailed follow-up.
>
> ---
>
> **Worse performance compared to data-based methods**
>
> We emphasize that we study the classical sampling problem, which assumes no access to data from the target distribution. This setting is considerably more challenging than scenarios where target samples are available, as it requires effective exploration and identification of the modes of the target distribution without guidance.
>
> In comparison to other diffusion-based sampling methods that operate without data, our approach achieves SOTA performance in dimensions that, to the best of our knowledge, have not previously been reached.
>
> ---
>
> **Method is only relevant in special cases and GMMs seem meaningless**
>
> The tasks considered (GMMs, DW, LJ) are standard benchmarks that are widely used in recent related literature. Our work goes a step further by
>
> a) increasing the dimensionality of the GMM targets substantially (from the typical range of $d=20$–$50$ to $d=2500$), and
>
> b) considering molecular Boltzmann densities without knowledge of reaction coordinates, thereby moving significantly closer to practically relevant scenarios.
>
> For both GMMs and Alanine Dipeptide, we set $\mathbb{P}_0$ as a Gaussian. Details of the experimental setup can be found in Appendix D.
>
> ---
>
> **Context within Boltzmann generators**
>
> We thank the reviewer for the suggestion and have added a detailed discussion to the manuscript.
>
> ---
>
> **Is  $\mathbb{P}\_{t|0}(x_t|x_0)$ given or learned? Is $\nabla \log \mathbb{P}\_{0|t}$ the usual score?**
>
> We thank the reviewer for the opportunity to clarify this. $\mathbb{P}_{t|0}$ is the transition density of the reference process $\mathrm dX_t = \sigma(t)\,\mathrm dB_t$, which is simply a Gaussian distribution; see (188).
>
> The score $\nabla \log \mathbb{P}_{t|0}$ in (23), whose conditional expectation is related to the backward drift $v$ via Proposition 2.4, can therefore be computed analytically without any computational overhead. We have added a clarifying sentence after Prop. 2.6 in the manuscript.
>
> ---
>
> **Why is (3)-(4) desirable? Why not optimal transport?**
>
> (3)-(4) is desirable as it constructs a transport between a prior and a target distribution, where one can tractably sample intermediate states $X_t$, allowing for efficient simulation-free training procedures analogous to flow/bridge matching. Entropic optimal transport (Schroedinger bridge) arises as a special case of the framework (Section 2.3).
>
> We updated the introduction to more clearly articulate the underlying motivation for our approach.
>
> ---
>
> **Is the loss for the Markov projection the only loss?**
>
> Yes, the loss function of the algorithm is the regression loss between $u$ and $\xi$. Although the derivation of the path-dependent drift $\xi$ may appear somewhat involved, the resulting algorithm is in fact straightforward to implement. We kindly refer to Algorithm 2 for a detailed description.
>
> ---
>
> **Dimensionality of Alanine Dipeptide (ALDP) and clarification of results**
>
> Alanine has 66 dimensions (or 22 atoms). We kindly refer to our answer to Reviewer bUR2 regarding the result discussion of ALDP.
>
> ---
>
> **The metrics used in the ALDP experiments are not comparable to those used in other papers**
>
> The work of von Klitzing et al. has access to model likelihoods, as they train a normalizing flow, allowing them to compute metrics such as ESS, EUBO, and others - something that is not straightforward for diffusion-based models. Moreover, they train on internal coordinates, which is significantly easier and less practically relevant, as it does not allow for transferability between systems.
>
> In the revised manuscript, however, we have added additional evaluation metrics commonly used in von Klitzing et al. and related works, including TICA-Wasserstein (TICA-W2) distances and free energy differences
>
> ---
>
> **Provide context w.r.t. other methods**
>
> We revised the related work section to provide further context. Specifically, we discuss different loss functions and categorize methods with matching-based objectives by specific requirements, such as $\sigma /p_{\mathrm{prior}}$ flexibility, use of importance weights, and whether they require multi-stage optimization.
>
> ---
>
> **Illustrate method in context where sample-based methods are inapplicable**
>
> We emphasize that we study the classical sampling problem, which is considerably more challenging - but also more relevant for many scientific applications - than settings in which data from the target distribution is available (see above).
>
> ---
>
> **Approximations only later in text**
>
> We note that the Markovian projection does not constitute an approximation, but rather defines our learning strategy. However, as mentioned above, we updated the introduction to more clearly articulate the underlying motivation

---

> > ### Author Rebuttal · Reviewer_kis2 · 2026-04-03
> >
> > Thank you for these clarifications, which have helped me better grasp the paper’s content.
> > Despite the merits of the theoretical approach, and even if I understand the constraint of not using generated data as well as the incremental improvements over concurrent methods of the same type,  I find the empirical performances lagging too far behind to position this work as a significant advancement in the broader ML landscape.

---

> > > ### Author Response · Authors · 2026-04-07
> > >
> > > Dear Reviewer kis2,
> > >
> > > Thank you for taking the time to read and acknowledge our rebuttal. We would, however, like to respectfully address your statement that "the empirical performance lags too far behind to position this work as a significant advancement in the broader ML landscape."
> > >
> > > Based on our results, we believe this assessment does not accurately reflect the empirical findings.
> > > Our method demonstrates clear and consistent improvements over existing approaches, in particular over the very recently published ASBS method (Oral, NeurIPS 2025). Moreover, we show that even the performance of ASBS itself can be significantly improved by incorporating the damping scheme introduced in our Section 2.5. We have carefully investigated this effect across multiple experiments (see, e.g., Figures 1, 4, and 7).
> > >
> > > We agree that, from a data-driven perspective, the benchmark systems considered in this work may appear small or not directly application-oriented. However, we would like to emphasize that the data-free setting considered here is substantially more challenging than scenarios where target samples are available. In this setting, the algorithm must autonomously explore the state space and identify the modes of the target distribution without any guidance from data.
> > >
> > > Neural samplers constitute a very active research area, and to the best of our knowledge, it has previously not been possible to successfully learn systems of the type considered here without incorporating substantial prior knowledge. We therefore view the numerical results as a central strength of the paper. We also kindly refer to the assessments of the other reviewers in this regard.
> > > In fact, we are not aware of any neural sampling work that achieves comparable performance in similarly high dimensions and without assuming prior knowledge of the target.
> > > For these reasons, we consider our work to represent a meaningful advancement in the landscape of neural sampling methods. We hope this clarification helps motivate a reconsideration of the evaluation.
> > >
> > > We sincerely thank you again for the time and effort invested in reviewing our work.

---

### Decision · Program_Chairs · 2026-04-30

**Decision:**

Accept (regular)

**Comment:**

This paper proposes a new sampling algorithm for the Boltzmann Sampling problem. For this meta-review, I relied on Reviewer zWUh's assessment, which matches my own. The crux of the idea lies in essentially training a diffusion model on its own data with Markovian Projections and Target Score Matching. Surprisingly, as the authors demonstrate in their rebuttal experiments, this is a powerful idea and bypasses a lot of the complexities of the Adjoint Sampling literature. I believe these insights are valuable to be shared within the broader community. In particular, I appreciated the empirical demonstration that the variants of AS, with new experiments for AIS, SMC, and PT, are strong. One limitation of this paper is that a large share of the presentation is decorative in that one can largely view this as a diffusion model with Target Score Matching. If the authors are willing to acknowledge that more boldly and be honest about their main contributions, then this work would be stronger for it. Given that this paper has several strong findings, I'm voting to accept this paper, and hope the authors can clarify the writing and add an updated limitations/discussion section around the points I flagged.